# ADA-DIFFUSER: LATENT-AWARE
# ADAPTIVE DIFFUSION FOR DECISION-MAKING

**Fan Feng**[1,3] , **Selena Ge**[1], **Minghao Fu**[1], **Zijian Li**[2,3], **Yujia Zheng**[2], **Zeyu Tang**[2,4],
**Yingyao Hu**[5†], **Biwei Huang**[1†], **Kun Zhang**[2,3†]
[1] University of California San Diego [2] Carnegie Mellon University
[3] MBZUAI, [4] Stanford University, [5] Johns Hopkins University
[†] Equal Senior Authorship

## ABSTRACT

Recent work has framed decision-making as a sequence modeling problem using generative models such as diffusion models. Although promising, these approaches often overlook latent factors that exhibit evolving dynamics, elements that are fundamental to environment transitions, reward structures, and high-level agent behavior. Explicitly modeling these hidden processes is essential for both precise dynamics modeling and effective decision-making. In this paper, we propose a unified framework that explicitly incorporates latent dynamic inference into generative decision-making from minimal yet sufficient observations. We theoretically show that under mild conditions, the latent process can be identified from small temporal blocks of observations. Building on this insight, we introduce `Ada-Diffuser`, a causal diffusion model that learns the temporal structure of observed interactions and the underlying latent dynamics simultaneously, and furthermore, leverages them for planning and control. With a modular design, `Ada-Diffuser` supports both planning and policy learning tasks, enabling adaptation to latent variations in dynamics, rewards, and latent actions. Experiments on locomotion and robotic manipulation benchmarks demonstrate its effectiveness in accurate latent inference, long-horizon planning, and adaptive policy learning[1].

## 1 INTRODUCTION

Learning and planning in partially observable environments is a fundamental challenge in building intelligent agents (Kaelbling et al., 1998). Recent work on casting decision-making as a generative modeling problem, taking advantage of powerful models such as transformers (Chen et al., 2021; Zheng et al., 2022; Kong et al., 2024) and diffusion models (Janner et al., 2022; Chi et al., 2023; Ren et al., 2025), has achieved impressive results in a wide range of tasks. However, these methods often fail to account for hidden latent variables and their temporal dynamics, factors that are prevalent in real-world settings such as robotics (Lauri et al., 2022), autonomous driving (Huang et al., 2024), healthcare (Hauskrecht & Fraser, 2000; Ehrmann et al., 2023), and economics (Brero et al., 2022). Ignoring such latent processes can result in suboptimal decision-making, particularly when the observational data does not provide full coverage of the latent factors underlying the environment's dynamics (Zintgraf et al., 2021; Xie et al., 2021; Swamy et al., 2022; Belkhale et al., 2023).

Early works address partial observability in reinforcement learning (RL) and imitation learning (IL) by encoding historical observations and actions into belief states or latent embeddings, which represent a distribution over the underlying latent state (Kaelbling et al., 1998; Hauskrecht, 2000; Guo et al., 2018; Igl et al., 2018; Liang et al., 2024a; Xie et al., 2021). Policy optimization or planning is then carried out based on these inferred belief states. However, learning such representations often requires access to the historical trajectories or data from a diverse set of environments. This can be prohibitively expensive, particularly in high-dimensional state or action spaces, posing challenges for integrating these methods into modern generative decision-making models, which typically prioritize scalability. Can we *identify* the latent factors that govern environment dynamics and rewards, and

---

[1] Project Page: https://sites.google.com/view/ada-diffuser.

integrate them into *scalable* generative decision-making models to enable adaptive planning and policy learning, using only *minimal observations*, while *preserving theoretical guarantees*?

In this paper, we pursue this goal by addressing two fundamental questions. First, what is the *minimum* set of observations required, *in principle*, to reliably identify the latent factors that govern the environment? Second, how can *latent identification* be effectively incorporated into generative models (e.g., diffusion models) to enable adaptive planning and policy learning? To answer the first question, we theoretically show that, under mild conditions, the latent factors at the time step $t$ can be block-wise identified using only four surrounding observable measurements (i.e., state-action trajectories) within a small temporal window. This identification result implies that a small temporal block is sufficient to infer the latent factors in observational RL trajectories in an *online* manner.

Guided by the theoretical findings, we propose `Ada-Diffuser`, a novel *causal diffusion* framework with latent identification from temporal blocks, designed to model the data generation process of RL trajectories influenced by latent factors. To reflect the autoregressive nature of sequential decision making, we introduce a *causal denoising schedule* that aligns the denoising steps with the underlying causal structure, drawing inspiration from recent advances in autoregressive diffusion models (Ho et al., 2022; Chen et al., 2024; Xie et al., 2024b; Sand-AI, 2025). For *temporal-block-wise latent identification*, during training, we propose a *denoise-then-refine* procedure that iteratively alternates between denoising the observations and refining latent estimates. This enables `Ada-Diffuser` to jointly learn a structured representation of latent variables and the corresponding observational distribution. At inference time, `Ada-Diffuser` generates actions and states while estimating latent variables in an online fashion. Since states and actions are conditioned on the latent factors, we employ a *zig-zag sampling* scheme that alternates between sampling state-action pairs and updating latent variables, ensuring consistency between generated sequences and their underlying latent dynamics.

`Ada-Diffuser` provides a unified generative framework for sequential decision-making. It is applicable to both *planning* and *policy learning* tasks by conditioning on different types of observations and adapting the conditional generative process accordingly. The framework is flexible and can accommodate various forms of latent, including ones that influence dynamics, rewards, or even represent high-level latent actions. Importantly, even in environments without explicitly designed latent variables, the block-wise latent identification mechanism improves generative modeling by implicitly capturing structured temporal dependencies.

**Contributions:** (1) We establish sufficient conditions under which latent factors influencing environment dynamics and rewards can be identified from short temporal windows of RL trajectories, without requiring full trajectory access or multi-environment data. (2) We develop `Ada-Diffuser`, a causal diffusion model that performs block-wise latent inference to jointly model latent contexts and observable trajectories. Unlike prior latent-augmented diffusion approaches, `Ada-Diffuser` introduces a minimal-sufficient block with backward refinement for identifiable latents and uses fully autoregressive denoising with zig–zag sampling to couple inference and generation. (3) `Ada-Diffuser` can be adapted to a wide range of decision-making tasks by conditioning on different types of observation. We empirically show the improved performance on a wide range of planning and control tasks, including 8 environments under 23 different settings.

## 2 BACKGROUND AND RELATED WORK

In this section, we provide background and related work on diffusion-based decision-making. Additional discussions are provided in Appendix E, including related work on (1) learning latent belief states in POMDPs (Kaelbling et al., 1998; Hauskrecht, 2000; Igl et al., 2018; Gregor et al., 2018; Goyal et al., 2021), particularly in the context of transfer, meta, and nonstationary RL/IL (Zintgraf et al., 2021; Xie et al., 2021; Feng et al., 2022; Ni et al., 2023; Liang et al., 2024a), and (2) autoregressive diffusion models (Chen et al., 2024; Xie et al., 2024b; Sand-AI, 2025; Wu et al., 2023).

Recent advances use diffusion models as planners and policies for both RL and IL. *I. Diffusion Planner*: Diffusion-based planning leverages generative models to sample future state-action trajectories from a given state, using guidance techniques (Dhariwal & Nichol, 2021; Ho & Salimans, 2022) to encourage desirable properties such as high expected rewards. Taking Denoising Diffusion Probabilistic Models (DDPM (Ho et al., 2020))-based approaches as an example, these methods learn a generative model over expert trajectories $\tau = \{(\mathbf{s}_0, \mathbf{a}_0), \ldots, (\mathbf{s}_T, \mathbf{a}_T)\}$ by modeling a forward-noising process:

$q(\mathbf{x}^t \mid \mathbf{x}^{t-1}) = \mathcal{N}(\mathbf{x}^t; \sqrt{\alpha_t}\,\mathbf{x}^{t-1}, (1-\alpha_t)\mathbf{I})$, and a parameterized denoising model $p_\theta(\mathbf{x}^{t-1} \mid \mathbf{x}^t)$ to reverse the process. Here, the superscript $t$ denotes diffusion steps, $T$ denotes the planning horizon, $\mathbf{x}^0$ is a clean subsequence sampled from the expert trajectory $\tau$, and $\alpha_t$ controls the variance schedule at diffusion step $t$. During inference, trajectories are generated by starting from Gaussian noise and iteratively denoising through the learned reverse process. This generation can be optionally conditioned on the initial state or other guidance signals $\mathbf{y}$ (e.g., goals, rewards): $\hat{\tau} \sim p_\theta(\tau \mid \mathbf{s}_0, \mathbf{y})$. *II. Diffusion Policy*: In contrast to diffusion planners, Diffusion Policy methods directly parameterize the policy $\pi_\theta(a \mid s)$ using diffusion models. For example, Diffusion Policy (Chi et al., 2023) uses a diffusion model to generate multi-step actions with expressive multimodal distributions. DPPO (Ren et al., 2025) extends this idea by modeling a two-layer MDP structure, which enables fine-tuning of diffusion-based policies in RL settings. Another line of work uses diffusion models to parameterize the policy networks for only the single current step (Wang et al., 2022; Hansen-Estruch et al., 2023; Chen et al., 2023; Lu et al., 2023). `Ada-Diffuser` can generally accommodate both diffusion planner and policies within the same framework.

## 3 LATENT IDENTIFICATION IN POMDP

In this section, we seek to formally model the structure of the decision-making system by answering the following questions. First, where do the latent factors reside, and how do they influence the observable variables such as states, actions, and rewards? Second, can they be identified from demonstration data alone? We model the system that extends the standard MDP to include unobservable, time-varying latent variables that affect both the transition dynamics and the reward function. This model generalizes the contextual MDP by allowing the context to evolve stochastically over time. We then formalize the data generation process under this model using structural causal models (SCMs) (Pearl, 2010). Finally, we present theoretical results that characterize the minimal observational requirements for identifying the latent variables.

### 3.1 LATENT CONTEXTUAL POMDP WITH TIME-DEPENDENT CONTEXT

We model the latent factors using a general contextual MDP framework, where the context itself evolves over time. Formally, we define a latent time-varying contextual MDP as a tuple $\mathcal{M} = (\mathcal{S}, \mathcal{A}, \mathcal{C}, \mathcal{T}, \mathcal{R}, \gamma)$, where $\mathcal{S}$ is the state space, $\mathcal{A}$ is the action space, $\mathcal{C}$ is the latent context space, $\mathcal{T}(\mathbf{s}_t \mid \mathbf{s}_{t-1}, \mathbf{a}_{t-1}, \mathbf{c}_t)$ is the transition distribution, $\mathcal{R}(\mathbf{s}_t, \mathbf{a}_t, \mathbf{c}_t)$ is the reward function, and $\gamma \in [0, 1)$ is the discount factor. The latent context $\mathbf{c}_t \in \mathcal{C}$ follows a time-dependent (possibly stochastic) process: $\mathbf{c}_t \sim p(\mathbf{c}_t \mid \mathbf{c}_{t-1})$, and is unobserved during training and inference. The agent only observes trajectories $\tau = \{(\mathbf{s}_0, \mathbf{a}_0), \dots, (\mathbf{s}_T, \mathbf{a}_T)\}$, and infers the latent context $\mathbf{c}_t$ from the observational data. This is naturally relevant to several MDP models, including (dynamic) hidden parameter MDPs (Doshi-Velez & Konidaris, 2016; Perez et al., 2020; Xie et al., 2021), Bayes-adaptive MDPs (Martin, 1965; Duff, 2002; Zintgraf et al., 2021), and factored MDPs (Guestrin et al., 2003). A full comparison and analysis is given in App. C.

Given trajectories generated under this model, we can describe the data generation process using the SCMs. Without the loss of generality, we consider the setting where an expert policy $\pi$ is assumed to generate the actions, as is standard in learning from demonstration data. The data generation process can therefore be expressed as (l.h.s. Fig. 1):

**Latent Dynamics:**   $\mathbf{c}_t = h(\mathbf{c}_{t-1}, \eta_t),$

**State Transitions:**   $\mathbf{s}_t = f(\mathbf{s}_{t-1}, \mathbf{a}_{t-1}, \mathbf{c}_t, \epsilon_t),$

**Action Generation:** $\mathbf{a}_t = \pi(\mathbf{s}_t, \mathbf{c}_t),$

**Reward Function:**   $r_t = g(\mathbf{s}_t, \mathbf{a}_t, \mathbf{c}_t, \delta_t),$

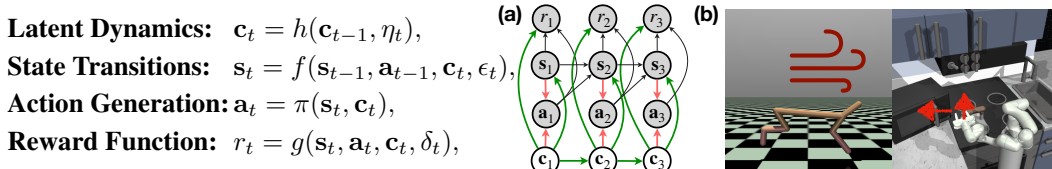

Figure 1: (a) SCM of the Latent Contextual POMDP. Gray/white nodes are observed/latent variables; green/red edges represent transitions driven by latents/expert policies, respectively. (b) Examples where latents influence either dynamics or rewards (affecting optimal actions).

where $\eta_t$, $\epsilon_t$, and $\delta_t$ denote i.i.d. exogenous noise variables. Fig. 1(a) shows the graphical model. Fig. 1(b) illustrates examples where latent factors on dynamics (e.g., external wind in locomotion) and rewards (e.g., varying target objects in robot control) influence optimal decisions.

## 3.2 Identifiability of Latent Factors with Minimal Measurements

To learn accurate dynamics and make reliable decisions, it is essential that the underlying latent factors influencing the environment are identifiable with observational data. We present theoretical results that characterize the minimal number of consecutive observations required for the identifiability of the latent variables, under a set of mild and natural assumptions.

**Assumption 1** (First-order MDP). *We consider the following conditions:*

$$P\left(\mathbf{s}_t, \mathbf{a}_t, r_t, \mathbf{c}_t \mid \mathbf{s}_{t-1}, \mathbf{a}_{t-1}, \mathbf{c}_{t-1}, \boldsymbol{\omega}_{<t-1}\right) = P\left(\mathbf{s}_t, \mathbf{a}_t, r_t, \mathbf{c}_t \mid \mathbf{s}_{t-1}, \mathbf{a}_{t-1}, \mathbf{c}_{t-1}\right),$$

*where* $\boldsymbol{\omega}_{<t-1} = \{\mathbf{s}_{t-2}, \dots, \mathbf{s}_1, \mathbf{a}_{t-2}, \dots, \mathbf{a}_1, \mathbf{c}_{t-2}, \dots, \mathbf{c}_1\}$.

This is naturally satisfied under our setting described in Section 3.1.

**Assumption 2** (Distributional Variability). *There exist observed state and action variables $\mathbf{x}_t$ such that for any $\mathbf{x}_t \in \mathcal{X}_t$, there exists a corresponding $\mathbf{x}_{t-1} \in \mathcal{X}_{t-1}$ and a neighborhood $\mathcal{N}^r$ around $(\mathbf{x}_t, \mathbf{x}_{t-1})$ satisfying that, for all $\mathbf{x}_{t-2} \in \mathcal{X}_{t-2}$, $\mathbf{x}_{t-1} \in \mathcal{X}_{t-1}$, $\mathbf{x}_t \in \mathcal{X}_t$, and $\mathbf{x}_{t+1} \in \mathcal{X}_{t+1}$, the following conditional distribution operators are injective: (i) $L_{\mathbf{x}_{t-2}|\mathbf{x}_{t+1}}$, (ii) $L_{\mathbf{x}_{t+1}|\mathbf{x}_t, \mathbf{c}_t}$, and (iii) $L_{\mathbf{x}_t|\mathbf{x}_{t-2}, \mathbf{x}_{t-1}}$, where the conditional operator $L$ represents transformations at the distribution level, that is, how one probability distribution is pushed forward to another (Dunford & Schwartz, 1971).*

> **Assumption justification.** Conceptually, the injectivity of these operator $L$ implies that different inputs induce different output distributions, thus imposing a minimal condition on distributional variability. In RL systems, this condition is naturally satisfied in most stochastic environments where transitions produce sufficient diversity across different states and actions. The assumption also aligns with the conditions in identifiability theory, particularly in works using spectral decomposition and latent variable models (Hu & Schennach, 2008; Hu & Shum, 2012; Fu et al., 2025). We further verify this empirically using MuJoCo RL trajectories with the context instantiated as time-varying wind (App. B.5.1).

**Assumption 3** (Uniqueness of Spectral Decomposition). *For any $\mathbf{x}_t \in \mathcal{X}_t$ and any $\bar{\mathbf{c}}_t \neq \tilde{\mathbf{c}}_t \in \mathcal{C}_t$, there exists a $\mathbf{x}_{t-1} \in \mathcal{X}_{t-1}$ and corresponding neighborhood $\mathcal{N}^r$ satisfying Assumption 2 such that, for some $(\bar{\mathbf{x}}_t, \bar{\mathbf{x}}_{t-1}) \in \mathcal{N}^r$ with $\bar{\mathbf{x}}_t \neq \mathbf{x}_t$, $\bar{\mathbf{x}}_{t-1} \neq \mathbf{x}_{t-1}$:*

  i. $0 < k(\mathbf{x}_t, \bar{\mathbf{x}}_t, \mathbf{x}_{t-1}, \bar{\mathbf{x}}_{t-1}, \mathbf{c}_t) < C < \infty$ *for any $\mathbf{c}_t \in \mathcal{C}_t$ and some constant $C$;*

  ii. $k(\mathbf{x}_t, \bar{\mathbf{x}}_t, \mathbf{x}_{t-1}, \bar{\mathbf{x}}_{t-1}, \bar{\mathbf{c}}_t) \neq k(\mathbf{x}_t, \bar{\mathbf{x}}_t, \mathbf{x}_{t-1}, \bar{\mathbf{x}}_{t-1}, \tilde{\mathbf{c}}_t)$, *where*

$$k(\mathbf{x}_t, \bar{\mathbf{x}}_t, \mathbf{x}_{t-1}, \bar{\mathbf{x}}_{t-1}, \mathbf{c}_t) = \frac{p_{\mathbf{x}_t|\mathbf{x}_{t-1}, \mathbf{c}_t}(\mathbf{x}_t \mid \mathbf{x}_{t-1}, \mathbf{c}_t) p_{\mathbf{x}_t|\mathbf{x}_{t-1}, \mathbf{c}_t}(\bar{\mathbf{x}}_t \mid \bar{\mathbf{x}}_{t-1}, \mathbf{c}_t)}{p_{\mathbf{x}_t|\mathbf{x}_{t-1}, \mathbf{c}_t}(\bar{\mathbf{x}}_t \mid \mathbf{x}_{t-1}, \mathbf{c}_t) p_{\mathbf{x}_t|\mathbf{x}_{t-1}, \mathbf{c}_t}(\mathbf{x}_t \mid \bar{\mathbf{x}}_{t-1}, \mathbf{c}_t)}. \quad (1)$$

> **Assumption justification.** Conceptually, Assumption 3 requires that k, which captures second-order variations in transition dynamics at time $t-1$ and $t$ under the latent variable $\mathbf{c}$, yields distinct values for different $\mathbf{c}$'s. This requirement is typically met in RL, as varied latent dynamics or rewards often cause significant, observable shifts in behavior. *Crucially, this variability is precisely what motivates the need for the identification of the latent variable $\mathbf{c}_t$, as it governs meaningful differences in learning underlying decision-making process.* We further verify this empirically using MuJoCo RL trajectories with the context instantiated as time-varying wind (App. B.5.2).

These assumptions are mild and natural. While Assumption 1 is standard in RL, it can be relaxed without violating our theory (App. B.3.4). Assumptions 2–3 are naturally satisfied in practice, as they simply formalize that latent variables influence the dynamic, motivating why we need the identification of them. Further validation and discussion are provided in App. A.4. Importantly, the more strongly the context influences the dynamics (and thus the more critical it becomes to account for $c$ in decision-making), the more strongly these two assumptions are satisfied: the transition operator becomes more injective as required in Assumption 2, and the spectral ratio $k$ becomes more separable across contexts as required in Assumption 3 (See empirical validation in App. B.5.3). Under these assumptions, we establish an identifiability theory that characterizes the conditions under which the latent factors can be recovered, and specifies the level of identifiability that can be achieved.

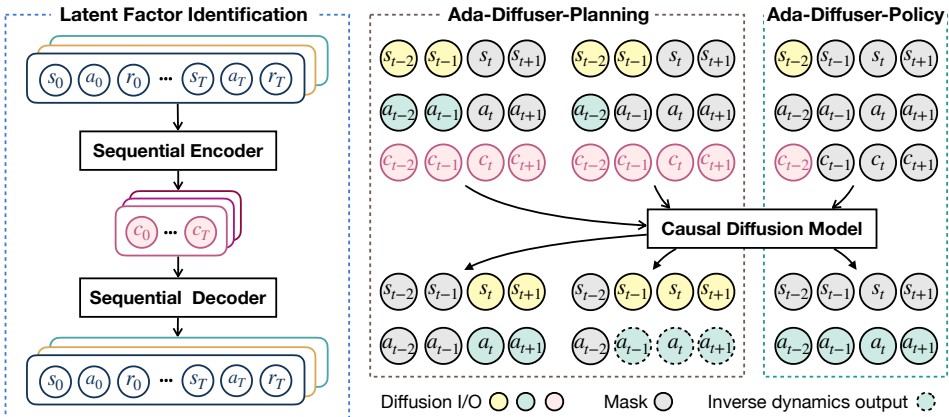

Figure 2: Overview of the `Ada-Diffuser` framework. The modular design consists of two main stages: latent context identification (Stage 1, Section 4.2), followed by a causal diffusion model (Stage 2, Section 4.3) that models the generative structure of the trajectories. The learned model is then used for planning or policy learning conditioned on the inferred latent context.

**Theorem 1** (Identifiability on Latent Factors). *Under Assumptions 1-3, the posterior distribution of latent factor with consecutive observations $p(\mathbf{c}_t \mid \mathbf{x}_{t-2:t+1})$ can be identifiable up to an invertible transformation on the latents $\hat{\mathbf{c}}_t = h(\mathbf{c}_t)$, where $\hat{\mathbf{c}}_t$ is estimated latents and $h$ is an invertible function.*

The proof is in App. B.2. Theorem 1 indicates that **a short temporal window of observations (with future frame at $t+1$)** contains *sufficient* information to *recover the posterior distribution over the true latent factors* (up to an invertible transformation) in an online manner, without requiring access to the full trajectory. This form of identifiability is standard in representation learning and is sufficient for downstream tasks such as dynamics modeling, planning, and control. Any policy or dynamics model that conditions on $\hat{c}_t$ can implicitly compose with $h^{-1}$ without loss of expressiveness. We further discuss the implications of this finding in greater detail in App. A.4.

## 4 LATENT-AWARE ADAPTIVE DIFFUSION PLANNER AND POLICY

Building on Theorem 1, we introduce the `Ada-Diffuser` framework for learning and planning with latent identification. As illustrated in Fig. 2, `Ada-Diffuser` models the trajectory generation process via two modules: (1) **latent factor identification block**, which estimates the sequence of latent variables from the observable trajectories; and (2) **causal diffusion model**, which learns the causal generative process of RL trajectories and explicitly infers latent context. Guided by the theoretical findings in Theorem 1 and the generative process (Sec C), we use autoregressive denoising for temporal dependencies and a backward-refinement step over a minimal–sufficient block, designed via a tailored noise schedule and zig–zag sampling, to recover the latent posterior in an online manner.

In this section, we first present a general formulation of conditional diffusion modeling with latent variables. We then describe the two modules of `Ada-Diffuser` in detail (Fig. 2). The complete algorithmic pseudocode of the training and inference procedures are given in App. D.1.

### 4.1 LATENT-AUGMENTED DIFFUSION MODEL FOR PLANNING AND POLICY LEARNING

Without loss of generality, we denote the observable trajectory as $\boldsymbol{\tau}_x$, which may correspond to a state-action sequence $\boldsymbol{\tau}_{sa}$ or a state-only sequence $\boldsymbol{\tau}_s$, depending on the task setting. To incorporate latent structure, we augment the observable trajectory with the estimated latent context, yielding the full trajectory representation $\boldsymbol{\tau} = [\boldsymbol{\tau}_x, \boldsymbol{\tau}_c]$, where $\boldsymbol{\tau}_c$ denotes the inferred sequence of latent variables.

We train a conditional diffusion model to generate trajectories conditioned on desired attributes $\boldsymbol{y}(\boldsymbol{\tau})$ (e.g., reward or goal specification) and the identified $\mathbf{c}$. The denoising model $\epsilon_\theta$ is trained to predict the noise added during the forward diffusion process via the objective: $\mathcal{L}_{\text{diff}} = \mathbb{E}_{\boldsymbol{\tau}^0, \boldsymbol{y}, t, \boldsymbol{\epsilon}} \left[ \left\| \epsilon_\theta(\boldsymbol{\tau}^t, t, \boldsymbol{y}(\boldsymbol{\tau}), \mathbf{c}) - \boldsymbol{\epsilon} \right\|^2 \right]$, where $\boldsymbol{\tau}^0$ is a clean trajectory sample, $\boldsymbol{\epsilon} \sim \mathcal{N}(0, \mathbf{I})$, and

the noisy trajectory at diffusion step $t$ is constructed as: $\boldsymbol{\tau}^t = \sqrt{\bar{\alpha}_t}\boldsymbol{\tau}^0 + \sqrt{1 - \bar{\alpha}_t}\boldsymbol{\epsilon}$, where $\bar{\alpha}_t$ denotes the cumulative product of the forward noise schedule. Here, the superscript $t$ indexes diffusion steps, and should not be confused with the environment time step indices within the trajectory.

`Ada-Diffuser` can flexibly adapt to generate different components of the trajectory depending on the task. In the ***planning*** setting, the model generates full trajectories $\boldsymbol{\tau} = \{\mathbf{x}_t, \mathbf{x}_{t+1}, \ldots, \mathbf{x}_{t+T_p}\}$, where $T_p$ denotes the planning horizon. Here, $\mathbf{x}_t$ may have two cases: (i) $\mathbf{x}_t = \{\mathbf{s}_t, \mathbf{a}_t\}$, when both states and actions are generated, (ii) $\mathbf{x}_t = \{\mathbf{s}_t\}$, when only states are generated. In the latter case, we train an inverse dynamics model (IDM) (Ajay et al., 2023a) to infer the corresponding actions from state transitions. In the ***policy learning*** setting, the model generates only actions, i.e., $\boldsymbol{\tau} = \{\mathbf{a}_{t+1}, \mathbf{a}_{t+2}, \ldots, \mathbf{a}_{t+T_a}\}$, where $T_a$ is the action generation horizon. While multi-step action generation methods (e.g., DP (Chi et al., 2023)) can also be viewed as a form of planning (Zhu et al., 2023), for generality, we categorize such settings under the policy framework. `Ada-Diffuser-Policy` accommodates both variants: multi-step action generation ($T_a > 1$), as in DP, and single-step decision-making ($T_a = 1$), as in IDQL (Hansen-Estruch et al., 2023).

## 4.2 STAGE 1: OFFLINE LATENT FACTOR IDENTIFICATION

Based on Theorem 1, we structure the latent inference process around *temporal blocks*, using short segments of trajectories to identify the latent context at each time step. We adopt a variational inference framework (Kingma & Welling, 2014) in which the latent variable $\mathbf{c}_t$ is inferred block-wise. That is, the prior distribution is conditioned on the latent variable from the previous step and the in-block history, while the posterior additionally incorporates future observations. Specifically, given a trajectory block $t - T_x : t+1$, where $T_x$ is the block size, we have prior $p_\phi(\mathbf{c}_t \mid \mathbf{c}_{t-1})$, and posterior $q_\psi(\mathbf{c}_t \mid \mathbf{x}_{t-T_x:t+1})$, where $\mathbf{x}$ denotes the observed variables and may correspond to $\{\mathbf{s}\}$, $\{\mathbf{s}, \mathbf{a}\}$, or $\{\mathbf{s}, \mathbf{a}, r\}$. We then optimize the evidence lower bound (ELBO) of the observed trajectories:

$$\mathcal{L}_{\text{ELBO},t} = \mathbb{E}_{q_\psi(\mathbf{c}_t \mid \mathbf{x}_{t-T_x:t+1})} \left[ -\log p_\theta(\mathbf{x}_t \mid \mathbf{x}_{t-1}, \mathbf{c}_t) \right] + D_{\text{KL}} \left( q_\psi(\mathbf{c}_t \mid \mathbf{x}_{t-T_x:t+1}) \,\|\, p_\phi(\mathbf{c}_t \mid \mathbf{c}_{t-1}) \right).$$

Here, the reconstruction term, $-\log p_\theta(\mathbf{x}_t \mid \mathbf{x}_{t-1}, \mathbf{c}_t)$ is instantiated based on the available observation modalities. Specifically, (i) when only states are observed, the model reconstructs $\mathbf{s}_t$ conditioned on $(\mathbf{s}_{t-1}, \mathbf{c}_t)$; and (ii) when rewards are available, the model also reconstructs $r_t$ from $(\mathbf{s}_t, \mathbf{a}_t, \mathbf{c}_t)$. The stage is learned through a sequential encoder and decoder (l.h.s., Fig. 2).

## 4.3 STAGE 2: CAUSAL DIFFUSION MODEL

We propose a **causal diffusion model** for learning the generative process described in Sec. 3.1. By "causal," we refer to the modeling of the true underlying data generation process, which incorporates two key desiderata: (1) the *autoregressive process* inherent in temporal sequential RL trajectories; and (2) the *latent factor process*, capturing the causal influence of the unobserved context variables $\mathbf{c}_t$ on the observations (e.g., $\mathbf{x}_t = [\mathbf{s}_t, \mathbf{a}_t, r_t]$). Thus, unlike prior diffusion-based RL methods and latent-augmented variants (Sec. 2; see Table A11 for a comparison), our approach incorporates the following design choices.

**Autoregressive Denoising**  To model the autoregressive structure of trajectory generation, and following the recent advances in autoregressive diffusion (Chen et al., 2024; Xie et al., 2024b; Wu et al., 2023), we introduce a **causal denoising schedule**. Under this mechanism, each time step within a local temporal block is assigned a denoising schedule that depends both on its temporal distance from the conditioning anchor and on the inferred latent variables. This reflects the intuition that later time steps exhibit higher uncertainty. Specifically, for a trajectory of length $T$, we assign monotonically increasing noise levels $\{k_1, \ldots, k_T\}$, sampled linearly as $k_i = \frac{i}{T}K$ where $i \in \{1, \ldots, T\}$ and $K$ denotes the maximum diffusion step.

Given the inferred latent context $\hat{\mathbf{c}}_{0:T}$, the model performs autoregressive denoising over the block in $T$ steps. The overall denoising process is defined as:

$$p_\theta \left( \mathbf{x}_0^0, \ldots, \mathbf{x}_{T-1}^0 \mid \mathbf{x}_0^{k_1}, \ldots, \mathbf{x}_{T-1}^{k_T}, \hat{\mathbf{c}}_{0:T} \right), \tag{2}$$

where $\mathbf{x}_i^{k_i}$ denotes the noisy observation at time step $i$, and $\mathbf{x}_i^0$ is the clean, denoised output.

Specifically, the first denoising step is: $p_\theta(\mathbf{x}_0^0, \mathbf{x}_1^{k_1}, \ldots, \mathbf{x}_{T-1}^{k_{T-1}} \mid \mathbf{x}_0^{k_1}, \ldots, \mathbf{x}_{T-1}^{k_T}, \hat{\mathbf{c}}_{0:T})$, where the first observation $\mathbf{x}_0$ has been fully denoised and other observations are partially denoised, followed by

the second step: $p_\theta(\mathbf{x}_1^0, \mathbf{x}_2^{k_1}, \ldots, \mathbf{x}_{T-1}^{k_{T-2}} \mid \mathbf{x}_0^0, \mathbf{x}_1^{k_1}, \ldots, \mathbf{x}_{T-1}^{k_{T-1}}, \hat{\mathbf{c}}_{0:T})$, and finally until all observations are denoised: $p_\theta(\mathbf{x}_{T-1}^0 \mid \mathbf{x}_0^0, \ldots, \mathbf{x}_{T-2}^0, \mathbf{x}_{T-1}^{k_1}, \hat{\mathbf{c}}_{0:T})$.

**Denoise-and-refine Mechanism** Theorem 1 indicates that both historical and future observations are required for recovering the latents. However, these future observations are not accessible during online inference, which results in a mismatch between identifiability requirements and available information. Hence, guided by this insight with preserving the causal structure of the generative process, we propose a novel *denoise-and-refine mechanism* that alternates between denoising the observable sequences and refining the latent estimates, and is applied consistently during both training and inference to ensure high-quality latent context recovery in an online manner. We introduce how we implement this during training and inference.

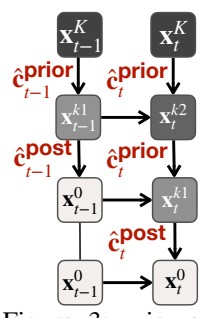

*Training*: Given a noisy input $\mathbf{x}_t^{k_t}$ with noise level $k_t$, we first sample an initial latent context from the prior: $\hat{\mathbf{c}}_t^{\text{prior}} \sim p_\phi(\mathbf{c}_t \mid \mathbf{c}_{t-1})$, and use it to denoise the observation: $\hat{\mathbf{x}}_t^{(0)} = \epsilon_\theta(\mathbf{x}_t^{k_t}, k_t, \hat{\mathbf{c}}_t^{\text{prior}})$. Then we infer the latent using the posterior network, conditioned on a broader temporal window including future observations (accessible in offline data): $\hat{\mathbf{c}}_t^{\text{post}} \sim q_\psi(\mathbf{c}_t \mid \mathbf{x}_{t-k:t+1})$, and obtain a **refined** denoised prediction: $\hat{\mathbf{x}}_t^{(0)\prime} = \epsilon_\theta(\mathbf{x}_t^{k_t}, k_t, \hat{\mathbf{c}}_t^{\text{post}})$.

We have two reconstruction losses: one from the prior-sampled latent, $\mathcal{L}_{\text{prior}} = \|\hat{\mathbf{x}}_t^{(0)} - \mathbf{x}_t^0\|^2$, and one from the posterior-sampled latent, $\mathcal{L}_{\text{post}} = \|\hat{\mathbf{x}}_t^{(0)\prime} - \mathbf{x}_t^0\|^2$. To encourage the posterior latent to produce better reconstructions, we introduce a **contrastive improvement loss**: $\mathcal{L}_{\text{rel}} = \text{softplus}\left(\log \mathcal{L}_{\text{post}} - \log \text{sg}\left(\mathcal{L}_{\text{prior}}\right) + m\right)$, where $\text{sg}(\cdot)$ denotes stop-gradient, $\text{softplus}(u) = \log(1 + e^u)$, and $m \geq 0$ is a margin hyperparameter. The final

Figure 3: zig-zag sampling (2 steps).

objective for this denoise-and-refine step is: $\mathcal{L}_{\text{d-r}} = \mathcal{L}_{\text{post}} + \lambda_{\text{prior}}\mathcal{L}_{\text{prior}} + \lambda_{\text{rel}}\mathcal{L}_{\text{rel}}$, where $\lambda_{\text{prior}}$ and $\lambda_{\text{rel}}$ are weighting coefficients. $\mathcal{L}_{\text{diff}}$ updates only $\theta$, $\mathcal{L}_{\text{post}}$ updates only $\psi$, $\mathcal{L}_{\text{prior}}$ updates only $\phi$, and $\mathcal{L}_{\text{rel}}$ updates both $\phi$ and $\psi$.

*Inference*: During inference, future observations are not available, which prevents direct use of the posterior network for latent inference. To address this, we adopt a *zig-zag sampling strategy*[2] that combines autoregressive denoising with latent refinement. Specifically, we first sample the entire trajectory by applying the forward diffusion process with the maximum noise level $K$. We then perform autoregressive denoising across time.

For each time step $t$, we begin by denoising $\mathbf{x}_t^K$ to an intermediate noise level $k_1$ using $\hat{\mathbf{c}}_t$ sampled from the prior: $\hat{\mathbf{c}}_t^{\text{prior}} \sim p_\phi(\mathbf{c}_t \mid \mathbf{c}_{t-1})$. We then obtain updated $\hat{\mathbf{c}}_t$ from the posterior latent distribution $\hat{\mathbf{c}}_t^{\text{post}} \sim q_\psi(\mathbf{c}_t \mid \mathbf{x}_{t-k:t-1}^0, \mathbf{x}_t^{k_1}, \mathbf{x}_{t+1}^{k_2})$, which is conditioned on the denoised history, the intermediate step with noise level $k_1$, and the next step with noise level $k_2$. We then use $\hat{\mathbf{c}}_t^{\text{post}}$ as the input to further denoise $\mathbf{x}_t^{k_1}$ to $\mathbf{x}_t^0$. An illustration of the zig-zag inference process is provided in Fig. 3[3].

In summary, `Ada-Diffuser` leverages autoregressive noise scheduling to reflect temporal structure, integrates latent context identification by the *denoise-and-refine* mechanism, and employs *zig-zag sampling* for online latent inference. This framework accommodates a wide range of scenarios, including latent dynamics/rewards, learning from action-free data with latent actions, and both state- and image-based environments. All variants share the same core, with task-specific modifications to the input/output only. Details of these architectural and variations are in App. H.

## 5 EXPERIMENTS

We aim to answer the following questions in the evaluation: (1) *Latent Identification*: How well can `Ada-Diffuser` capture latent factors in the environment? (2) *Learning with Latent Factors*: How effective is `Ada-Diffuser` in planning and control when learning with the latent context on dynamics and reward? And can `Ada-Diffuser` infer latent actions from action-free demonstrations? (3) *Learning with Environments w/o Explicit Latents*: In environments without explicit latent factors,

---

[2]Note on terminology: our use of "zig–zag" is purely descriptive, and there is no connection between the proposed sampling and Bai et al. (2024).

[3]A larger illustration with 4 steps are given in App. Fig. A2.

Figure 4: (a). Identification Results (i.e., Linear Probing MSE, $R^2$) and normalized rewards on the Cheetah environment with time-varying wind as the latent factor, evaluated across different block sizes. (b). Results (i.e., average success rate) on planning with action-free demonstrations on Robomimic benchmark. "AF" denotes Action-free.

can modeling latent processes still bring performance gains? (4) *Ablation Studies*: What is the impact of key design choices in the framework?

## 5.1 SETTINGS

**Benchmarks** We consider a diverse set of benchmarks, including Mujoco-based locomotion tasks (`Cheetah`, `Ant`, `Walker`), a robot navigation task (`Maze2D`), and a robot arm control task (`Franka-Kitchen`) (Gupta et al., 2020), all from the D4RL benchmark suite (Fu et al., 2020). We also consider robotic manipulation tasks from `RobotMimic` (Mandlekar et al., 2021) and `LIBERO-10` (Liu et al., 2023). A detailed description and illustration of these environments is provided in App. F. We introduce latent factors affecting both dynamics ($\mathbf{c}^s$) and reward functions ($\mathbf{c}^r$) in the Cheetah and Ant environments, considering two types of variations: episodic changes (E) and fine-grained, time-varying step-wise changes (S). The specific change functions for each setting are detailed in App. F.1. For evaluating latent action modeling, we follow the setup from LDP (Xie et al., 2025), using action-free, pixel-based demonstrations from the LIBERO benchmark (Liu et al., 2023). We use our framework to learn the inverse dynamics model to infer the latent actions (details are in App. G.1). In total, we evaluate on 8 environments with 23 settings.

**Baselines** We compare `Ada-Diffuser` with a diverse set of baselines for fair and comprehensive evaluation. *(1) Vanilla diffusion models*: For planning, we consider Diffuser (Janner et al., 2022) and DD (Ajay et al., 2022). For policy learning, we include DP and IDQL (Hansen-Estruch et al., 2023). We also evaluate LDCQ (Venkatraman et al., 2024), which learns a latent skill space and optimizes a value function conditioned on both states and latent skills. *(2) Latent context modeling*: We include MetaDiffuser (Ni et al., 2023) that learns contextual representations from multiple environments. We also consider using LILAC (Xie et al., 2021) and DynaMITE (Liang et al., 2024a) which models nonstationarity in RL through latent context learning using belief states. For a fair comparison, we integrate their context modules into diffusion planners and policies as plug-in components (detailed analysis in App. H.1). *(3) Latent action modeling*: We compare with LDP (Xie et al., 2025) with action-free demonstrations for planning. In total, we compare with 9 baselines across these settings.

**Architecture Choices** (Details are in App. D.2) For latent factor identification, we use GRU (Cho et al., 2014) embedding with MLP layers as both prior and posterior encoders to produce Gaussian distribution over latents. For decoders, we use MLP layers. For planning and policy learning, we use UNet (Ronneberger et al., 2015) or Transformers (Vaswani et al., 2017) as denoising networks and use MLPs to learn the IDM. We use VAE (Kingma & Welling, 2014) for the visual encoders.

## 5.2 RESULTS AND ANALYSIS

**Results on Latent Identification** To verify our identification theory, we evaluate model performance under different block sizes that contain varying amounts of temporal context. We include settings where all blocks have sufficient observations, as well as a challenging case with insufficient observations (i.e., without access to future observations). To quantify the quality of the learned latent representations, we adopt linear probing and the coefficient of determination $R^2$ as the evaluation metric. The results, together with normalized results, are shown in Fig. 4(a). Similarly, we also provide the clustering result in App. Fig. A6. The yellow region indicates settings with insufficient observations, resulting in lower identification results. The purple region corresponds to sufficient observations and yields relatively strong performance, and the green region reflects larger block sizes,

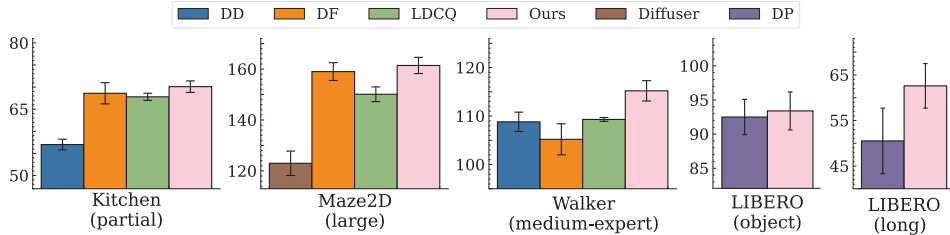

Figure 5: Results on environments without explicitly designed latent factors. Complete results are provided in App. Table A6–A9.

| Environment | Diffuser | DF | DF + DynaMITE | DF + LILAC | MetaDiffuser | Ours |
|---|---|---|---|---|---|---|
| Cheetah-Wind-E ($\mathbf{c}^s$) | -120.4 ± 12.7 | -105.8 ± 9.6 | -82.3 ± 8.2 | -91.5 ± 7.8 | -95.3 ± 7.4 | **-68.9**± 7.6 |
| Cheetah-Wind-S ($\mathbf{c}^s$) | -148.5 ± 9.8 | -102.0 ± 10.2 | -87.2 ± 10.4 | -96.7 ± 9.5 | -105.6 ± 14.5 | **-73.5**± 8.7 |
| Cheetah-Vel-E ($\mathbf{c}^r$) | -102.4 ± 18.2 | -85.6 ± 18.3 | -60.2 ± 10.8 | -67.8 ± 11.0 | -62.6 ± 11.1 | **-45.8**± 9.5 |
| Ant-Dir-E ($\mathbf{c}^r$) | 188.6 ± 39.2 | 195.4 ± 47.0 | 266.7 ± 28.1 | 233.6 ± 31.9 | 229.4 ± 32.6 | **285.3**± 24.5 |

Table 1: Results (5 seeds) on `Ada-Diffuser-Planner` with latent factors that affects dynamics and rewards. $\mathbf{c}^s$ and $\mathbf{c}^r$ indicate the changes on dynamics and reward, $E$ and $S$ represent the episodic and time-step changes. All results are averaged over 5 random seeds.

which lead to degraded results due to redundant information or inherent difficulty for optimization. Notably, the reward is positively associated with the accuracy of latent identification, validating the importance of identifying latent factors in RL trajectories.

**Results on Decision-making** We consider three groups based on the kind of latent factors.

**> *Group I: Latent factors on dynamics and reward*:** Table 1 presents the results of learning under latent factors that affect dynamics and rewards in locomotion tasks. To ensure a fair comparison, we implement autoregressive variants of DynaMITE and LILAC using the DF backbone. Results for the DP-backbone counterparts are provided in App. Tables A4, which are consistently worse than DF. Additional results, including using DP as backbones (`Ada-Diffuser`-policy), oracle variants and meta-learned versions of `Ada-Diffuser` that use ground-truth latents as input, are provided in App. Table A4–A5. From the results, we observe that `Ada-Diffuser` consistently achieves the best performance, with a significant margin over all baselines. In particular, it outperforms Diffusion planners and policies even when those models are enhanced with latent context modules such as DynaMITE and LILAC (pink area), which are most comparable to our setting. Furthermore, `Ada-Diffuser` outperforms DF, showing the effectiveness of our framework.

**> *Group II: Latent Actions*:** Following Xie et al. (2025), we consider learning from action-free demonstration data, where actions are treated as latent factors to be inferred. We adopt the same setup as in (Xie et al., 2025), using a pre-trained visual encoder obtained via a VAE to learn the latent space from pixel observations. We then train a latent planner and an IDM using a diffusion-based approach. Unlike prior work, our diffusion-based latent planner additionally incorporates latent factors **c** to model latent context. Importantly, we train only the planner using additional action-free demonstrations. Detailed training procedures are provided in App. G.1. Results on several tasks in `Robomimic` benchmark show that we can bring improvements on all tasks via modeling the latent process supplementary to the latent planner in (Xie et al., 2025). Here, the IDM is trained solely on expert demonstrations. Complete results are provided in App. Table A3.

**> *Group III: Environments w/o Explicitly Designed Latents*:** Crucially, in this scenario, the latent variable **c** effectively serves as a form of Bayesian filtering over the observed trajectories, capturing the inherent stochasticity in the data (a more detailed discussion in App. D.3). Such variability commonly arises from system noise, expert action noise, or high-level unobserved factors. The results, shown in Fig. 5 (full results provided in App. Table A6–A9), support this interpretation. Even in environments without explicitly designed latent contexts, incorporating latent modeling allows `Ada-Diffuser` to achieve performance that is comparable to or better than these baselines. By recovering the latent variables that capture stochasticity, nonstationarity, or unobserved structure in

| Latent Design | Orig. | w/o latents | Freeze | 0.5× | 2× | 4× | 6× |
|---|---|---|---|---|---|---|---|
| Cheetah ($\mathbf{c}^s$) | **-73.5** | -103.5 | -110.4 | -85.2 | -77.6 | -89.5 | -102.4 |
| LIBERO | **93.4** | 89.3 | 90.2 | 90.9 | 89.4 | 87.6 | 85.0 |

| Diffusion Design | Orig. | w/o refine | w/o zigzag | same NS | random NS |
|---|---|---|---|---|---|
| Cheetah ($\mathbf{c}^s$) | **-73.5** | -82.0 | -91.6 | -89.7 | -84.6 |
| LIBERO | **93.4** | 83.9 | 91.4 | 85.2 | 88.5 |

Table 2: Ablations on Cheetah-Wind-S (planner) and LIBERO (DP-policy).

the offline trajectories, the model can produce rollouts that better match the underlying dynamics, even when the demonstrations are imperfect. These findings suggest that our framework can consistently capture implicit latent process in the data, improving both trajectory modeling and planning.

## 5.3 ABLATION STUDIES

We conduct ablation studies to evaluate the contributions of key components in our framework. For *latent factor identification*, Fig. 4(a) shows the effect of different temporal block sizes, illustrating the benefit of incorporating future observations during inference. Here, we also consider ablations where (i) the entire latent identification module is removed, (ii) the latent identification network is frozen after the first $10\%$ of training steps, and (iii) different numbers of latent updates are used. For the *causal diffusion model*, we examine the impact of the following design choices: (i) removing the refinement step (*w/o refine*); (ii) removing zig-zag sampling (*w/o zig-zag*); (iii) replacing the causal noise schedule with a fixed noise level across time steps (vanilla diffusion) or with random noise scaling as in DF (Chen et al., 2024) (*same NS, random NS*). The results in Table 2 demonstrate the effectiveness of these modules in our framework in both settings: with and without explicit latent factors. Specifically, For the **latent identification ablations**, we find that the latent variables play a critical role. In particular, freezing the latent module makes the model perform poorly, because the latent context follows a temporal process and must continue adapting during training. Varying the latent dimensionality within a moderate range (about 0.5×–2×) does not significantly change performance, but using overly large latent dimensions (e.g., 4×–6×) degrades results, likely due to redundant capacity and harder optimization.

In terms of causal diffusion, for refinement and zig–zag, we hypothesize the gains come from reducing posterior mismatch. We therefore run a latent probing test on Cheetah with changing wind and report linear-probe MSE across variants; Ada-Diffuser with both refinement and zig–zag attains the lowest error (Table A16; Details are in App. I.2.4). Removing backward refinement yields the largest degradation ($0.18 \rightarrow 0.28$), consistent with the role of refinement in letting future evidence within a block update the latent posterior and reduce temporal lag. Disabling zig–zag also harms accuracy ($0.18 \rightarrow 0.23$), suggesting that alternating conditioning helps align the denoising trajectory with the latent dynamics rather than purely following the forward temporal pass. Moreover, the gap between our full model (0.18) and the oracle that has access to true futures (0.12) is small, verifying that the predicted future is already sufficiently informative for reliable latent inference in practice.

Additional ablations are provided in App. I.2, including full results, comparisons of alternative noise schedules beyond linear (App. I.2.2), sweeps over temporal block length (App. I.2.3), and analyses of long-horizon planning (App. I.2.5). Notably, we show that our method introduces no significant computational overhead in terms of training runtime and inference latency (App. I.1, Table A12-A13).

## 6 CONCLUSIONS

We demonstrate that identifying latent factors from sequential observations is critical for effective decision-making. We provide theoretical results that establish conditions under which latent variables can be identified using small temporal blocks of observations. This insight enables a principled integration of latent identification into a diffusion-based generative framework, allowing us to capture the underlying causal process while maintaining scalability. Ada-Diffuser is broadly applicable to a variety of settings, including planning and control tasks with or without explicit latent structure, and even action-free demonstrations. Results across diverse benchmarks show substantial improvements, validating the effectiveness of our method not only in environments with designed latent factors but also in general settings where latent structure is implicit but influential.

ACKNOWLEDGEMENT

We would like to acknowledge the support from NSF Award No. 2229881, AI Institute for Societal Decision Making (AI-SDM), the National Institutes of Health (NIH) under Contract R01HL159805, and grants from Quris AI, Florin Court Capital, MBZUAI-WIS Joint Program, and the Al Deira Causal Education project.

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

# Appendix of Latent-Aware Adaptive Diffusion for Decision-Making

## A    Discussions and Overview

In this section, we expand on the design and motivation behind `Ada-Diffuser`, including the rationale for modeling latent factors in decision-making, key architectural choices, and additional analysis of the experimental results presented in Section 5. We then provide an overview of the remaining contents of this appendix.

### A.1    Broader Impact

Our work aims to identify and leverage latent processes in generative decision-making, with applications in real-world domains such as robotics and healthcare. While these tasks may entail potential

societal risks, we do not believe any specific concerns need to be highlighted here. Instead, by uncovering and modeling the underlying hidden processes, our approach promotes greater transparency in decision-making, which can ultimately lead to more reliable and trustworthy outcomes.

## A.2   LIMITATIONS AND FUTURE WORK

One current limitation is that this work focuses primarily on theoretical formulation and algorithmic development. Although we evaluate on a variety of established benchmarks, real-world deployment, such as in self-driving, aerial drones, and physical robotics, remains an important direction for future work.

## A.3   DISCUSSIONS ON THE CORE IDEA

**Q1: On Latent Modeling.** *Why is it necessary to model latent processes when we already have access to a large amount of demonstration data?*

In many decision-making systems, there exist unobservable variables that influence both the dynamics and the reward structure. More generally, these latent variables often evolve over time. Such scenarios are common in real-world settings, for example, in robotic control, system dynamics can be affected by external forces (e.g., wind, friction), or by varying user demands (e.g., different target positions). In these cases, learning an optimal policy requires conditioning on the latent factors, especially when they are non-stationary or when transferring to new domains. Prior work has demonstrated the importance of latent variable modeling in both reinforcement learning (RL) and imitation learning (IL) (Zintgraf et al., 2021; Liang et al., 2024a; Nguyen et al., 2021; Rakelly et al., 2019; Ni et al., 2023; Xie et al., 2021).

Even with access to large demonstration datasets, it remains difficult to ensure sufficient coverage over the full space of environmental or task-specific latent factors relevant to decision-making. This limitation has been widely acknowledged in recent efforts focused on analyzing data quality and designing data collection protocols to promote generalization (Belkhale et al., 2023; Xie et al., 2024a; Hejna et al., 2024; Gao et al., 2024a). However, most of these works target fixed or task-specific latent variables. In contrast, we consider a more general setting where latent factors evolve over time and are not predefined. Our framework provides theoretical guarantees for identifying such latent variables from partial observations and seamlessly integrates this identification process into diffusion models, enabling scalability across complex decision-making tasks.

**Q2: On the Scenarios w/o Explicit Latents.** *What does latent modeling represent when no explicit latent factors are defined, and why can it still benefit decision-making?*

First, **Latent stochasticity is always present** (in real-world systems). Even in settings where all task-relevant observations are available, e.g., in locomotion tasks where full physical state information is provided, or in robotic manipulation with access to both proprioceptive and visual inputs, there may still exist underlying processes that are not directly observed. These include domain-specific factors such as external forces (e.g., wind) or dynamically changing task goals (e.g., target positions), which can be viewed as implicit latent variables. Hence, it is crucial to infer and condition on these latent factors

In the extreme case where such factors are also *fully observed*, latent modeling can still offer significant benefits. Specifically, it can capture residual stochasticity present in the environment or demonstration data, serving to explain variability not accounted for by observable features. As shown in our formulation: $s_t = f(s_{t-1}, a_{t-1}, \epsilon_t), \quad r_t = g(s_t, a_t, \delta_t)$, the residual stochasticity $(\epsilon, \delta)$ can be interpreted as implicit latent variables (sometimes can be time-correlated) influencing transitions and rewards. The model can then identify meaningful structure from irrelevant or noisy variations, for instance, filtering out visual background artifacts that are not predictive of dynamics or optimal actions. In this sense, the learning framework is conceptually similar to Bayesian Filtering

Moreover, **partial observability and attribution gaps exist even in clean data**. Even in environments with consistent near-deterministic demonstrations, the agent often lacks access to the full set of latent causal factors or attributes that influence behavior. Specifically, many systems exhibit

structured yet unobserved variability (e.g., task goals, preferences, intentions), and modeling this variability with latent variables improves generalization.

**Q3: On the Identification Theory.** *What does the identification theory establish, and how does it inform algorithm design?*

The identification theory (Theorem 1) establishes that the distribution over latent variables can be provably recovered from observable trajectories using only a small temporal window, specifically, a small temporal block of four time steps. This provides a general non-parametric theoretical guarantee that latent factors can be identified without requiring strong inductive biases or restrictive assumptions on the model class or functional form.

This "four-step" result has direct implications for algorithm design. It suggests that latent identification can be effectively performed using a short temporal block, which aligns naturally with block-wise generative modeling approaches such as diffusion models. These models operate over segments or chunks of data, and our theoretical results justify using local temporal blocks to infer latent variables in a principled and scalable manner.

## A.4 Discussions on the Theoretical Assumptions and Results

**Q4: On the Assumptions.** *What do Assumption 2 (Distributional Variability) and Assumption 3 (Uniqueness of Spectral Decomposition) mean, and why are they considered mild?*

We expand on the intuition and practical relevance of these two assumptions below.

*Distributional Variability* (Assumption 2) refers to the requirement that the conditional distributions

$$p(\mathbf{x}_{t-2} \mid \mathbf{x}_{t+1}), \quad p(\mathbf{x}_{t+1} \mid \mathbf{x}_t, \mathbf{c}_t), \quad \text{and} \quad p(\mathbf{x}_t \mid \mathbf{x}_{t-2}, \mathbf{x}_{t-1})$$

are sufficiently sensitive to variations in their input. That is, for different input pairs within a local neighborhood, the output distributions differ meaningfully, ensuring the system exhibits enough variability for identification. This assumption aligns with real-world decision-making settings (e.g., locomotion or robotic manipulation), where changes in inputs such as physical state, control policy, or reward function lead to observable changes in output distributions.

*Uniqueness of Spectral Decomposition* (Assumption 3) builds on this by ensuring that changes in the latent variable $\mathbf{c}_t$ induce distinct influences on the transition dynamics, specifically on the mapping from $\mathbf{x}_{t-1}$ to $\mathbf{x}_t$. To formalize this, we consider the operator $k$:

$$k(\mathbf{x}_t, \bar{\mathbf{x}}_t, \mathbf{x}_{t-1}, \bar{\mathbf{x}}_{t-1}, \mathbf{c}_t) = \frac{p(\mathbf{x}_t \mid \mathbf{x}_{t-1}, \mathbf{c}_t) \cdot p(\bar{\mathbf{x}}_t \mid \bar{\mathbf{x}}_{t-1}, \mathbf{c}_t)}{p(\bar{\mathbf{x}}_t \mid \mathbf{x}_{t-1}, \mathbf{c}_t) \cdot p(\mathbf{x}_t \mid \bar{\mathbf{x}}_{t-1}, \mathbf{c}_t)}, \tag{A1}$$

which separates into two multiplicative components:

$$k_1 = \frac{p(\mathbf{x}_t \mid \mathbf{x}_{t-1}, \mathbf{c}_t)}{p(\mathbf{x}_t \mid \bar{\mathbf{x}}_{t-1}, \mathbf{c}_t)}, \tag{A2}$$

$$k_2 = \frac{p(\bar{\mathbf{x}}_t \mid \bar{\mathbf{x}}_{t-1}, \mathbf{c}_t)}{p(\bar{\mathbf{x}}_t \mid \mathbf{x}_{t-1}, \mathbf{c}_t)}. \tag{A3}$$

Here, $k_1$ and $k_2$ measure how changes in historical inputs affect the transition distribution at the current time step. The assumption requires that for any two distinct values of $\mathbf{c}_t$, the corresponding operator $k$ is different, indicating that the latent variable has a sufficiently strong influence on the system dynamics.

Since $\bar{\mathbf{x}}$ is in the neighborhood of $\mathbf{x}$, this formulation effectively captures second-order changes in the transition dynamics with respect to the latent variable $\mathbf{c}_t$. This reflects many real-world RL systems, where even unobservable latent factors (e.g., wind speed or goal target) cause noticeable and structured changes in transition behavior over time, for instance, by considering velocity as states.

In summary, these two assumptions are not only theoretically necessary for identification, but also naturally hold in many RL and control systems. They justify *the need to explicitly model and identify latent variables*, as such variables often induce meaningful and structured changes in both dynamics and optimal decision-making behavior.

**Q5: On the Identification of Posterior Distribution and up to the Invertible Function (Theorem 1).** *Why do we aim to identify the posterior distribution over latent variables, and what is the role of the invertible function $h$ between the estimated and true latents?*

Theorem 1 establishes that the posterior distribution over latent factors given surrounding observations, $p(\mathbf{c}_t \mid \mathbf{x}_{t-2:t+1})$, is identifiable up to an invertible transformation. That is, the estimated latent $\hat{\mathbf{c}}_t$ satisfies $\hat{\mathbf{c}}_t = h(\mathbf{c}_t)$ for some invertible function $h$.

This form of identifiability is sufficient for downstream tasks such as dynamics modeling, planning, and control. Specifically, the learned dynamics or policy can be composed with $h^{-1}$ without loss of expressiveness or utility. Since we only need to condition on the inferred latent $\hat{\mathbf{c}}_t$ to perform these tasks, any invertible transformation of the latent space preserves the representational capacity required for decision-making. In other words, although we may not recover the true latent variable $\mathbf{c}_t$ exactly, the recovered representation $\hat{\mathbf{c}}_t$ contains the same information and can be used equivalently in practice.

Therefore, identifying the posterior distribution (up to an invertible transformation) is both theoretically meaningful and practically sufficient for learning accurate dynamics models and optimal policies.

### A.5  Discussions on the Model Design

**Q6: On Different Settings (Planning and Policy).** *How is* `Ada-Diffuser` *applied to both planning and policy learning settings?*

`Ada-Diffuser` is designed as a unified and generic framework that accommodates different types of inputs $\mathbf{x}$ (e.g., states, state-action pairs) and outputs (e.g., actions, trajectories, or state sequences). This flexibility allows it to support a wide range of planning and policy learning paradigms. We summarize four representative settings below:

- **Planning with state-action generation:** The model generates both states and actions, with latent variables influencing dynamics or rewards. This setting aligns with prior work such as Diffuser (Janner et al., 2022).

- **Planning with state-only generation:** The model generates future states, and an inverse dynamics model is used to recover the corresponding actions. This setup follows Decision Diffuser (Ajay et al., 2023a).

- **Planning from action-free demonstrations:** Only state sequences are available, and latent variables are assumed to capture high-level behaviors or skills. This setting extends latent diffusion planning (Xie et al., 2025).

- **Policy learning:** The model generates actions conditioned on the current or recent history of states. This includes multi-step action generation (as in Diffusion Policy (Chi et al., 2023)) and one-step action generation (as in Implicit Diffusion Q-Learning, IDQL (Hansen et al., 2022)). In both cases, latent factors may affect the underlying dynamics or rewards.

These diverse settings demonstrate the universality of our framework and highlight that uncovering latent structure is a broadly applicable and critical problem in generative decision-making.

**Q7: On the Latent Identification.** *How is Stage 1 (Latent Identification) trained, and does it introduce additional computational overhead?*

In Stage 1, we train the latent identification module using an offline dataset, as commonly done in offline RL and imitation learning tasks. Specifically, we employ a lightweight variational autoencoder (VAE) to optimize the ELBO defined in Section 4.2. Empirically, this stage introduces minimal computational overhead (Appendix I.1). We further provide an ablation study in Appendix I.1 showing the impact of the number of training samples on the effectiveness of the latent identification module.

**Q8: On the Temporal Block Design.** *How does this reflect Theorem 1, and why do we not use exactly four steps in practice?*

Our approach reflects the theoretical result in Theorem 1 by identifying latent variables using small temporal blocks in both Stage 1 and Stage 2. In Stage 1, we segment trajectories into local blocks and optimize the ELBO to learn the posterior over latent variables. In Stage 2, we apply block-wise refinement to improve the posterior estimates using both past and one-step future observations, making a more accurate identification than using the prior alone.

While Theorem 1 shows that four consecutive time steps are sufficient for identifiability in principle, we do not strictly limit the block size to four in practice. Empirically, we find that using slightly larger blocks (typically between 6 and 20 steps) leads to more stable optimization and better performance. Our ablations in Appendix I.2 show that without access to future observations, identifiability degrades, aligning with the theory.

We treat the "four-step" condition not as a strict architectural constraint but as a theoretical justification (sufficient condition) for using small temporal blocks. The optimal number of steps in practice may vary depending on data properties, task complexity, and model capacity.

**Q9: On the Refinement Step.** *Why is the refinement step necessary, how does it work, and does it introduce additional computational overhead?*

The refinement step is motivated by the identification theory, which suggests that incorporating the current and future observations (other than only using historical ones) allows the model to infer a more informative posterior over latent variables than relying on the prior alone. This posterior refinement helps the model better capture latent dynamics by leveraging richer temporal context.

During training, the refinement step encourages the model to extract meaningful information from the posterior. Since Stage 1 optimizes the ELBO, the learned prior is already aligned with the posterior to some extent. This prevents the prior from collapsing into a trivial solution. The refinement step builds on this by using the pre-trained prior while further improving inference through contrastive learning between prior and posterior samples.

Importantly, this procedure does not introduce significant computational overhead. As shown in Appendix I.2, the refinement uses the same denoising network with different latent inputs ($\mathbf{c}$) and adds only a lightweight contrastive loss, making it efficient in practice.

### A.6 OVERVIEW

In this appendix, we first present the theoretical analysis in Section B, including the proof of Theorem 1 and accompanying discussion, followed by the ELBO derivation for `Ada-Diffuser`. In Section C, we provide an in-depth analysis of different types of MDPs and their interconnections. Section H details the full `Ada-Diffuser` algorithm, model architectures, and its relation to Bayesian filtering. Section E expands on related work, covering diffusion-based decision-making, latent state estimation via belief learning, and autoregressive diffusion models. Finally, Sections F, G, H, and I provide additional details on benchmarks, baseline implementations, and complete experimental results.

## B THEORY

### B.1 NOTATION LIST

We summarize the key notations used throughout the paper in Table A1, including variables for observed and latent states, temporal indices, and relevant mappings. These notations are used consistently in our theoretical analysis and algorithmic framework.

Also, we formally define the operators used in the following.

**Definition 1** (**Linear Operator** (Dunford & Schwartz, 1971)). *Let $\mathbf{a}$ and $\mathbf{b}$ be random variables with supports $\mathcal{A}$ and $\mathcal{B}$, respectively. The linear operator $L_{\mathbf{b}|\mathbf{a}}$ is defined as a mapping from a probability*

| Index | Explanation | Support |
|---|---|---|
| $\mathbf{x}_t$ | $[\mathbf{s}_t, \mathbf{a}_t]$, observed trajectories including state and action at time step $t$ | $\mathcal{X}_t \subseteq \mathbb{R}^{d_a + d_s}$ |
| $d_x$ | dimension of observed variables | $d_a + d_s$ |
| $\mathbf{s}_t$ | state variable at time $t$ | $\mathbf{s}_t \in \mathcal{S}_t$ |
| $\mathbf{a}_t$ | action variable at time $t$ | $\mathbf{a}_t \in \mathcal{A}_t$ |
| $r_t$ | reward received at time $t$ | $r_t \in \mathbb{R}$ |
| $\mathbf{c}_t$ | latent context variable at time $t$ | $\mathbf{c}_t \in \mathcal{C}_t$ |
| $\tau$ | trajectory sequence of $(\mathbf{s}_t, \mathbf{a}_t)$ | $\{(\mathbf{s}_0, \mathbf{a}_0), \ldots, (\mathbf{s}_T, \mathbf{a}_T)\}$ |
| $\tau_x$ | observable trajectory (states or state-actions) | $\tau_{sa}$ or $\tau_s$ |
| $\tau_c$ | sequence of latent contexts | $\{\mathbf{c}_0, \ldots, \mathbf{c}_T\}$ |
| $\boldsymbol{\tau}$ | augmented trajectory with context | $[\tau_x, \tau_c]$ |
| **Function** | | |
| $\mathcal{T}$ | transition dynamics conditioned on $\mathbf{c}_t$ | $\mathcal{T}(\mathbf{s}_t \mid \mathbf{s}_{t-1}, \mathbf{a}_{t-1}, \mathbf{c}_t)$ |
| $\mathcal{R}$ | reward function conditioned on state, action, and context | $\mathcal{R}(\mathbf{s}_t, \mathbf{a}_t, \mathbf{c}_t)$ |
| $\pi^E$ | expert policy used for generating demonstrations | $\pi^E(\mathbf{s}_t, \mathbf{c}_t)$ |
| $q_\psi$ | variational posterior for latent inference | $q_\psi(\mathbf{c}_t \mid \mathbf{x}_{t-T_x:t+1})$ |
| $p_\phi$ | latent prior distribution | $p_\phi(\mathbf{c}_t \mid \mathbf{c}_{t-1})$ |
| $p_\theta$ | generative model for transitions | $p_\theta(\mathbf{x}_t \mid \mathbf{x}_{t-1}, \mathbf{c}_t)$ |
| $\epsilon_\theta$ | denoising network in diffusion process | $\epsilon_\theta(\cdot)$ |
| **Symbol** | | |
| $\eta_t, \epsilon_t, \delta_t$ | exogenous noise in latent dynamics, state transitions, and reward | i.i.d. samples from noise distributions |
| $L_{a\mid b}$ | distribution operator from $b$ to $a$ | defined in Dunford & Schwartz (1971) |
| $k(\cdot)$ | ratio of joint probabilities used in uniqueness assumption | defined in Eq. 1 |
| $\bar{\alpha}_t$ | cumulative noise schedule in diffusion | product of forward noise factors |
| $K$ | maximum number of diffusion steps | $K \in \mathbb{N}$ |
| $T_p, T_a$ | planning and action generation horizons | $T_p, T_a \in \mathbb{N}$ |
| $T_x$ | temporal block size for latent inference | $T_x \in \mathbb{N}$ |

Table A1: List of notations, explanations, and corresponding definitions.

function $p_\mathbf{a} \in \mathcal{F}(\mathcal{A})$ to a probability function $p_\mathbf{b} \in \mathcal{F}(\mathcal{B})$, given by

$$\mathcal{F}(\mathcal{A}) \to \mathcal{F}(\mathcal{B}): \quad p_\mathbf{b} = L_{\mathbf{b}\mid\mathbf{a}} \circ p_\mathbf{a} = \int_\mathcal{A} p_{\mathbf{b}\mid\mathbf{a}}(\cdot \mid \mathbf{a})\, p_\mathbf{a}(\mathbf{a})\, d\mathbf{a}. \tag{A4}$$

Intuitively, this operator characterizes the transformation of probability distributions induced by the conditional distribution $p_{\mathbf{b}\mid\mathbf{a}}$. It provides a general representation of distributional change from $\mathbf{a}$ to $\mathbf{b}$, without imposing any parametric assumptions on the underlying distributions.

**Definition 2** (**Diagonal Operator**). *Let $\mathbf{a}$ and $\mathbf{b}$ be random variables with associated density functions $p_\mathbf{a}$ and $p_\mathbf{b}$ defined on supports $\mathcal{A}$ and $\mathcal{B}$, respectively. For a fixed value $\mathbf{b} \in \mathcal{B}$, the diagonal operator $D_{\mathbf{b}\mid\mathbf{a}}$ is defined as a linear operator that maps a density function $p_\mathbf{a} \in \mathcal{F}(\mathcal{A})$ to a function in $\mathcal{F}(\mathcal{A})$ via pointwise multiplication:*

$$D_{\mathbf{b}\mid\mathbf{a}} \circ p_\mathbf{a} = p_{\mathbf{b}\mid\mathbf{a}}(\mathbf{b} \mid \cdot) \cdot p_\mathbf{a}, \tag{A5}$$

*where $D_{\mathbf{b}\mid\mathbf{a}} = p_{\mathbf{b}\mid\mathbf{a}}(\mathbf{b} \mid \cdot)$ acts as a multiplication operator indexed by $\mathbf{b}$.*

### B.2 PROOF OF THEOREM 1

*Proof.* By the definition of data generation process (Fig. 1), the observed density is represented by:

$$p_{\mathbf{x}_{t+1}, \mathbf{x}_t, \mathbf{x}_{t-1}, \mathbf{x}_{t-2}}$$

$$= \int_{\mathcal{C}_t} \int_{\mathcal{C}_{t-1}} p_{\mathbf{x}_{t+1}, \mathbf{x}_t, \mathbf{c}_t, \mathbf{c}_{t-1}, \mathbf{x}_{t-1}, \mathbf{x}_{t-2}}\, d\mathbf{c}_t d\mathbf{c}_{t-1}$$

$$= \int_{\mathcal{C}_t} \int_{\mathcal{C}_{t-1}} p_{\mathbf{x}_{t+1} \mid \mathbf{x}_t, \mathbf{x}_{t-1}, \mathbf{x}_{t-2}, \mathbf{c}_t, \mathbf{c}_{t-1}} p_{\mathbf{x}_t, \mathbf{c}_t \mid \mathbf{x}_{t-1}, \mathbf{x}_{t-2}, \mathbf{c}_{t-1}} p_{\mathbf{c}_{t-1}, \mathbf{x}_{t-1}, \mathbf{x}_{t-2}}\, d\mathbf{c}_t d\mathbf{c}_{t-1}$$

$$= \int_{\mathcal{C}_t} \int_{\mathcal{C}_{t-1}} p_{\mathbf{x}_{t+1} \mid \mathbf{x}_t, \mathbf{c}_t} p_{\mathbf{x}_t, \mathbf{c}_t \mid \mathbf{x}_{t-1}, \mathbf{c}_{t-1}} p_{\mathbf{c}_{t-1}, \mathbf{x}_{t-1}, \mathbf{x}_{t-2}}\, d\mathbf{c}_t d\mathbf{c}_{t-1}$$

$$= \int_{\mathcal{C}_t} \int_{\mathcal{C}_{t-1}} p_{\mathbf{x}_{t+1} \mid \mathbf{x}_t, \mathbf{c}_t} p_{\mathbf{x}_t \mid \mathbf{x}_{t-1}, \mathbf{c}_t, \mathbf{c}_{t-1}} p_{\mathbf{c}_t \mid \mathbf{x}_{t-1}, \mathbf{x}_{t-2}, \mathbf{c}_{t-1}} p_{\mathbf{x}_{t-1}, \mathbf{x}_{t-2}, \mathbf{c}_{t-1}}\, d\mathbf{c}_t d\mathbf{c}_{t-1}.$$

$$= \int_{\mathcal{C}_t} \int_{\mathcal{C}_{t-1}} p_{\mathbf{x}_{t+1} \mid \mathbf{x}_t, \mathbf{c}_t} p_{\mathbf{x}_t \mid \mathbf{x}_{t-1}, \mathbf{c}_t, \mathbf{c}_{t-1}} p_{\mathbf{c}_t, \mathbf{x}_{t-1}, \mathbf{x}_{t-2}, \mathbf{c}_{t-1}}\, d\mathbf{c}_t d\mathbf{c}_{t-1}.$$

Then, the property of Markov process presents conditional independence, organized as follows:

$$p_{\mathbf{x}_{t+1},\mathbf{x}_t,\mathbf{x}_{t-1},\mathbf{x}_{t-2}} = \int_{\mathcal{C}_t} p_{\mathbf{x}_{t+1}|\mathbf{x}_t,\mathbf{c}_t} p_{\mathbf{x}_t|\mathbf{x}_{t-1},\mathbf{c}_t} \left( \int_{\mathcal{C}_{t-1}} p_{\mathbf{c}_t,\mathbf{c}_{t-1},\mathbf{x}_{t-1},\mathbf{x}_{t-2}} \, d\mathbf{c}_{t-1} \right) d\mathbf{c}_t$$

$$= \int_{\mathcal{C}_t} p_{\mathbf{x}_{t+1}|\mathbf{x}_t,\mathbf{c}_t} p_{\mathbf{x}_t|\mathbf{x}_{t-1},\mathbf{c}_t} p_{\mathbf{c}_t,\mathbf{x}_{t-1},\mathbf{x}_{t-2}} \, d\mathbf{c}_t. \tag{A6}$$

Eq. A6 can be denoted in terms of operators: given values of $(\mathbf{x}_t, \mathbf{x}_{t-1}) \in \mathcal{X}_t \times \mathcal{X}_{t-1}$, Eq. A6 is

$$L_{\mathbf{x}_{t+1},\mathbf{x}_t,\mathbf{x}_{t-1},\mathbf{x}_{t-2}} = L_{\mathbf{x}_{t+1}|\mathbf{x}_t,\mathbf{c}_t} D_{\mathbf{x}_t|\mathbf{x}_{t-1},\mathbf{c}_t} L_{\mathbf{c}_t,\mathbf{x}_{t-1},\mathbf{x}_{t-2}}. \tag{A7}$$

Notably, Eq. A7 is the operator representation of the observed density function in 4 measurements. Furthermore, the structure of Markov process implies the following two equalities:

$$p_{\mathbf{x}_{t+1},\mathbf{x}_t,\mathbf{x}_{t-1},\mathbf{x}_{t-2}} = \int_{\mathcal{C}_t} p_{\mathbf{x}_{t+1}|\mathbf{x}_t,\mathbf{c}_t} p_{\mathbf{x}_t,\mathbf{c}_t,\mathbf{x}_{t-1},\mathbf{x}_{t-2}} \, d\mathbf{c}_t,$$

$$p_{\mathbf{x}_t,\mathbf{c}_t,\mathbf{x}_{t-1},\mathbf{x}_{t-2}} = \int_{\mathcal{C}_{t-1}} p_{\mathbf{x}_t,\mathbf{c}_t|\mathbf{x}_{t-1},\mathbf{c}_{t-1}} p_{\mathbf{c}_{t-1},\mathbf{x}_{t-1},\mathbf{x}_{t-2}} \, d\mathbf{c}_{t-1}. \tag{A8}$$

For any fixed $(\mathbf{x}_t, \mathbf{x}_{t-1}) \in \mathcal{X}_t \times \mathcal{X}_{t-1}$, we notate Eq. A8 in terms of operators as follows:

$$L_{\mathbf{x}_{t+1},\mathbf{x}_t,\mathbf{x}_{t-1},\mathbf{x}_{t-2}} = L_{\mathbf{x}_{t+1}|\mathbf{x}_t,\mathbf{c}_t} L_{\mathbf{x}_t,\mathbf{c}_t,\mathbf{x}_{t-1},\mathbf{x}_{t-2}},$$

$$L_{\mathbf{x}_t,\mathbf{c}_t,\mathbf{x}_{t-1},\mathbf{x}_{t-2}} = L_{\mathbf{x}_t,\mathbf{c}_t|\mathbf{x}_{t-1},\mathbf{c}_{t-1}} L_{\mathbf{c}_{t-1},\mathbf{x}_{t-1},\mathbf{x}_{t-2}}. \tag{A9}$$

Substituting the second line in Eq. A9 into R.H.S. of the first equation, we obtain

$$L_{\mathbf{x}_{t+1},\mathbf{x}_t,\mathbf{x}_{t-1},\mathbf{x}_{t-2}} = L_{\mathbf{x}_{t+1}|\mathbf{x}_t,\mathbf{c}_t} L_{\mathbf{x}_t,\mathbf{c}_t|\mathbf{x}_{t-1},\mathbf{c}_{t-1}} L_{\mathbf{c}_{t-1},\mathbf{x}_{t-1},\mathbf{x}_{t-2}}$$

$$\Leftrightarrow L_{\mathbf{x}_t,\mathbf{c}_t|\mathbf{x}_{t-1},\mathbf{c}_{t-1}} L_{\mathbf{c}_{t-1},\mathbf{x}_{t-1},\mathbf{x}_{t-2}} = L_{\mathbf{x}_{t+1}|\mathbf{x}_t,\mathbf{c}_t}^{-1} L_{\mathbf{x}_{t+1},\mathbf{x}_t,\mathbf{x}_{t-1},\mathbf{x}_{t-2}}. \tag{A10}$$

The second line above uses Assumption 2 that $L_{\mathbf{x}_{t+1}|\mathbf{x}_t,\mathbf{c}_t}^{-1}$ is injective. Next, we show how to eliminate $L_{\mathbf{c}_{t-1},\mathbf{x}_{t-1},\mathbf{x}_{t-2}}$ from the above. Consider 3 measurements $\{\mathbf{x}_t, \mathbf{x}_{t-1}, \mathbf{x}_{t-2}\}$, we have

$$p_{\mathbf{x}_t,\mathbf{x}_{t-1},\mathbf{x}_{t-2}} = \int_{\mathcal{C}_{t-1}} p_{\mathbf{x}_t|\mathbf{x}_{t-1},\mathbf{c}_{t-1}} p_{\mathbf{c}_{t-1},\mathbf{x}_{t-1},\mathbf{x}_{t-2}} \, d\mathbf{c}_{t-1}, \tag{A11}$$

which, in operator notation (for fixed $\mathbf{x}_{t-1}$), is denoted as

$$L_{\mathbf{x}_t,\mathbf{x}_{t-1},\mathbf{x}_{t-2}} = L_{\mathbf{x}_t|\mathbf{x}_{t-1},\mathbf{c}_{t-1}} L_{\mathbf{c}_{t-1},\mathbf{x}_{t-1},\mathbf{x}_{t-2}},$$

$$\Rightarrow \quad L_{\mathbf{c}_{t-1},\mathbf{x}_{t-1},\mathbf{x}_{t-2}} = L_{\mathbf{x}_t|\mathbf{x}_{t-1},\mathbf{c}_{t-1}}^{-1} L_{\mathbf{x}_t,\mathbf{x}_{t-1},\mathbf{x}_{t-2}}. \tag{A12}$$

The R.H.S. applies Assumption 2. Hence, substituting the above into Eq. A10, we obtain:

$$L_{\mathbf{x}_t,\mathbf{c}_t|\mathbf{x}_{t-1},\mathbf{c}_{t-1}} L_{\mathbf{x}_t|\mathbf{x}_{t-1},\mathbf{c}_{t-1}}^{-1} L_{\mathbf{x}_t,\mathbf{x}_{t-1},\mathbf{x}_{t-2}} = L_{\mathbf{x}_{t+1}|\mathbf{x}_t,\mathbf{c}_t}^{-1} L_{\mathbf{x}_{t+1},\mathbf{x}_t,\mathbf{x}_{t-1},\mathbf{x}_{t-2}}$$

$$\Rightarrow \quad L_{\mathbf{x}_t,\mathbf{c}_t|\mathbf{x}_{t-1},\mathbf{c}_{t-1}} = L_{\mathbf{x}_{t+1}|\mathbf{x}_t,\mathbf{c}_t}^{-1} L_{\mathbf{x}_{t+1},\mathbf{x}_t,\mathbf{x}_{t-1},\mathbf{x}_{t-2}} L_{\mathbf{x}_t,\mathbf{x}_{t-1},\mathbf{x}_{t-2}}^{-1} L_{\mathbf{x}_t,\mathbf{x}_{t-1},\mathbf{c}_{t-1}}. \tag{A13}$$

The second line applies Assumption 2 to post-multiply by $L_{\mathbf{x}_t,\mathbf{x}_{t-1},\mathbf{x}_{t-2}}^{-1}$, while in the third line, we postmultiply both sides by $L_{\mathbf{x}_t|\mathbf{x}_{t-1},\mathbf{c}_{t-1}}$.

For all $\mathbf{x}_t$, choose a $\mathbf{x}_{t-1}$ and a neighborhood $\mathcal{N}^r$ around $(\mathbf{x}_t, \mathbf{x}_{t-1})$ to satisfy Assumption 2, and pick a $(\bar{\mathbf{x}}_t, \bar{\mathbf{x}}_{t-1})$ within the neighborhood $\mathcal{N}^r$. Because $(\bar{\mathbf{x}}_t, \bar{\mathbf{x}}_{t-1}) \in \mathcal{N}^r$, we also know that $(\mathbf{x}_t, \bar{\mathbf{x}}_{t-1})$, $(\bar{\mathbf{x}}_t, \mathbf{x}_{t-1}) \in \mathcal{N}^r$. The joint distribution of of observations can be represented by Eq. A7:

$$L_{\mathbf{x}_{t+1},\mathbf{x}_t,\mathbf{x}_{t-1},\mathbf{x}_{t-2}} = L_{\mathbf{x}_{t+1}|\mathbf{x}_t,\mathbf{c}_t} D_{\mathbf{x}_t|\mathbf{x}_{t-1},\mathbf{c}_t} L_{\mathbf{c}_t,\mathbf{x}_{t-1},\mathbf{x}_{t-2}}. \tag{A14}$$

The first term on the R.H.S., $L_{\mathbf{x}_{t+1}|\mathbf{x}_t,\mathbf{c}_t}$, does not depend on $\mathbf{x}_{t-1}$, and the last term $L_{\mathbf{c}_t,\mathbf{x}_{t-1},\mathbf{x}_{t-2}}$ does not depend on $\mathbf{x}_t$. This feature suggests that, by evaluating Eq. A7 at the four pairs of points

$(\mathbf{x}_t, \mathbf{x}_{t-1})$, $(\bar{\mathbf{x}}_t, \mathbf{x}_{t-1})$, $(\mathbf{x}_t, \bar{\mathbf{x}}_{t-1})$, $(\bar{\mathbf{x}}_t, \bar{\mathbf{x}}_{t-1})$, each pair of equations will share the same operator representation in common. Specifically:

$$L_{\mathbf{x}_{t+1},\mathbf{x}_t,\mathbf{x}_{t-1},\mathbf{x}_{t-2}} = L_{\mathbf{x}_{t+1}|\mathbf{x}_t,\mathbf{c}_t} D_{\mathbf{x}_t|\mathbf{x}_{t-1},\mathbf{c}_t} L_{\mathbf{c}_t,\mathbf{x}_{t-1},\mathbf{x}_{t-2}}, \tag{A15}$$

$$L_{\mathbf{x}_{t+1},\bar{\mathbf{x}}_t,\mathbf{x}_{t-1},\mathbf{x}_{t-2}} = L_{\mathbf{x}_{t+1}|\bar{\mathbf{x}}_t,\mathbf{c}_t} D_{\bar{\mathbf{x}}_t|\mathbf{x}_{t-1},\mathbf{c}_t} L_{\mathbf{c}_t,\mathbf{x}_{t-1},\mathbf{x}_{t-2}}, \tag{A16}$$

$$L_{\mathbf{x}_{t+1},\mathbf{x}_t,\bar{\mathbf{x}}_{t-1},\mathbf{x}_{t-2}} = L_{\mathbf{x}_{t+1}|\mathbf{x}_t,\mathbf{c}_t} D_{\mathbf{x}_t|\bar{\mathbf{x}}_{t-1},\mathbf{c}_t} L_{\mathbf{c}_t,\bar{\mathbf{x}}_{t-1},\mathbf{x}_{t-2}}, \tag{A17}$$

$$L_{\mathbf{x}_{t+1},\bar{\mathbf{x}}_t,\bar{\mathbf{x}}_{t-1},\mathbf{x}_{t-2}} = L_{\mathbf{x}_{t+1}|\bar{\mathbf{x}}_t,\mathbf{c}_t} D_{\bar{\mathbf{x}}_t|\bar{\mathbf{x}}_{t-1},\mathbf{c}_t} L_{\mathbf{c}_t,\bar{\mathbf{x}}_{t-1},\mathbf{x}_{t-2}}. \tag{A18}$$

Assumption 2 implies that $L_{\mathbf{x}_{t+1}|\bar{\mathbf{x}}_t,\mathbf{c}_t}$ is injective. Moreover, Assumption 3 implies $p_{\mathbf{x}_t|\mathbf{x}_{t-1},\mathbf{c}_t}(\mathbf{x}_t \mid \mathbf{x}_{t-1}, \mathbf{c}_t) > 0$ for all $\mathbf{c}_t$, so that $D_{\bar{\mathbf{x}}_t|\mathbf{x}_{t-1},\mathbf{c}_t}$ is invertible. We can then solve for $L_{\mathbf{c}_t,\mathbf{x}_{t-1},\mathbf{x}_{t-2}}$ from Eq. A16 as

$$D^{-1}_{\bar{\mathbf{x}}_t|\mathbf{x}_{t-1},\mathbf{c}_t} L^{-1}_{\mathbf{x}_{t+1}|\bar{\mathbf{x}}_t,\mathbf{c}_t} L_{\mathbf{x}_{t+1},\bar{\mathbf{x}}_t,\mathbf{x}_{t-1},\mathbf{x}_{t-2}} = L_{\mathbf{c}_t,\mathbf{x}_{t-1},\mathbf{x}_{t-2}}. \tag{A19}$$

Plugging this expression into Eq. A15 leads to

$$L_{\mathbf{x}_{t+1},\mathbf{x}_t,\mathbf{x}_{t-1},\mathbf{x}_{t-2}} = L_{\mathbf{x}_{t+1}|\mathbf{x}_t,\mathbf{c}_t} D_{\mathbf{x}_t|\mathbf{x}_{t-1},\mathbf{c}_t} D^{-1}_{\bar{\mathbf{x}}_t|\mathbf{x}_{t-1},\mathbf{c}_t} L^{-1}_{\mathbf{x}_{t+1}|\bar{\mathbf{x}}_t,\mathbf{c}_t} L_{\mathbf{x}_{t+1},\bar{\mathbf{x}}_t,\mathbf{x}_{t-1},\mathbf{x}_{t-2}}. \tag{A20}$$

At this point, we have decomposed the observable joint operator and expressed it in terms of latent-conditioned transitions, enabling spectral analysis for identifying latent structure.

Lemma 1 of (Hu & Schennach, 2008) shows that, given the injectivity of $L_{\mathbf{x}_{t-2},\bar{\mathbf{x}}_{t-1},\mathbf{x}_t,\mathbf{x}_{t+1}}$ as in Assumption 2, we can postmultiply by $L^{-1}_{\mathbf{x}_{t+1},\mathbf{x}_t,\mathbf{x}_{t-1},\mathbf{x}_{t-2}}$ to obtain:

$$\mathbf{M} \equiv L_{\mathbf{x}_{t+1},\mathbf{x}_t,\mathbf{x}_{t-1},\mathbf{x}_{t-2}} L^{-1}_{\mathbf{x}_{t+1},\mathbf{x}_t,\bar{\mathbf{x}}_{t-1},\mathbf{x}_{t-2}} = L_{\mathbf{x}_{t+1}|\mathbf{x}_t,\mathbf{c}_t} D_{\mathbf{x}_t|\mathbf{x}_{t-1},\mathbf{c}_t} D^{-1}_{\bar{\mathbf{x}}_t|\mathbf{x}_{t-1},\mathbf{c}_t} L^{-1}_{\mathbf{x}_{t+1}|\bar{\mathbf{x}}_t,\mathbf{c}_t}. \tag{A21}$$

Similarly, manipulations of Eq. A17 and A18 lead to

$$\mathbf{N} \equiv L_{\mathbf{x}_{t+1},\bar{\mathbf{x}}_t,\mathbf{x}_{t-1},\mathbf{x}_{t-2}} L^{-1}_{\mathbf{x}_{t+1},\mathbf{x}_t,\bar{\mathbf{x}}_{t-1},\mathbf{x}_{t-2}} = L_{\mathbf{x}_{t+1}|\bar{\mathbf{x}}_t,\mathbf{c}_t} D_{\bar{\mathbf{x}}_t|\bar{\mathbf{x}}_{t-1},\mathbf{c}_t} D^{-1}_{\mathbf{x}_t|\bar{\mathbf{x}}_{t-1},\mathbf{c}_t} L^{-1}_{\mathbf{x}_{t+1}|\mathbf{x}_t,\mathbf{c}_t}. \tag{A22}$$

Assumption 2 guarantees that, for any $\mathbf{x}_t$, $(\bar{\mathbf{x}}_t, \mathbf{x}_{t-1}, \bar{\mathbf{x}}_{t-1})$ exist so that Eq. A21 and Eq. A22 are valid operations. Finally, we postmultiply Eq. A21 by Eq. A22 to obtain:

$$\mathbf{MN} = L_{\mathbf{x}_{t+1}|\mathbf{x}_t,\mathbf{c}_t} D_{\mathbf{x}_t|\mathbf{x}_{t-1},\mathbf{c}_t} D^{-1}_{\bar{\mathbf{x}}_t|\mathbf{x}_{t-1},\mathbf{c}_t} \left( L_{\mathbf{x}_{t+1}|\bar{\mathbf{x}}_t,\mathbf{c}_t} L_{\mathbf{x}_{t+1}|\bar{\mathbf{x}}_t,\mathbf{c}_t} \right) \times D_{\bar{\mathbf{x}}_t|\bar{\mathbf{x}}_{t-1},\mathbf{c}_t} D^{-1}_{\mathbf{x}_t|\bar{\mathbf{x}}_{t-1},\mathbf{c}_t} L^{-1}_{\mathbf{x}_{t+1}|\mathbf{x}_t,\mathbf{c}_t}$$

$$= L_{\mathbf{x}_{t+1}|\mathbf{x}_t,\mathbf{c}_t} \left( D_{\mathbf{x}_t|\mathbf{x}_{t-1},\mathbf{c}_t} D^{-1}_{\bar{\mathbf{x}}_t|\mathbf{x}_{t-1},\mathbf{c}_t} D_{\bar{\mathbf{x}}_t|\bar{\mathbf{x}}_{t-1},\mathbf{c}_t} D^{-1}_{\mathbf{x}_t|\bar{\mathbf{x}}_{t-1},\mathbf{c}_t} \right) L^{-1}_{\mathbf{x}_{t+1}|\mathbf{x}_t,\mathbf{c}_t}$$

$$\equiv L_{\mathbf{x}_{t+1}|\mathbf{x}_t,\mathbf{c}_t} D_{\mathbf{x}_t,\bar{\mathbf{x}}_t,\mathbf{x}_{t-1},\bar{\mathbf{x}}_{t-1},\mathbf{c}_t} L^{-1}_{\mathbf{x}_{t+1}|\mathbf{x}_t,\mathbf{c}_t}, \tag{A23}$$

where

$$\left( D_{\mathbf{x}_t,\bar{\mathbf{x}}_t,\mathbf{x}_{t-1},\bar{\mathbf{x}}_{t-1},\mathbf{c}_t} h \right)(\mathbf{c}_t) = \left( D_{\mathbf{x}_t|\mathbf{x}_{t-1},\mathbf{c}_t} D^{-1}_{\bar{\mathbf{x}}_t|\mathbf{x}_{t-1},\mathbf{c}_t} D_{\bar{\mathbf{x}}_t|\bar{\mathbf{x}}_{t-1},\mathbf{c}_t} D^{-1}_{\mathbf{x}_t|\bar{\mathbf{x}}_{t-1},\mathbf{c}_t} h \right)(\mathbf{c}_t)$$

$$= \frac{p_{\mathbf{x}_t|\mathbf{x}_{t-1},\mathbf{c}_t}(\mathbf{x}_t \mid \mathbf{x}_{t-1}, \mathbf{c}_t) p_{\mathbf{x}_t|\mathbf{x}_{t-1},\mathbf{c}_t}(\bar{\mathbf{x}}_t \mid \bar{\mathbf{x}}_{t-1}, \mathbf{c}_t)}{p_{\mathbf{x}_t|\mathbf{x}_{t-1},\mathbf{c}_t}(\bar{\mathbf{x}}_t \mid \mathbf{x}_{t-1}, \mathbf{c}_t) p_{\mathbf{x}_t|\mathbf{x}_{t-1},\mathbf{c}_t}(\mathbf{x}_t \mid \bar{\mathbf{x}}_{t-1}, \mathbf{c}_t)} h(\mathbf{c}_t)$$

$$\equiv k(\mathbf{x}_t, \bar{\mathbf{x}}_t, \mathbf{x}_{t-1}, \bar{\mathbf{x}}_{t-1}, \mathbf{c}_t) h(\mathbf{c}_t). \tag{A24}$$

This equation implies that the observed operator $\mathbf{MN}$ on the L.H.S. of Eq. A25 has an inherent eigenvalue–eigenfunction decomposition, with the eigenvalues corresponding to the function $k(\mathbf{x}_t, \bar{\mathbf{x}}_t, \mathbf{x}_{t-1}, \bar{\mathbf{x}}_{t-1}, \mathbf{c}_t)$ and the eigenfunctions corresponding to the density $p_{\mathbf{x}_{t+1}|\mathbf{x}_t,\mathbf{c}_t}(\cdot \mid \mathbf{x}_t, \mathbf{c}_t)$.

The decomposition in Eq. A25 is similar to the decomposition in nonparametric identification (Hu & Schennach, 2008; Carroll et al., 2010). First, Assumption 3 ensures this decomposition is unique. Second, the operator $\mathbf{MN}$ on the L.H.S. has the same spectrum as the diagonal operator $D_{\mathbf{x}_t,\bar{\mathbf{x}}_t,\mathbf{x}_{t-1},\bar{\mathbf{x}}_{t-1},\mathbf{c}_t}$. Assumption 3 guarantees that the spectrum of the diagonal operator is bounded. Since an operator is bounded by the largest element of its spectrum, Assumption 3 also implies that the operator $\mathbf{MN}$ is bounded, whence we can apply Theorem XV.4.3.5 from (Dunford & Schwartz, 1971) to show the uniqueness of the spectral decomposition of bounded linear operators:

$$L_{\mathbf{x}_{t+1}|\mathbf{x}_t,\mathbf{c}_t} = C L_{\mathbf{x}_{t+1}|\mathbf{x}_t,\mathbf{c}_t} P^{-1}. \quad D_{\mathbf{x}_t,\bar{\mathbf{x}}_t,\mathbf{x}_{t-1},\bar{\mathbf{x}}_{t-1},\mathbf{c}_t} = P D_{\mathbf{x}_t,\bar{\mathbf{x}}_t,\mathbf{x}_{t-1},\bar{\mathbf{x}}_{t-1},\mathbf{c}_t} P^{-1} \tag{A25}$$

where $C$ is a scalar accounting for scaling indeterminacy and $P$ is a permutation on the order of elements in $D_{\hat{\mathbf{x}}_t|\hat{\mathbf{c}}_t}$, as discussed in (Dunford & Schwartz, 1971). These forms of indeterminacy are analogous to those in eigendecomposition, which can be viewed as a finite-dimensional special case.

We will show why the uniqueness of spectral decomposition is informative for identifications. First,

$$\int_{\hat{\mathcal{X}}_{t+1}} p_{\hat{\mathbf{x}}_{t+1}|\hat{\mathbf{x}}_t,\hat{\mathbf{c}}_t} \, d\hat{\mathbf{x}}_{t+1} = 1 \tag{A26}$$

must hold for every $\hat{\mathbf{c}}_t$ due to normalizing condition, one only solution is to set $C = 1$.

Second, Assumption 3 implies that Eq. A25 imply that the eigenvalues $k(\mathbf{x}_t, \bar{\mathbf{x}}_t, \mathbf{x}_{t-1}, \bar{\mathbf{x}}_{t-1}, \mathbf{c}_t)$ are distinct for different values $\mathbf{c}_t$. If several $\mathbf{c}_t$ yield identical eigenvalues, the associated eigenfunctions cannot be uniquely identified, as any linear combination of them remains valid. Therefore, for each $\mathbf{x}_t$, one can choose $\bar{\mathbf{x}}t, \mathbf{x}t-1, \bar{\mathbf{x}}_{t-1}$ such that the eigenvalues differ for all $\mathbf{c}_t$.

Ultimately, the unorder of eigenvalues/eigenfunctions is left. The operator, $L_{\mathbf{x}_{t+1}|\mathbf{x}_t,\mathbf{c}_t}$, corresponding to the set $\{p_{\mathbf{x}_{t+1}|\mathbf{x}_t,\mathbf{c}_t}(\cdot \mid \mathbf{x}_t, \mathbf{c}_t)\}$ for all $\mathbf{x}_t, \mathbf{c}_t$, admits a unique solution (orderibng ambiguity of eigendecomposition only changes the entry position):

$$\{p_{\mathbf{x}_{t+1}|\mathbf{x}_t,\mathbf{c}_t}(\cdot \mid \mathbf{x}_t, \mathbf{c}_t)\} = \{p_{\mathbf{x}_{t+1}|\hat{\mathbf{x}}_t,\hat{\mathbf{c}}_t}(\mathbf{x}_{t+1} \mid \hat{\mathbf{x}}_t, \hat{\mathbf{c}}_t)\}, \quad \text{for all } \mathbf{x}_t, \mathbf{c}_t, \hat{\mathbf{x}}_t, \hat{\mathbf{c}}_t \tag{A27}$$

Due to the set is unorder, the only way to match the R.H.S. with the L.H.S. in a consistent order is to exchange the conditioning variables, that is,

$$\begin{aligned} & \{p_{\mathbf{x}_{t+1}|\mathbf{x}_t,\mathbf{c}_t}(\cdot \mid \mathbf{x}_t^{(1)}, \mathbf{c}_t^{(1)}), \, p_{\mathbf{x}_{t+1}|\mathbf{x}_t,\mathbf{c}_t}(\cdot \mid \mathbf{x}_t^{(2)}, \mathbf{c}_t^{(2)}), \, \ldots\} \\ & = \{p_{\mathbf{x}_{t+1}|\hat{\mathbf{x}}_t,\hat{\mathbf{c}}_t}(\cdot \mid \hat{\mathbf{x}}_t^{(1)}, \hat{\mathbf{c}}_t^{(1)}), \, p_{\mathbf{x}_{t+1}|\hat{\mathbf{x}}_t,\hat{\mathbf{c}}_t}(\cdot \mid \hat{\mathbf{x}}_t^{(2)}, \hat{\mathbf{c}}_t^{(2)}), \, \ldots\} \end{aligned} \tag{A28}$$

$$\begin{aligned} \Rightarrow \quad & [p_{\mathbf{x}_{t+1}|\mathbf{x}_t,\mathbf{c}_t}(\cdot \mid \mathbf{x}_t^{(\pi(1))}, \mathbf{c}_t^{(\pi(1))}), \, p_{\mathbf{x}_{t+1}|\mathbf{x}_t,\mathbf{c}_t}(\cdot \mid \mathbf{x}_t^{(\pi(2))}, \mathbf{c}_t^{(\pi(2))}), \, \ldots] \\ & = [p_{\mathbf{x}_{t+1}|\hat{\mathbf{x}}_t,\hat{\mathbf{c}}_t}(\cdot \mid \hat{\mathbf{x}}_t^{(\pi(1))}, \hat{\mathbf{c}}_t^{(\pi(1))}), \, p_{\mathbf{x}_{t+1}|\hat{\mathbf{x}}_t,\hat{\mathbf{c}}_t}(\cdot \mid \hat{\mathbf{x}}_t^{(\pi(2))}, \hat{\mathbf{c}}_t^{(\pi(2))}), \, \ldots] \end{aligned}$$

where superscript $(\cdot)$ denotes the index of the conditioning variables $[\mathbf{x}_t, \mathbf{c}_t]$, and $\pi$ is reindexing the conditioning variables. We use a relabeling map $H$ to represent its corresponding value mapping:

$$p_{\mathbf{x}_{t+1}|\mathbf{x}_t,\mathbf{c}_t}(\cdot \mid H(\mathbf{x}_t, \mathbf{c}_t)) = p_{\mathbf{x}_{t+1}|\hat{\mathbf{x}}_t,\hat{\mathbf{c}}_t}(\cdot \mid \hat{\mathbf{x}}_t, \hat{\mathbf{c}}_t), \quad \text{for all } \mathbf{x}_t, \mathbf{c}_t, \hat{\mathbf{x}}_t, \hat{\mathbf{c}}_t \tag{A29}$$

By Assumption 3, different $\mathbf{c}_t$ corresponds to different $p_{\mathbf{x}_{t+1}|\mathbf{x}_t,\mathbf{c}_t}(\cdot \mid H(\mathbf{x}_t, \mathbf{c}_t))$, which indicates that there is no repeated element in $\{p_{\mathbf{x}_{t+1}|\mathbf{x}_t,\mathbf{c}_t}(\cdot \mid H(\mathbf{x}_t, \mathbf{c}_t))\}$ and $\{p_{\mathbf{x}_{t+1}|\hat{\mathbf{x}}_t,\hat{\mathbf{c}}_t}(\cdot \mid \hat{\mathbf{x}}_t, \hat{\mathbf{c}}_t)\}$. Such uniqueness ensure that the relabelling map $H$ is one-to-one.

Furthermore, Assumption 3 implies that $p_{\mathbf{x}_{t+1},|\mathbf{x}_t,\mathbf{c}_t}(\cdot \mid H(\mathbf{x}_t, \mathbf{c}_t))$ corresponds a unique $H(\mathbf{x}_t, \mathbf{c}_t)$. The same holds for the $p_{\mathbf{x}_{t+1}|\hat{\mathbf{x}}_t,\hat{\mathbf{c}}_t}(\cdot \mid \hat{\mathbf{x}}_t, \hat{\mathbf{c}}_t)$, implying that

$$p_{\mathbf{x}_{t+1}|\mathbf{x}_t,\mathbf{c}_t}(\cdot \mid H(\mathbf{x}_t, \mathbf{c}_t)) = p_{\mathbf{x}_{t+1}|\hat{\mathbf{x}}_t,\hat{\mathbf{c}}_t}(\cdot \mid \hat{\mathbf{x}}_t, \hat{\mathbf{c}}_t) \implies \hat{\mathbf{x}}_t, \hat{\mathbf{c}}_t = H(\mathbf{x}_t, \mathbf{c}_t) \tag{A30}$$

Since the observation $\mathbf{x}_t$ is known and suppose $\hat{\mathbf{x}}_t = \mathbf{x}_t$, this relationship indeed represents an invertible transformation between $\hat{\mathbf{c}}_t$ and $\mathbf{c}_t$ as

$$\hat{\mathbf{c}}_t = h(\mathbf{c}_t). \tag{A31}$$

which ensures that $p(\mathbf{c}_t \mid \mathbf{x}_{t-2:t+1})$ can be identifiable up to an invertible transformation on the latent variables $\hat{\mathbf{c}}_t = h(\mathbf{c}_t)$ $\qquad\square$

### B.3 THEORY-ALGORITHM ALIGNMENT

Here, we provide a more detailed description of our theoretical foundations, model design guidance, and algorithmic implementation. This complements the high-level summary in the main paper and offers additional context about the technical depth behind our contributions.

### B.3.1 NON-PARAMETRIC IDENTIFIABILITY THEORY

We establish this non-parametric identifiability result that gives sufficient conditions for recovering latent contexts from reinforcement learning (RL) trajectories using short temporal blocks. Each block

includes a small number of future steps, which allows the model to reason about both the immediate past and the near future. Formally, we prove that under mild and broadly applicable assumptions, the latent context $\mathbf{c}_t$ driving the generative process of the observed states and actions $(\mathbf{x}_t, \mathbf{a}_t)$ can be recovered up to an equivalence class. This identifiability guarantee is important because it shows that latent-aware planning can be theoretically justified even when the environment contains unobserved factors or task-dependent variations.

### B.3.2 MODEL DESIGN GUIDANCE

The theoretical result directly guides the design of our causal diffusion model. To leverage Theorem 1, the diffusion model must not only generate trajectories but also recover the true latent factors. Concretely, the model must:

1. capture **temporal dependencies** across short blocks of states and actions;

2. jointly model **observable and latent variables**;

3. enforce **conditions for identifiability**, ensuring that the latent $\mathbf{c}_t$ can be isolated from the observed sequence.

These design requirements inform our noise schedule and the coupling of autoregressive denoising with latent refinement.

This provides guidance for the algorithm design:

**Autoregressive Denoising.** We model temporal dependencies over both observable and latent variables using an autoregressive diffusion process. At each step, $\mathbf{x}_t$ is denoised while conditioning on partially denoised past states and inferred latent variables from a short temporal block (Section 4.1). This schedule results in a structured temporal-latent modeling process that better preserves long-range dependencies.

**Backward Refinement.** To explicitly identify latent contexts whose posterior depends on future observations (e.g., $x_{t+1}$, guided by Theorem 1), we introduce a backward refinement step. At the second-to-last denoising stage, we refine a partial state $x_t^{k_1}$ using the initial estimate of $\hat{c}_t$ sampled from the prior and $x_{t+1}$ as additional evidence. The refined $\hat{c}_t$ is then used to produce the final denoised state $x_t^0$. During training, this backward refinement is enforced to satisfy the identifiability conditions. At inference time (zig-zag sampling), we substitute $x_{t+1}$ with a predicted estimate to maintain efficiency.

**Unification.** The autoregressive denoising and backward refinement are integrated into a single noise schedule, enabling joint modeling of temporal dependencies and latent variables. Our implementation follows a four-step refinement scheme but can be extended to more steps if needed. Notably, despite the additional refinement, the method remains computationally efficient (see Appendix I.1). Further acceleration is possible via Picard iteration, which parallelizes refinement steps and reduces inference runtime by about $25\%$.

### B.3.3 DISCUSSION ON ASSUMPTIONS

### B.3.4 RELAXING ASSUMPTION 1 (BEYOND FIRST–ORDER MARKOV)

We can relax the first–order Markov assumption to an $n$-order Markov structure with delayed/cumulative influences without altering the core identifiability argument. Suppose the generative process satisfies

$$p(\mathbf{x}_{t+1} \mid \mathbf{x}_{1:t}, \mathbf{a}_{1:t}, \mathbf{c}_{1:t}) \;=\; p\big(\mathrm{v}(x_{t+1} \mid \mathbf{x}_{t:t-n+1}, \mathbf{a}_{t:t-n+1}, \mathbf{c}_{t:t-n+1}\big),$$

and that the conditioning sets across *non overlapping* lags exhibit block–wise conditional independence (the same separation conditions used in Theorem 1). Then there exists a finite window of observations whose statistics identify the contemporaneous block $[\mathbf{c}_t, \mathbf{x}_t]$ up to an invertible reparameterization.

| Latent Type | Block size | Probing Acc | $R^2$ |
|---|---|---|---|
| Delayed | 6 | 0.81 | 0.72 |
| Delayed | 8 | 0.85 | 0.78 |
| Delayed | 10 | 0.88 | 0.81 |
| Delayed | 20 | 0.91 | 0.86 |
| Cumulative | 6 | 0.84 | 0.75 |
| Cumulative | 8 | 0.87 | 0.79 |
| Cumulative | 10 | 0.89 | 0.83 |
| Cumulative | 20 | 0.93 | 0.88 |

Table A2: Identification under delayed and cumulative latent effects. Larger is better.

**Concrete identification statement.** Let $W_t = (\mathbf{x}_{t-2n:t+2n})$ denote a $4n+1$-length observation window.[4] Assume: (i) time direction is known (so $[\mathbf{c}_t, \mathbf{x}_t] \rightarrow [\mathbf{c}_{t+1}\mathbf{v}, x_{t+1}]$ is oriented); (ii) the variability (support) conditions from Theorem 1 hold for the $n$-lag blocks; and (iii) block–wise independence across non–overlapping lags is satisfied. Then there exists an invertible map $H$ such that
$$[\mathbf{c}_t, \mathbf{x}_t] = H(W_t),$$
so $\mathbf{c}_t$ (and $\mathbf{x}_t$) are identifiable up to an invertible transformation from a finite window of observations.

**Illustration for $n=2$.** When $n=2$, block–wise separations allow identification of the joint variables $[\mathbf{c}_t, \mathbf{c}_{t+1}, \mathbf{x}_t, \mathbf{x}_{t+1}]$ from $\mathbf{x}_{t-4:t+3}$ (length $8+1$). Knowing the temporal direction disambiguates $[\mathbf{c}_t, \mathbf{x}_t]$ from $[\mathbf{c}_{t+1}, \mathbf{x}_{t+1}]$. Because $\mathbf{x}_t$ is observed, we obtain $\mathbf{c}_t = h(\mathbf{x}_{t-4:t+3})$ for some invertible $h$, and thus the contemporaneous pair $[\mathbf{c}_t, \mathbf{x}_t]$ is identified.

**Connection to delayed/cumulative rewards.** Delayed and cumulative effects fit naturally in the $n$-order view. For a delay $\ell$,
$$\mathbf{r}_{t+\ell} = \rho(\mathbf{x}_{t+\ell}, \mathbf{a}_{t+\ell}, \mathbf{c}_t) \quad \text{(delayed effect)},$$
while cumulative influence over a horizon $L$ can be written as
$$\mathbf{r}_{t+k} = \rho_k(\mathbf{x}_{t+k}, \mathbf{a}_{t+k}, \mathbf{c}_t), \qquad k = 0, \dots, L-1,$$
both of which are encompassed by the $n$-order Markov factorization above. Our cheetah variants instantiate these with, e.g., $r_{t+\ell} = -\|\mathbf{v}_{t+\ell} - \mathbf{c}_t\|^2$ (delayed) and $r_{t+k} = -\|\mathbf{v}_{t+k} - \mathbf{c}_t\|^2$ (cumulative), where $\mathbf{v}_t$ denotes speed; the identification results remain valid.

**Results.** We evaluate identification under delayed and cumulative latent effects in the Cheetah environment using observation windows of length $6, 8, 10$, and $20$. In all cases, linear probes recover the latent with high accuracy, and performance improves monotonically with longer context. For *delayed* effects, probing accuracy rises from $0.81$ to $0.91$ and $R^2$ from $0.72$ to $0.86$ as block size increases from 6 to 20. For *cumulative* effects, probing accuracy increases from $0.84$ to $0.93$ and $R^2$ from $0.75$ to $0.88$ over the same range. These results confirm that (i) the latent $c_t$ is behaviorally consequential in non–first-order settings and (ii) moderate temporal context suffices for accurate recovery, supporting our relaxed $n$-order Markov analysis.

### B.3.5 Cases in Assumption 2.

The assumption of the injectivity of a linear operator is commonly employed in the nonparametric identification (Hu & Schennach, 2008; Carroll et al., 2010; Hu & Shum, 2012). Intuitively, it means that different input distributions of a linear operator correspond to different output distributions of that operator. For a better understanding, we provide several examples in Fu et al. (2025) that describe the mapping from $p_a \Rightarrow p_b$, where $a$ and $b$ are random variables:

**Example 1** (Invertible). $b = g(a)$, *where $g$ is an invertible function.*

---

[4]Any window of length at least $4n+1$ suffices; we state one concrete choice for clarity.

**Example 2** (Additive). *$b = a + \epsilon$, where $p(\epsilon)$ must not vanish everywhere after the Fourier transform.*

**Example 3** (Nonlinear Additive). *$b = g(a) + \epsilon$, where conditions from **Examples** 1-2 are required.*

**Example 4** (Post-nonlinear). *$b = g_1(g_2(a) + \epsilon)$, a post-nonlinear model with invertible nonlinear functions $g_1, g_2$, combining the assumptions in **Examples** 1-3.*

**Example 5** (Nonlinear with Exponential Family). *$b = g(a, \epsilon)$, where the joint distribution $p(a, b)$ follows an exponential family.*

**Example 6** (Nonparametric). *$b = g(a, \epsilon)$, a general nonlinear formulation. Certain deviations from the nonlinear additive model (**Example** 3), e.g., polynomial perturbations, can still be tractable.*

### B.4    ELBO

In this section, we provide analysis on the $\mathbf{x}^0$-prediction Mean Squared Error (MSE) loss objectives used in the Denoise-and-Refine Mechanism of `Ada-Diffuser`. Our main argument establishes that minimizing the reconstruction losses $\mathcal{L}_{\text{prior}}$ and $\mathcal{L}_{\text{post}}$ corresponds to optimizing an ELBO on the conditional log-likelihood of the clean observation $\mathbf{x}_t^0$, given a noisy observation $\mathbf{x}_t^k$ and an inferred latent context $\mathbf{c}_t$.

Let $\mathbf{x}_t^0 \sim q(\mathbf{x}_t^0)$ be a clean data sample from the true data distribution at sequence time step $t$. Let $\mathbf{c}_t$ be the inferred latent context relevant to $\mathbf{x}_t^0$.

The forward diffusion process gradually adds Gaussian noise to $\mathbf{x}_t^0$ over $K$ diffusion steps:

$$q(\mathbf{x}_t^k|\mathbf{x}_t^{k-1}) = \mathcal{N}(\mathbf{x}_t^k; \sqrt{\alpha_k}\mathbf{x}_t^{k-1}, (1 - \alpha_k)\mathbf{I})$$

for $k \in \{1, ..., K\}$, where $\alpha_k \in (0, 1)$ are predefined noise schedule parameters. This process allows sampling $\mathbf{x}_t^k$ directly from $\mathbf{x}_t^0$:

$$\mathbf{x}_t^k = \sqrt{\bar{\alpha}_k}\mathbf{x}_t^0 + \sqrt{1 - \bar{\alpha}_k}\boldsymbol{\epsilon}, \quad \text{where } \boldsymbol{\epsilon} \sim \mathcal{N}(0, \mathbf{I}), \text{ and } \bar{\alpha}_k = \prod_{i=1}^{k}\alpha_i.$$

The reverse process $p_\theta(\mathbf{x}_t^{k-1}|\mathbf{x}_t^k, \mathbf{c}_t)$ that parameterized by $\theta$ aims to denoise $\mathbf{x}_t^k$ to $\mathbf{x}_t^{k-1}$ conditioned on $\mathbf{c}_t$.

The derivation of the ELBO for diffusion models is standard following DDPM related derivations (Ho et al., 2020; Chen et al., 2024). The conditional log-likelihood $\log p_\theta(\mathbf{x}_t^0|\mathbf{c}_t)$ can be lower-bounded using the ELBO:

$$\log p_\theta(\mathbf{x}_t^0|\mathbf{c}_t) \geq \mathbb{E}_{q(\mathbf{x}_t^{1:K}|\mathbf{x}_t^0)}\left[\log p_\theta(\mathbf{x}_t^K|\mathbf{c}_t) + \sum_{k=1}^{K}\log \frac{p_\theta(\mathbf{x}_t^{k-1}|\mathbf{x}_t^k, \mathbf{c}_t)}{q(\mathbf{x}_t^{k-1}|\mathbf{x}_t^k, \mathbf{x}_t^0)}\right]$$

Assuming $p_\theta$ satisfies Markov Property (i.e., $p_\theta(\mathbf{x}_t^{k-1}|\mathbf{x}_t^k, \ldots, \mathbf{x}_t^K, \mathbf{c}_t) = p_\theta(\mathbf{x}_t^{k-1}|\mathbf{x}_t^k, \mathbf{c}_t))$, which is a standard structural assumption for diffusion models, the ELBO can be rewritten as:

$$\log p_\theta(\mathbf{x}_t^0|\mathbf{c}_t) \geq \underbrace{\mathbb{E}_{q(\mathbf{x}_t^1|\mathbf{x}_t^0)}[\log p_\theta(\mathbf{x}_t^0|\mathbf{x}_t^1, \mathbf{c}_t)]}_{L_0}$$
$$- \sum_{k=2}^{K}\underbrace{\mathbb{E}_{q(\mathbf{x}_t^k|\mathbf{x}_t^0)}[D_{KL}(q(\mathbf{x}_t^{k-1}|\mathbf{x}_t^k, \mathbf{x}_t^0)||p_\theta(\mathbf{x}_t^{k-1}|\mathbf{x}_t^k, \mathbf{c}_t))]}_{L_{k-1}}$$
$$- \underbrace{D_{KL}(q(\mathbf{x}_t^K|\mathbf{x}_t^0)||p_\theta(\mathbf{x}_t^K|\mathbf{c}_t))}_{L_K},$$

This inequality holds with equality if and only if the model's true posterior over the latent diffusion path, $p_\theta(\mathbf{x}_t^{1:K}|\mathbf{x}_t^0, \mathbf{c}_t)$, is identical to the approximate posterior used to derive the ELBO, which is the forward noising process $q(\mathbf{x}_t^{1:K}|\mathbf{x}_t^0)$. This bound can also include an additive constant $C(\mathbf{x}_t^0, \mathbf{c}_t)$ which does not depend on the model parameters $\theta$ and is thus typically omitted when focusing on terms relevant to parameter optimization.

To maximize $\log p_\theta(\mathbf{x}_t^0|\mathbf{c}_t)$, we aim to maximize this lower bound by optimizing $L_0$ (i.e., maximizing this term) and each $L_{k-1}$ term (i.e., minimizing these $D_{KL}$ terms, as they appear with a negative

sign). The term $L_K$ is often treated as a constant (or absorbed into $C(\mathbf{x}_t^0, \mathbf{c}_t)$) if $p_\theta(\mathbf{x}_t^K|\mathbf{c}_t)$ is set to a standard Gaussian $\mathcal{N}(0, \mathbf{I})$ and $\bar{\alpha}_K \approx 0$.

We parameterize the reverse process $p_\theta(\mathbf{x}_t^{k-1}|\mathbf{x}_t^k, \mathbf{c}_t)$ as a Gaussian:

$$p_\theta(\mathbf{x}_t^{k-1}|\mathbf{x}_t^k, \mathbf{c}_t) = \mathcal{N}(\mathbf{x}_t^{k-1}; \boldsymbol{\mu}_\theta(\mathbf{x}_t^k, k, \mathbf{c}_t), \sigma_k^2 \mathbf{I})$$

The true posterior step $q(\mathbf{x}_t^{k-1}|\mathbf{x}_t^k, \mathbf{x}_t^0)$ is also Gaussian:

$$q(\mathbf{x}_t^{k-1}|\mathbf{x}_t^k, \mathbf{x}_t^0) = \mathcal{N}(\mathbf{x}_t^{k-1}; \tilde{\boldsymbol{\mu}}_k(\mathbf{x}_t^k, \mathbf{x}_t^0), \tilde{\sigma}_k^2 \mathbf{I})$$

where $\tilde{\boldsymbol{\mu}}_k(\mathbf{x}_t^k, \mathbf{x}_t^0) = \frac{\sqrt{\bar{\alpha}_{k-1}}(1-\alpha_k)}{1-\bar{\alpha}_k}\mathbf{x}_t^0 + \frac{\sqrt{\alpha_k}(1-\bar{\alpha}_{k-1})}{1-\bar{\alpha}_k}\mathbf{x}_t^k$ and $\tilde{\sigma}_k^2 = \frac{1-\bar{\alpha}_{k-1}}{1-\bar{\alpha}_k}(1-\alpha_k)$ is the variance.

For an $\mathbf{x}^0$-prediction model, denoted as $\epsilon_\theta(\mathbf{x}_t^k, k, \mathbf{c}_t)$ in the main paper, that aims to predict $\mathbf{x}_t^0$ from the noisy input $\mathbf{x}_t^k$ and context $\mathbf{c}_t$, the mean of the reverse model $\boldsymbol{\mu}_\theta$ can be expressed as:

$$\boldsymbol{\mu}_\theta(\mathbf{x}_t^k, k, \mathbf{c}_t) = \frac{\sqrt{\bar{\alpha}_{k-1}}(1-\alpha_k)}{1-\bar{\alpha}_k}\epsilon_\theta(\mathbf{x}_t^k, k, \mathbf{c}_t) + \frac{\sqrt{\alpha_k}(1-\bar{\alpha}_{k-1})}{1-\bar{\alpha}_k}\mathbf{x}_t^k$$

Choosing $\sigma_k^2 = \tilde{\sigma}_k^2$, the KL divergence term $L_{k-1}$ simplifies to:

$$L_{k-1} = \mathbb{E}_{q(\mathbf{x}_t^k|\mathbf{x}_t^0)}\left[\frac{1}{2\sigma_k^2}\left\|\tilde{\boldsymbol{\mu}}_k(\mathbf{x}_t^k, \mathbf{x}_t^0) - \boldsymbol{\mu}_\theta(\mathbf{x}_t^k, k, \mathbf{c}_t)\right\|^2\right] + C'_k$$

$$= \mathbb{E}_{\mathbf{x}_t^0, \boldsymbol{\epsilon}}\left[\frac{1}{2\sigma_k^2}\left(\frac{\sqrt{\bar{\alpha}_{k-1}}(1-\alpha_k)}{1-\bar{\alpha}_k}\right)^2\left\|\mathbf{x}_t^0 - \epsilon_\theta(\sqrt{\bar{\alpha}_k}\mathbf{x}_t^0 + \sqrt{1-\bar{\alpha}_k}\boldsymbol{\epsilon}, k, \mathbf{c}_t)\right\|^2\right] + C'_k$$

where $C'_k$ are constants not depending on $\theta$. The expectation $\mathbb{E}_{\mathbf{x}_t^0, \boldsymbol{\epsilon}}$ denotes averaging over clean data $\mathbf{x}_t^0$ and the noise $\boldsymbol{\epsilon}$ used to construct $\mathbf{x}_t^k$. Thus, maximizing the ELBO contribution from $-L_{k-1}$ is equivalent to minimizing the following weighted MSE term:

$$\mathbb{E}_{\mathbf{x}_t^0, \boldsymbol{\epsilon}, \mathbf{c}_t}\left[w(k)\left\|\mathbf{x}_t^0 - \epsilon_\theta(\sqrt{\bar{\alpha}_k}\mathbf{x}_t^0 + \sqrt{1-\bar{\alpha}_k}\boldsymbol{\epsilon}, k, \mathbf{c}_t)\right\|^2\right] \tag{A32}$$

where $w(k) = \frac{1}{2\sigma_k^2}\left(\frac{\sqrt{\bar{\alpha}_{k-1}}(1-\alpha_k)}{1-\bar{\alpha}_k}\right)^2$ is a positive weighting factor.

The term $L_0 = \mathbb{E}_{q(\mathbf{x}_t^1|\mathbf{x}_t^0)}[\log p_\theta(\mathbf{x}_t^0|\mathbf{x}_t^1, \mathbf{c}_t)]$ can also be made proportional to an MSE if $p_\theta(\mathbf{x}_t^0|\mathbf{x}_t^1, \mathbf{c}_t)$ is a Gaussian centered at $\epsilon_\theta(\mathbf{x}_t^1, 1, \mathbf{c}_t)$:

$$\log p_\theta(\mathbf{x}_t^0|\mathbf{x}_t^1, \mathbf{c}_t) = -\frac{1}{2\sigma_1^2}\left\|\mathbf{x}_t^0 - \epsilon_\theta(\mathbf{x}_t^1, 1, \mathbf{c}_t)\right\|^2 + \text{const}$$

Maximizing $L_0$ is then equivalent to minimizing this MSE.

The diffusion model $\epsilon_\theta$ is typically trained by minimizing a simplified objective (e.g., (Ho et al., 2020)), often an unweighted or equally weighted sum of these MSE terms over uniformly sampled diffusion steps $k \in [1, K]$ and data $\mathbf{x}_t^0$:

$$\mathcal{L}_{\text{simple}}(\theta) = \mathbb{E}_{k \sim U[1,K], \mathbf{x}_t^0, \boldsymbol{\epsilon}, \mathbf{c}_t}\left[\left\|\mathbf{x}_t^0 - \epsilon_\theta(\sqrt{\bar{\alpha}_k}\mathbf{x}_t^0 + \sqrt{1-\bar{\alpha}_k}\boldsymbol{\epsilon}, k, \mathbf{c}_t)\right\|^2\right]$$

This simplification is justified by arguing that reweighting terms $w(k)$ in Equation A32 can be absorbed into the network or do not significantly alter the optimal solution for expressive models, allowing $w(k)$ to be effectively set to 1.

The Denoise-and-Refine losses are:

$$\mathcal{L}_{\text{prior}} = \mathbb{E}_{\mathbf{x}_t^0, \boldsymbol{\epsilon}, \hat{\mathbf{c}}_t^{\text{prior}}}\left[\left\|\mathbf{x}_t^0 - \epsilon_\theta(\mathbf{x}_t^{k_i}, k_i, \hat{\mathbf{c}}_t^{\text{prior}})\right\|^2\right]$$

$$\mathcal{L}_{\text{post}} = \mathbb{E}_{\mathbf{x}_t^0, \boldsymbol{\epsilon}, \hat{\mathbf{c}}_t^{\text{post}}}\left[\left\|\mathbf{x}_t^0 - \epsilon_\theta(\mathbf{x}_t^{k_i}, k_i, \hat{\mathbf{c}}_t^{\text{post}})\right\|^2\right]$$

where $\mathbf{x}_t^{k_i} = \sqrt{\bar{\alpha}_{k_i}}\mathbf{x}_t^0 + \sqrt{1-\bar{\alpha}_{k_i}}\boldsymbol{\epsilon}$, and $k_i$ is the specific input noise level for the observation $\mathbf{x}_t$ determined by the causal denoising schedule $k_i = \frac{i}{T}K$. These losses, $\mathcal{L}_{\text{prior}}$ and $\mathcal{L}_{\text{post}}$, are specific

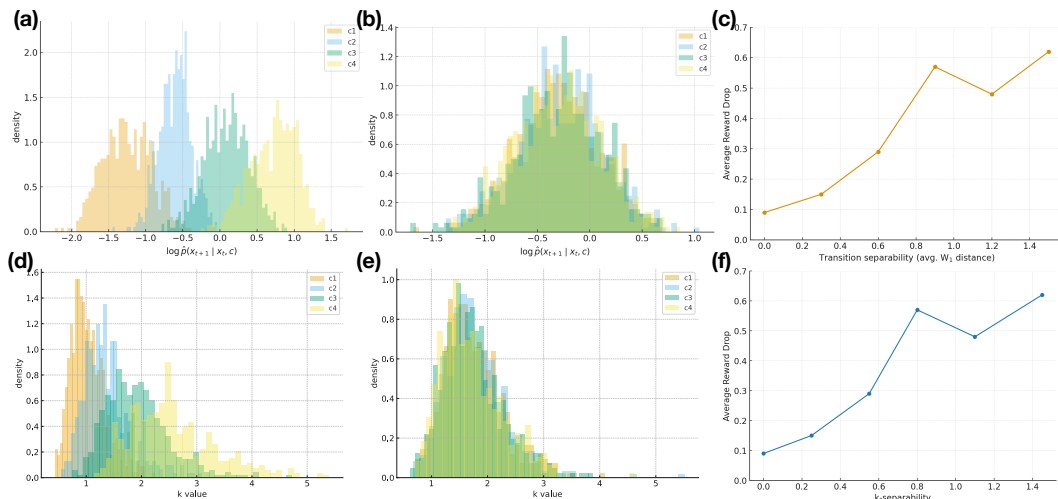

Figure A1: **Verification of the assumptions**. (a) Transition separability in Cheetah under the hyperparameter setting $(m, n) = (5, 0.5)$. (b) Transition separability under a weak–context setting $(m, n) = (0.2, 0.2)$, where the context barely affects the dynamics. (c) Average reward drop when planning *with* vs. *without* conditioning on $c$, plotted against the transition separability. (d) $k$–distributions for $(5, 0.5)$, (e) $k$–distributions for $(0.2, 0.2)$, (f) reward drop versus $k$–separability.

instances of the simplified MSE loss objective in equation A32 with $w(k_i) \approx 1$, conditioned on the inferred contexts $\hat{\mathbf{c}}_t^{\text{prior}}$ and $\hat{\mathbf{c}}_t^{\text{post}}$ respectively. Consequently, minimizing these MSE losses directly optimizes the corresponding terms in the ELBO for $\log p_\theta(\mathbf{x}_t^0 | \mathbf{c}_t)$.

Therefore, we have proven that minimizing $\mathcal{L}_{\text{prior}}$ and $\mathcal{L}_{\text{post}}$ as defined in the Denoise-and-Refine mechanism serves to maximize a variational lower bound on the conditional log-likelihood $\log p_\theta(\mathbf{x}_t^0 | \mathbf{c}_t)$. The underlying diffusion model $\epsilon_\theta(\cdot, k, \cdot)$ is trained to be proficient at denoising from a range of noise levels $k$, as captured by objectives such as $\mathcal{L}_{\text{simple}}$. The specific monotonically increasing noise schedule $k_i$ used in $\mathcal{L}_{\text{prior}}$ and $\mathcal{L}_{\text{post}}$ represents a particular instance from this range of noise levels. Thus, these objectives are theoretically grounded in the principles of variational inference for diffusion models, adapted to conditioning on the inferred latent context $\mathbf{c}_t$ and applied at specific noise levels relevant to the autoregressive denoising process of Ada-Diffuser.

## B.5 ASSUMPTION VERIFICATION

Here, we test whether Assumption 2 and Assumption 3 hold in practice, explain why we view them as mild, and, importantly, analyze what happens when they fail. We use the Cheetah environment, where the latent context corresponds to a time-varying wind speed $f_w = 5 + m \sin(nt)$ that perturbs the agent's dynamics. We sweep over combinations of $(m, n)$ to address two questions: (1) whether the setting used in the paper, $(m.n) = (5, 0.5)$, indeed satisfies these assumptions; and (2) how violations of the assumptions affect our method and the associated analyses.

### B.5.1 ABOUT ASSUMPTION 2

To evaluate Assumption 2, which requires that the conditional dynamics $P(\mathbf{x}_{t+1} \mid \mathbf{x}_t, \mathbf{c}_t)$ be injective in the context variable $\mathbf{c}_t$, we perform an empirical test to determine whether different contexts induce measurably different transition dynamics. Given a fitted probabilistic dynamics model $\hat{p}(\mathbf{x}_{t+1} \mid \mathbf{x}_t, \mathbf{c}_t)$, we estimate the distribution of next states under each context $\mathbf{c} \in \{\mathbf{c}_1, \ldots, \mathbf{c}_M\}$ by drawing samples from the replay buffer and computing $\hat{p}(\mathbf{x}_{t+1} \mid \mathbf{x}_t, \mathbf{c})$. For every pair of contexts $(\mathbf{c}_i, \mathbf{c}_j)$, we quantify the difference between their induced transition distributions using the 1-Wasserstein distance:

$$\text{Inj}(\mathbf{c}_i, \mathbf{c}_j) = W_1\big(p(\mathbf{x}_{t+1} \mid \mathbf{x}_t, \mathbf{c}_i), p(\mathbf{x}_{t+1} \mid \mathbf{x}_t, \mathbf{c}_j)\big). \quad (A33)$$

Large values of $\text{Inj}(\mathbf{c}_i, \mathbf{c}_j)$ indicate that distinct contexts lead to distinct transition kernels, consistent with injectivity, while values near zero suggest that different contexts produce nearly indistinguishable

dynamics. When $(m, n) = (5, 0.5)$ (Fig. A1(a)), we observe consistently non-zero Wasserstein distances across contexts, indicating that $P(\mathbf{x}_{t+1} \mid \mathbf{x}_t, \mathbf{c}_t)$ is context-injective in the regime studied. In contrast, when we reduce the context variation $(m, n) = (0.2, 0.2)$ (Fig. A1(b)), the distances are toward zero, showing the failure mode of the assumption. This shows that Assumption 2 is mild in the latent-aware decision-making.

### B.5.2 About Assumption 3

We provide an empirical test of the spectral ratio $k$ to examine Assumption 3 in the RL setting. Using the dynamics model on Cheetah $\hat{p}(\mathbf{x}_t \mid \mathbf{x}_{t-1}, \mathbf{c}_t)$, we compute

$$k(\mathbf{x}_t, \bar{\mathbf{x}}_t, \mathbf{x}_{t-1}, \bar{\mathbf{x}}_{t-1}, \mathbf{c}_t) = \frac{\hat{p}(\mathbf{x}_t \mid \mathbf{x}_{t-1}, \mathbf{c}_t) \, \hat{p}(\bar{\mathbf{x}}_t \mid \bar{\mathbf{x}}_{t-1}, \mathbf{c}_t)}{\hat{p}(\bar{\mathbf{x}}_t \mid \mathbf{x}_{t-1}, \mathbf{c}_t) \, \hat{p}(\mathbf{x}_t \mid \bar{\mathbf{x}}_{t-1}, \mathbf{c}_t)}. \tag{A34}$$

We draw transitions $(\mathbf{x}_{t-1}, \mathbf{x}_t)$ from the replay buffer (600 samples) and form cross-paired transitions $(\bar{\mathbf{x}}_{t-1}, \bar{\mathbf{x}}_t)$ by swapping endpoints across trajectories. For each context $\mathbf{c} \in \{\mathbf{c}_1, \ldots, \mathbf{c}_M\}$, this yields an empirical distribution of $k(\cdot\,; c)$. We then quantify how well $k$ separates contexts using the 1-Wasserstein distance between pairs of $k$-distributions, i.e.,

$$\mathrm{Sep}(\mathbf{c}_i, \mathbf{c}_j) = W_1\big(p(k \mid \mathbf{c}_i),\, p(k \mid \mathbf{c}_j)\big). \tag{A35}$$

When Assumption 3 holds, $k$ remains bounded and its distribution varies across contexts. Empirically, we observe clear multi-modal separation across contexts in the paper's setting $((m, n) = (5, 0.5)$, Fig. A1(d)), whereas in regimes where the dynamics become less context-dependent, the $k$-distributions overlap heavily.

When $k$ is not distinguishable across contexts $c$ $((m, n) = (0.2, 0.2)$, Fig A1(e)), it implies that $c$ does not exert a noticeable effect on the transition dynamics. In this regime, explicitly modeling the context is unnecessary, since the environment effectively behaves as a single-context system. Hence, we believe Assumption 3 is mild in our main regime and also clarifies the failure mode when it is violated.

### B.5.3 Policy Learning under Different Separability

When the conditional transition $P(x_{t+1} \mid x_t, c)$ is not injective in $c$ or $k$ is nearly the same for different $c$, different contexts induce nearly identical transition kernels. This means the context is not identifiable from the dynamics and does not meaningfully alter the environment; in such cases, explicitly modeling $c$ brings little benefit for policy learning. Figures A1(c) and (f) illustrate this effect. We vary $(m, n)$ to change the strength of the latent wind context, and compare policy performance when planning with the ground-truth context $c$ versus ignoring $c$. We then plot the resulting performance gap (reward-drop ratio) against transition separability and $k$–separability. The gap shrinks when separability is small, indicating that when both the transition and $k$ are weakly context-dependent, modeling $c$ is unnecessary. Overall, these results verify that Assumption 2 and Assumption 3 are not only mild in our setting, but also clarify why modeling the latent context is important precisely in regimes where the dynamics are strongly context-dependent.

## C Summary on Different MDPs

Our work considers a contextual POMDP setting with an evolving latent process, which naturally relates to several established MDP formulations, including contextual MDPs (Hallak et al., 2015), hidden-parameter MDPs (HiP-MDPs) (Doshi-Velez & Konidaris, 2016), and their variants. In this section, we provide formal definitions of these models and discuss their relationships and distinctions.

### C.1 Contexutal MDPs

A contextual Markov decision process (CMDP) (Hallak et al., 2015) is defined by the tuple $\langle \mathcal{C}, \mathcal{S}, \mathcal{A}, \mathcal{M} \rangle$, where $\mathcal{C}$ is the context space, $\mathcal{S}$ is the state space, and $\mathcal{A}$ is the action space. The mapping $\mathcal{M}$ assigns to each context $c \in \mathcal{C}$ a set of MDP parameters $\mathcal{M}(c) = \{R^c, T^c\}$, where $R^c$ and $T^c$ are the reward and transition functions associated with context $c$.

Sodhani et al. (2021) and Liang et al. (2024a) extend the CMDP framework to settings in which the context variable $\mathbf{c}$ evolves according to its own Markovian dynamics $p(\mathbf{c}_{t+1} \mid \mathbf{c}_t)$, closely aligning with our formulation of a latent process evolving over time.

## C.2   HIDDEN-PARAMETER MDPS

Hidden-Parameter MDPs (HiP-MDPs) (Doshi-Velez & Konidaris, 2016) are defined by the tuple $\mathcal{M} = \langle \mathcal{S}, \mathcal{A}, \Theta, \mathcal{T}, \mathcal{R}, \gamma, P_\Theta \rangle$, where $\mathcal{S}$ is the state space, $\mathcal{A}$ is the action space, and $\Theta$ is the space of task-specific latent parameters. For each $\theta \in \Theta$, the transition and reward functions are given by $\mathcal{T}_\theta : \mathcal{S} \times \mathcal{A} \rightarrow \mathcal{P}(\mathcal{S})$ and $\mathcal{R}_\theta : \mathcal{S} \times \mathcal{A} \rightarrow \mathbb{R}$, respectively. The parameter $\theta$ is sampled from a prior distribution $P_\Theta$ at the beginning of an episode and remains fixed during the episode. The discount factor is denoted by $\gamma \in [0, 1)$. This framework defines a family of MDPs indexed by the latent parameter $\theta$, with each $\theta$ inducing a different set of dynamics and reward functions. It can be seen as a special case of a contextual MDP where the context is latent and fixed per episode. Xie et al. (2021) further generalize this framework by allowing the task parameter $\theta$ to evolve dynamically across episodes, rather than being fixed.

Bayes-Adaptive MDPs (BAMDPs) are closely related to both HiP-MDPs and contextual MDPs (CMDPs). In BAMDPs, the agent maintains a posterior distribution over MDPs based on its interaction history. Specifically, it maintains a belief $b_t(R, T) = p(R, T \mid \boldsymbol{\tau}_{:t})$, where $\tau_{:t} = \{\mathbf{s}_0, \mathbf{a}_0, r_0, \ldots, \mathbf{s}_t\}$ denotes the trajectory observed up to time $t$. This belief captures the agent's uncertainty about the underlying transition and reward functions.

The transition and reward functions can then be defined in expectation over this posterior, effectively conditioning decision-making on the belief $b_t$. When the environment is driven by hidden contextual variables or latent task parameters, such as in CMDPs or HiP-MDPs—this belief can be interpreted as a distribution over these latent variables. In this view, BAMDPs provide a non-parametric framework for reasoning over hidden structure, while approaches like ours explicitly model such latent variables and infer their posterior distributions using amortized inference. Both aim to enable adaptive planning and learning under uncertainty, but differ in how latent structure is represented and inferred.

## C.3   DISCUSSIONS AND COMPARISONS

The key distinction between contextual MDPs and hidden-parameter MDPs lies in how the latent factors are represented: contextual MDPs explicitly treat them as latent variables, while HiP-MDPs model them implicitly as parameters governing the transition and reward functions. In our work, we adopt the contextual MDP perspective, where the latent process is modeled as a random variable that evolves over time.

However, our identification theory, focused on recovering the posterior distribution over latent variables, also applies to the HiP-MDP setting. Once the posterior over the hidden parameters is identified, the corresponding transition and reward functions can be recovered as well.

Additionally, our framework, which models a factorization over observed states and latent variables, is conceptually related to factored MDPs (Guestrin et al., 2003). In a factored MDP, the state space $\mathcal{S}$ is represented as a set of variables $\mathcal{S} = \{s^{(1)}, s^{(2)}, \ldots, s^{(n)}\}$, and the transition and reward functions are decomposed over these factors:

$$T(\mathbf{s}' \mid \mathbf{s}, \mathbf{a}) = \prod_{i=1}^{n} T_i \left( s'^{(i)} \mid \mathrm{Pa}_T^{(i)}(\mathbf{s}, \mathbf{a}) \right), \quad R(\mathbf{s}, \mathbf{a}) = \sum_{j=1}^{m} R_j \left( \mathrm{Pa}_R^{(j)}(\mathbf{s}, \mathbf{a}) \right),$$

where $\mathrm{Pa}_T^{(i)}$ and $\mathrm{Pa}_R^{(j)}$ denote the parent variables (i.e., dependencies) for each transition and reward component, respectively. our framework, while not relying on an explicit graphical structure, shares conceptual similarities with factored MDPs (Guestrin et al., 2003) through its coarse-grained factorization over observed states and latent variables. Specifically, we distinguish between latent variables that affect the transition dynamics and those that affect the reward function. Formally, we express the generative process as:

$$T(\mathbf{s}_{t+1} \mid \mathbf{s}_t, \mathbf{a}_t, \mathbf{c}_t^{\mathrm{s}}), \quad R(r_t \mid \mathbf{s}_t, \mathbf{a}_t, \mathbf{c}_t^{\mathrm{r}}),$$

where $\mathbf{c}_t^{\mathrm{s}}$ and $\mathbf{c}_t^{\mathrm{r}}$ are distinct (or potentially overlapping) latent factors that influence transitions and rewards, respectively. This separation enables flexible modeling of partially observable environments where different unobserved processes govern the dynamics and task objectives.

# D   DETAILS ON ADA-DIFFUSER

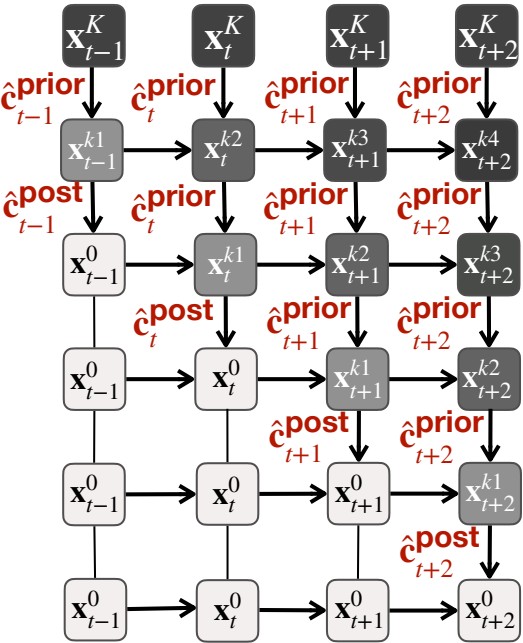

Figure A2: An illustration of the zig-zag sampling process with a block of 4 time steps. ↓ and | indicate denoising and identity mapping, respectively.

## D.1   FULL ALGORITHM AND RESULTS

Our framework consists of two stages: latent factor identification and diffusion-based planning or policy learning. Below, we provide the algorithmic pseudocode for both stages. Specifically, Algorithm A1 describes Stage 1: latent factor identification, while Algorithms A2 and A3 correspond to Ada-Diffuser-Planner and Ada-Diffuser-Policy, respectively.

For clarity, we omit the detailed step-by-step procedures for denoise-and-refine and zig-zag sampling (Lines 7–8, 11, and 19–22 in Algorithm A2; Lines 6–7 and 13 in Algorithm A3), as these are fully described in Section 4.3. For Ada-Diffuser-Policy, we show a Diffusion Policy (DP)-based algorithm, which provides a general framework for multi-step action generation. In the IDQL-based variant, both the action execution horizon and observation horizon are set to 1, corresponding to single-step policy inference conditioned only on the current observation.

---

**Algorithm A1:** Latent Factor Identification.

---

1: **Input:** offline dataset $\mathcal{D}$
2: Randomly initialize decoder $p_\theta(\mathbf{s}_{t+1}, r_t \mid \mathbf{s}, \mathbf{a}, \mathbf{c})$,
   encoder $q_\psi(\mathbf{c}_t \mid \mathbf{s}_{t-T_x:t+1}, \mathbf{a}_{t-T_x:t+1}, r_{t-T_x:t+1})$ and prior network $p_\phi(\mathbf{c}_t \mid \mathbf{c}_{t-1})$,
3: **while** not done **do**
4:     Sample batches of trajectories from $\mathcal{D}$
5:     Compute ELBO and update $\theta, \psi, \phi$
6: **end while**

---

Additionally, we provide the full results for all experiments: Table A3 reports results for the action-free setting; Tables A4 and A5 present results for environments with latent factors affecting dynamics and rewards; and Tables A6, A7, A8, and A9 summarize results for environments without explicitly modeled latent factors.

---

**Algorithm A2:** `Ada-Diffuser-Planner`.

---

1: **Input:** `Env`, offline dataset $\mathcal{D}$, pre-trained encoder $q_\psi$ and prior network $p_\phi$
   observation horizon $T_o$, planning horizon $T_p$, action execution horizon $T_a$, condition $\mathbf{y}$
   // **Training**
2: Initialize noise predictor $\epsilon_\theta$, inverse dynamics model $f_\phi$
3: **while** not done **do**
4:      Sample $\mathbf{x}_{t-T_o:t+T_p}$ from $\mathcal{D}$
5:      Sample $\hat{\mathbf{c}}^{\text{prior}}_{t:t+T_p}$ and $\hat{\mathbf{c}}^{\text{post}}_{t:t+T_p-2}$ from $p_\phi$ and $q_\psi$
6:      **if** using inverse dynamics model **then**
7:          Train Causal Diffusion Model (noise predictor $\epsilon_\theta$) with $\mathbf{x}_{t-T_o:t}$, $\hat{\mathbf{c}}^{\text{prior}}_{t-T_o:t}$, and $\hat{\mathbf{c}}^{\text{post}}_{t-T_o:t}$ and
            other conditions $\mathbf{y}$, target outputs are $\mathbf{s}_{t+1:t+T_p}$
8:          Train encoder $q_\psi$ with the contrastive improvement loss $\mathcal{L}_{\text{contrast}}$
9:          Train Inverse Dynamics Model $f_\phi$ to generate actions $\mathbf{a}_{t+1:t+T_p}$
10:      **else**
11:          Train Causal Diffusion Model (noise predictor $\epsilon_\theta$) with $\mathbf{x}_{t-T_o:t}$, $\hat{\mathbf{c}}^{\text{prior}}_{t-T_o:t}$, and $\hat{\mathbf{c}}^{\text{post}}_{t-T_o:t}$ and
            other conditions $\mathbf{y}$, target outputs are $\{\mathbf{s}_{t+1:t+T_p}, \mathbf{a}_{t+1:t+T_p}\}$
12:          Train encoder $q_\psi$ with the contrastive improvement loss $\mathcal{L}_{\text{contrast}}$
13:      **end if**
14: **end while**
   // **Execution**
15: Initialize environment: $s_0 \sim$ `Env.reset()`, set $t \leftarrow 0$
16: **while** not done **do**
        // Observe and infer latent factors
17:      Observe recent trajectory $\mathbf{x}_{t-T_o:t}$
18:      Sample latent variables $\hat{\mathbf{c}}^{\text{prior}}_{t:t+T_p}$ from $p_\phi$
        // Generate candidate trajectory
19:      **if** using inverse dynamics model **then**
20:          Generate future states (zig-zag sampling) $\hat{\mathbf{s}}_{t+1:t+T_p}$ conditioned on $\mathbf{x}_{t-T_o:t}$, $\hat{\mathbf{c}}^{\text{prior}}_{t:t+T_p}$, and $\mathbf{y}$
            via learned noise predictor $\epsilon_\theta$
21:          Generate actions $\hat{\mathbf{a}}_{t+1:t+T_p} \leftarrow f_\phi(\hat{\mathbf{s}}_{t+1:t+T_p}, \hat{\mathbf{s}}_{t:t+T_p-1})$
22:      **else**
23:          Generate future trajectory $\{\hat{\mathbf{s}}_{t+1:t+T_p}, \hat{\mathbf{a}}_{t+1:t+T_p}\}$ conditioned on $\mathbf{x}_{t-T_o:t}$, $\hat{\mathbf{c}}^{\text{prior}}_{t:t+T_p}$, and $\mathbf{y}$
            via learned noise predictor $\epsilon_\theta$
24:      **end if**
        // Execute action(s) in environment
25:      **for** each step $i = 1$ to $T_a$ **do**
26:          Execute $\hat{\mathbf{a}}_{t+i}$ in `Env`, observe $s_{t+i+1}, r_{t+i}$
27:          Append $(s_{t+i}, \hat{a}_{t+i}, r_{t+i})$ to trajectory buffer
28:      **end for**
29:      Update $t \leftarrow t + T_a$
30: **end while**

---

## D.2 Architecture Choices and Hyper-parameters

We detail the architectural design choices and hyperparameter settings used for model components, loss functions, and training procedures across all `Ada-Diffuser` variants under different environments and benchmarks.

### D.2.1 Latent Factor Identification

**Architectures** We use a variational autoencoder (VAE) (Kingma & Welling, 2014) to optimize the evidence lower bound (ELBO). The same architectural design is used across all variants of `Ada-Diffuser` and all benchmark settings.

For the encoder, we first embed states, actions, and rewards using separate MLPs with ReLU activations. The resulting embeddings are concatenated and passed through a two-layer MLP (each

---

**Algorithm A3:** `Ada-Diffuser-Policy (DP-based)`

---

1: **Input:** `Env`, offline dataset $\mathcal{D}$, pre-trained encoder $q_\psi$ and prior network $p_\phi$
 observation horizon $T_o$, action generation horizon $T_p$, action execution horizon $T_a$, condition $\mathbf{y}$
 **// Training**
2: Initialize noise predictor $\epsilon_\theta$
3: **while** not done **do**
4:   Sample $\mathbf{x}_{t-T_o:t+T_p}$ from $\mathcal{D}$
5:   Sample latent variables $\hat{\mathbf{c}}^{\text{prior}}_{t:t+T_p} \sim p_\phi$, $\hat{\mathbf{c}}^{\text{post}}_{t:t+T_p-2} \sim q_\psi$
6:   Train causal diffusion model (noise predictor $\epsilon_\theta$) to generate actions $\mathbf{a}_{t+1:t+T_p}$, conditioned
   on $\mathbf{x}_{t-T_o:t}$, $\hat{\mathbf{c}}^{\text{prior}}_{t:t+T_p}$, $\hat{\mathbf{c}}^{\text{post}}_{t:t+T_p-2}$, and $\mathbf{y}$
7:   Train encoder $q_\psi$ with the contrastive improvement loss $\mathcal{L}_{\text{contrast}}$
8: **end while**
 **// Execution**
9: Initialize environment: $s_0 \sim$ `Env.reset()`, set $t \leftarrow 0$
10: **while** not done **do**
   *// Observe and infer latent factors*
11:   Observe recent trajectory $\mathbf{x}_{t-T_o:t}$
12:   Sample latent variables $\hat{\mathbf{c}}^{\text{prior}}_{t:t+T_p} \sim p_\phi$
   *// Generate actions using causal diffusion model*
13:   Generate actions (zig-zag sampling) $\hat{\mathbf{a}}_{t+1:t+T_p}$ conditioned on $\mathbf{x}_{t-T_o:t}$, $\hat{\mathbf{c}}_{t:t+T_p}$, and $\mathbf{y}$ via
   learned noise predictor $\epsilon_\theta$
   *// Execute action(s) in environment*
14:   **for** each step $i = 1$ to $T_a$ **do**
15:     Execute $\hat{\mathbf{a}}_{t+i}$ in `Env`, observe $s_{t+i+1}, r_{t+i}$
16:     Append $(s_{t+i}, \hat{a}_{t+i}, r_{t+i})$ to trajectory buffer
17:   **end for**
18:   Update $t \leftarrow t + T_a$
19: **end while**

---

| Environment | LDP (AF) | Ours (AF) | LDP (AF, SubOpt) | Ours (AF, SubOpt) |
|---|---|---|---|---|
| Lift | $0.67_{\pm 0.01}$ | $\mathbf{0.78}_{\pm 0.05}$ | $\mathbf{1.00}_{\pm 0.00}$ | $0.98_{\pm 0.02}$ |
| Can | $0.78_{\pm 0.04}$ | $\mathbf{0.85}_{\pm 0.07}$ | $\mathbf{0.98}_{\pm 0.00}$ | $\mathbf{0.98}_{\pm 0.02}$ |
| Square | $0.47_{\pm 0.03}$ | $\mathbf{0.54}_{\pm 0.05}$ | $0.83_{\pm 0.01}$ | $\mathbf{0.89}_{\pm 0.03}$ |

Table A3: **Results (success rate) on action-free demonstrations.** Here, AF and SubOpt indicate using Action-free and suboptimal demonstrations on Robomimic tasks, respectively (following the settings in LDP (Xie et al., 2025)).

layer of size 64) followed by a GRU. The GRU output is used to parameterize a Gaussian distribution from which the latent variables are sampled.

The state and reward decoders are implemented as separate MLPs, each consisting of two fully connected layers of size 64 with ReLU activations. For the prior network, we use the output of the previous step's latent distribution embedding (shared GRU) and feed it into a two-layer MLP (each layer of size 32) to predict the parameters of the prior distribution. For the dimensionality of latents, we choose 20 for Cheetah, Walker, Ant, Maze; 64 for Robomimic, Kitchen, Libero.

**Loss Function**   At each time step $t$, we optimize the following losses:

$$\mathcal{L}_{\text{ELBO},t} = \underbrace{\mathbb{E}_{q_\psi(\mathbf{c}_t \mid \mathbf{x}_{t-T_x:t+1})} \left[ -\log p_\theta(\mathbf{x}_t \mid \mathbf{x}_{t-1}, \mathbf{c}_t) \right]}_{\text{Reconstruction loss}} + \underbrace{D_{\text{KL}} \left( q_\psi(\mathbf{c}_t \mid \mathbf{x}_{t-T_x:t+1}) \,\|\, p_\phi(\mathbf{c}_t \mid \mathbf{c}_{t-1}) \right)}_{\text{KL regularization}}.$$

Here, $\mathbf{x}_t$ may include different components depending on the setting (e.g., $\mathbf{x}_t = \{\mathbf{s}_t, \mathbf{a}_t\}$ or $\mathbf{x}_t = \mathbf{s}_t$), and $\mathbf{c}_t$ denotes the latent context variable inferred from a temporal block of observations. The first term encourages accurate reconstruction of the current observation $\mathbf{x}_t$ conditioned on its immediate past and the latent $\mathbf{c}_t$, while the second term regularizes the posterior to remain close to the learned prior $p_\phi(\mathbf{c}_t \mid \mathbf{c}_{t-1})$.

We implement the ELBO loss as a weighted combination of the reconstruction loss and the KL divergence:

$$\mathcal{L}_{\text{ELBO}} = \sum_{t=1}^{T-2} \left[ \|\hat{\mathbf{x}}_t - \mathbf{x}_t\|_2^2 + \lambda_{\text{KL}} \cdot D_{\text{KL}} \left( q_\psi(\mathbf{c}_t \mid \mathbf{x}_{t-T_x:t+1}) \,\|\, p_\phi(\mathbf{c}_t \mid \mathbf{c}_{t-1}) \right) \right],$$

where $\hat{\mathbf{x}}_t$ is the model's reconstruction of the observation $\mathbf{x}_t$, and $\lambda_{\text{KL}}$ is weighting coefficient. The reconstruction is computed using mean squared error (MSE), and the KL divergence is computed in closed form for Gaussian posteriors and priors. The hyperparameter $\lambda_{\text{KL}}$ is set to be $0.01$ and the learning rate is set to be $3e-4$.

### D.2.2 PLANNER

For the planner, we consider two scenarios: (i) generating both states and actions, and (ii) generating states only. For the former, we build upon the Diffuser framework (Janner et al., 2022), which directly models full trajectories. For the latter, we adopt the Decision Diffuser (DD) framework (Ajay et al., 2022), where the model generates future states and uses an inverse dynamics model to recover the corresponding actions via inverse dynamics model.

For type (i) (full state-action trajectory generation), we apply our method to the Cheetah and Ant environments. For the noise predictor, we use a 1D U-Net (Ronneberger et al., 2015) with a kernel size of 5, channel multipliers set to (1, 2, 2, 2), and a base channel width of 32. The model is trained using the Adam optimizer (Kingma, 2014) with a learning rate of $3 \times 10^{-4}$, a batch size of 64, and for 1 million training steps. We adopt classifier guidance (CG) (Ho et al., 2020) with gradient guidance on computed return, with a guidance scale $\omega = 1.5$. The observation horizon is set to 10 for both environments. The planning horizon $T_p$ is set to 16 for Cheetah and 32 for Ant, with an action execution horizon of 1. These hyperparameters are kept consistent across baselines, including Diffuser, DF, MetaDiffuser, and Diffuser combined with LILAC and DynaMITE for the Cheetah and Ant experiments (those in Table 1 and Appendix Table A4). For other components (e.g., VAE) in LDCQ, we employ all the hyperparameters in their original implementation (Venkatraman et al., 2024).

For type (ii) (state-only generation with inverse dynamics), we use a Transformer-based noise predictor with a hidden dimension of 256 and a head dimension of 32. The architecture includes 2 DiT blocks for Walker, Kitchen, and Maze2D, and 8 DiT blocks for LIBERO. The model is trained using the Adam optimizer (Kingma, 2014) with a learning rate of $3 \times 10^{-4}$, a batch size of 128, and for 1 million training steps. The number of diffusion timesteps is 500. The observation horizon is set to 4 for Kitchen, 2 for LIBERO, and 10 for the other environments. The planning horizon $T_p$ is set to 16 for Kitchen, 10 for LIBERO, and 32 for the others. The action execution horizon is 8 for both Kitchen and LIBERO, and 10 for the remaining environments. For the inverse dynamics model, we use an MLP-based diffusion model consisting of a 3-layer MLP with 128 hidden units, preceded by a 2-layer embedding module with 64 hidden units. This model is trained for 1 million gradient steps.

For both cases, we set the coefficient of the contrastive improvement loss $\mathcal{L}_{\text{contrast}} = \max\{0, \mathcal{L}_{\text{prior}} - \mathcal{L}_{\text{post}}\}$ to 0.1. The key hyper-parameters are summarized in Table A10.

### D.2.3 POLICY

For the DP-based policy, we adopt the same architecture as the planner described earlier for Cheetah, Maze2D, Kitchen, Ant, and Walker. For LIBERO, we use a Transformer-based noise predictor with a decoder architecture comprising 12 layers, 12 attention heads, and a hidden embedding dimension of 768. Following DP (Chi et al., 2023), we apply dropout with a rate of 0.1 to both the input embeddings and attention weights. The number of diffusion timesteps is 500. When conditioning is used, we incorporate a Transformer encoder with 4 layers to encode the condition tokens, which include a sinusoidal timestep embedding and projected observed trajectory tokens (all mapped to

the same embedding dimension). In this encoder-decoder setup, causal masking is applied to ensure autoregressive generation. In the unconditioned case, we prepend the sinusoidal timestep embedding to the input sequence and use a BERT-style encoder-only Transformer. All environments (Cheetah, Ant, Kitchen, Maze2D, Walker, and LIBERO) are trained using the AdamW optimizer with a learning rate of $10^{-4}$, weight decay $10^{-3}$, $\beta_1 = 0.9$, and $\beta_2 = 0.95$. Layer normalization is applied before each Transformer block for stability. The observation, planning, and action horizons follow the same settings used for the planner in each environment.

For the IDQL-based policy, we align all hyperparameters for Cheetah and Ant with the original IDQL implementation, using an observation, planning, and action horizon of 1. Hence, in IDQL-based ones, we do not consider autoregressive modeling. Similarly, for both cases, we use consider the coefficient before the contrastive improvement loss as 0.1.

### D.2.4 HYPERPARAMETERS OF CONTRASTIVE IMPROVEMENT LOSS

We set $\lambda_{\text{prior}}$, $\lambda_{\text{rel}}$ to be fixed as 0.1 across all settings. $m$ is set to be the $0.05 \times \mathcal{L}_{\text{prior}}$ during the beginning of each epoch.

### D.3 CONNECTION TO BAYESIAN FILTERING

In the absence of explicitly designed latent variables, our model can be interpreted as a form of *Bayesian filtering* (Chen et al., 2003). Under a general formulation of the hidden Markov model (HMM) (Rabiner & Juang, 1986) with an additional latent dependency on observation ($\mathbf{c} \rightarrow \mathbf{x}$), the latent process over $\mathbf{c}$ captures the underlying stochasticity present in the demonstration data, which arises from both the environment dynamics and the behavior policy. In this view, the latent variable acts as a compact and expressive representation that summarizes the uncertainty in past observations, thereby improving the prediction of future observations. This, in turn, facilitates more robust policy learning and planning in the general settings.

## E EXTENDED RELATED WORKS

### E.1 DIFFUSION MODEL-BASED DECISION-MAKING

Recent advances use diffusion models as the planner and policy for both reinforcement learning (RL) and imitation learning (IL). RL agent aims to learn a policy that maximizes cumulative rewards through interaction with an environment (Sutton et al., 1998). The agent observes a sequence of transitions $(\mathbf{s}_t, \mathbf{a}_t, r_t, \mathbf{s}_{t+1})$, where $\mathbf{s}_t \in \mathcal{S}$ denotes the state, $\mathbf{a}_t \in \mathcal{A}$ the action, $r_t \in \mathbb{R}$ the received reward, and $\mathbf{s}_{t+1}$ the next state. The goal is to learn a policy $\pi(\mathbf{a} \mid \mathbf{s})$ that maximizes the expected return: $\pi^* = \arg\max_\pi \mathbb{E}_\pi \left[ \sum_{t=0}^{\infty} \gamma^t r_t \right]$, where $\gamma \in [0, 1)$ is the discount factor. In contrast, IL (Hussein et al., 2017) focuses on learning policies from expert demonstrations, often without access to the reward signal. A common approach is behavior cloning (BC) (Pomerleau, 1991), which formulates IL as a supervised learning problem by maximizing the likelihood of expert actions given observed states, i.e., learning a policy $\pi(\mathbf{a} \mid \mathbf{s})$ that closely imitates the expert policy $\pi_e(\mathbf{a} \mid \mathbf{s})$.

**Diffusion Planner** Diffusion-based planning methods are commonly used to approximate the sequence of future states and actions from a given current state. By leveraging the conditional generation capabilities of diffusion models—such as guidance techniques (Dhariwal & Nichol, 2021; Ho & Salimans, 2022)—these methods can generate plans (i.e., state trajectories) that satisfy desired properties, such as maximizing expected rewards. Taking Denoising Diffusion Probabilistic Models (DDPM (Ho et al., 2020))-based approaches as an example, these methods learn a generative model over expert trajectories $\tau = \{(\mathbf{s}_0, \mathbf{a}_0), \ldots, (\mathbf{s}_T, \mathbf{a}_T)\}$ by modeling a forward-noising process: $q(\mathbf{x}^k \mid \mathbf{x}^{k-1}) = \mathcal{N}(\mathbf{x}^k; \sqrt{\alpha_k} \mathbf{x}^{k-1}, (1 - \alpha_k)\mathbf{I})$, and a parameterized denoising model $p_\theta(\mathbf{x}^{k-1} \mid \mathbf{x}^k)$ to reverse the process. Here, $k$ denotes the diffusion step, $\mathbf{x}^0$ is a clean sub-sequence sampled from the expert trajectory $\tau$, and $\alpha_k$ controls the variance schedule at step $k$.

During inference, trajectories are generated by starting from Gaussian noise and iteratively denoising through the learned reverse process. This generation can be optionally conditioned on the initial state or other guidance signals $\mathbf{y}$, such as rewards, goals, or other constraints: $\hat{\tau} \sim p_\theta(\tau \mid \mathbf{s}_0, \mathbf{y})$.

These methods generally fall into two main categories: (1) learning a joint distribution over state-action trajectories, as in Diffuser (Janner et al., 2022), or (2) learning only state trajectories via diffusion and using an inverse dynamics model to recover actions, as in Decision Diffuser (DD) (Ajay et al., 2022). Beyond these, several variants extend diffusion-based planning in different directions. For example, Latent Diffuser (Li, 2024) plans in a high-level latent skill space to improve generalization and LDP (Xie et al., 2025) plans with high-level latent actions directly from high-dimensional action-free demonstrations. Other approaches incorporate multi-task context to enhance adaptation and performance in unseen tasks, including MetaDiffuser (Ni et al., 2023), AdaptDiffuser (Liang et al., 2023), and MTDiff-p (He et al., 2023). In addition, recent efforts have explored various extensions of diffusion planning, such as ensuring safety during generation (Xiao et al., 2025), handling multi-agent scenarios (Jiang et al., 2023; Ajay et al., 2023b), learning skills (Liang et al., 2024b), and application in RL from human feedback (RLHF) (Dong et al., 2024).

**Diffusion Policy** In contrast to diffusion-based planners, Diffusion Policy methods directly parameterize the policy $\pi_\theta(\mathbf{a} \mid \mathbf{s})$ using diffusion models. For example, Diffusion Policy (Chi et al., 2023) uses a diffusion model to generate actions with expressive, multimodal distributions. DPPO (Ren et al., 2025) extends this idea by modeling a two-layer MDP structure, where the inner MDP represents the denoising process and the outer MDP corresponds to the environment. This framework enables fine-tuning of diffusion-based policies in RL settings. Another line of work integrates diffusion models with model-free methods for offline RL by using diffusion models as to model the action distributions (Wang et al., 2022; Hansen-Estruch et al., 2023; Chen et al., 2023; Lu et al., 2023).

Recent explorations have also aimed to unify diffusion-based planning and policy learning within a single framework. For example, the Unified Video Action model (UVA) (Li et al., 2025) and Unified World Models (UWM) (Zhu et al., 2025) leverage diffusion models to jointly model planning and action generation, demonstrating scalability on large-scale robotic tasks with pre-training. In a similar spirit, `Ada-Diffuser` provides a general framework that can be integrated into both diffusion planners and diffusion-based policies. However, `Ada-Diffuser` differs in its explicit modeling of latent factors that influence the data generation process. By incorporating latent identification directly into the diffusion process, `Ada-Diffuser` enables more structured, context-aware decision-making in partially observable and dynamically changing environments.

### E.2 LATENT BELIEF STATE LEARNING IN POMDP

In partially observable Markov decision processes (POMDPs), single-step observations are typically insufficient for making optimal decisions. A common strategy to overcome this limitation involves encoding an agent's history, encoding past observations and actions into a belief state that captures a distribution over latent environmental states. Although such belief representations can, in theory, support optimal policy derivation (Kaelbling et al., 1998; Hauskrecht, 2000; Gangwani et al., 2020), their exact computation depends on full knowledge of the transition and observation models. This requirement quickly becomes intractable in high-dimensional settings.

To address this, recent work has focused on learning approximate belief representations directly from data. Notable approaches include those using recurrent neural networks (Guo et al., 2018) and variational inference methods (Igl et al., 2018; Gregor et al., 2018), which enable agents to encode temporal structure and uncertainty into compact latent embeddings. These representations are then used to inform downstream policy learning, optimizing for cumulative rewards.

This direction also aligns with developments in meta-reinforcement learning and non-stationarity, where belief states or Bayesian embeddings are used to capture hidden task contexts. Agents trained across a distribution of tasks can use these latent variables to infer new environments and adapt quickly (Zintgraf et al., 2021; Nguyen et al., 2021; Huang et al., 2021; Rakelly et al., 2019; Xie et al., 2021; Feng et al., 2022; Feng & Magliacane, 2023; Liang et al., 2024a; Wang & Huang, 2025). For example, MetaDiffuser (Ni et al., 2023) incorporates task context as conditioning input to diffusion-based decision models. Similarly, Pertsch et al. (2021) and Zeng et al. (2023) use similar variational objectives (ELBO loss) to learn latent skill priors and predictive information for RL, where these latents greatly help policy learning.

Our approach diverges from these by offering theoretical guarantees on the identifiability of latent factors from minimal temporal observations. Rather than depending on diverse multi-environment data, we introduce a framework that captures the full data generation process in RL using diffusion

models. In contrast to MetaDiffuser, which assumes static task-level context, our model treats the latent context as a dynamic, time-evolving process that governs both environment transitions and agent behavior, capturing the underlying temporal structure of RL trajectories more faithfully.

### E.3 AUTOREGRESSIVE DIFFUSION MODELS

To model temporal consistency and dynamics in sequential data such as videos and audios, recent work has incorporated autoregressive structures into diffusion models. These approaches differ in how they condition on prior time steps during generation and can be categorized into two main categories. **(1) Conditioning on clean (denoised) inputs** ((Zheng et al., 2024; Gao et al., 2024b; Blattmann et al., 2023)). At each time step $t$, the denoising model is conditioned on the previously denoised outputs $\{\mathbf{x}^0_{<t}\}$: $p_\theta(\mathbf{x}^{k-1}_t \mid \mathbf{x}^k_t, \mathbf{x}^0_{<t})$, where $\mathbf{x}^k_t$ is the current noisy input, and $\mathbf{x}^0_{<t}$ denotes the clean (fully denoised) observations from earlier time steps. **(2) Conditioning on noisy inputs** ((Ho et al., 2022; Chen et al., 2024; Xie et al., 2024b; Sand-AI, 2025)). These methods instead condition on previous time steps at their corresponding noise levels. This setting can be further divided into two cases: (a) *fully noisy conditioning* (Ho et al., 2022): the model conditions on all prior time steps at the same noise level $k$: $p_\theta(\mathbf{x}^{k-1}_{<t}, \mathbf{x}^{k-1}_t \mid \mathbf{x}^k_t, \mathbf{x}^k_{<t},)$. (b) *partially noisy conditioning*: each previous time step $i < t$ is conditioned at its own noise level $k_i$, which may vary over time: $p_\theta(\mathbf{x}^{k_0-1}_0, \mathbf{x}^{k_1-1}_1, \ldots, \mathbf{x}^{k_T-1}_T \mid \mathbf{x}^{k_0}_0, \mathbf{x}^{k_1}_1, \ldots, \mathbf{x}^{k_T}_T)$. Specifically, Diffusion Forcing (DF) (Chen et al., 2024) proposes a general framework in which each time step $\mathbf{x}_t$ assigns an independent noise level. In contrast, other works adopt time-dependent noise schedules that vary with the temporal index (Xie et al., 2024b; Sand-AI, 2025; Wu et al., 2023).

To model the causal generative process of RL trajectories, our approach also employs time-dependent noise scheduling to capture temporal dynamics. However, unlike prior work, we further integrate the identification of latent factors directly into the denoising process. This is achieved through a structured reinforcement step during training and a zig-zag inference procedure at test time, enabling our model to more faithfully recover the underlying causal structure in sequential decision-making.

### E.4 SUMMARY

To sum up, we compare our approach with representative diffusion- and meta-learning–based baselines (Table A11). Diffuser, DP, IDQL, and DD do not model or infer latent contexts; DF adopts autoregressive denoising but still lacks context inference. Meta-Diffuser, LILAC, and DynaMITE learn latents via meta-learning but omit our minimal–sufficient block design and backward refinement. LDCQ and LDP model only high-level latent actions/skills without explicit context identification. In contrast, our method jointly models latent factors, employs full autoregressive denoising with zig-zag sampling, and introduces a backward refinement mechanism that enables identifiable latent contexts.

| Method | Latent Factors | AR Denoising | Min. & Suff. Obs. |
|---|---|---|---|
| **Ours** | Yes (dyn., rew., act.) | Yes | Yes (refine, zig-zag) |
| Diffuser / DP / DD / IDQL | No | No | No |
| DF | No | Yes | No |
| Meta-Diffuser / LILAC / DynaMITE | Yes (dyn., rew. only) | No | No |
| LDCQ | Yes (hi-level act.) | No | No |
| LDP | Yes (hi-level act.) | No | No |

Table A11: Comparison with representative baselines on whether they model latent contexts, use autoregressive (AR) denoising, and enforce minimal & sufficient observation blocks.

## F BENCHMARK SETTINGS AND ILLUSTRATIONS

### F.1 LATENT CHANGE FACTORS DESIGN

We consider the latent change factors on dynamics and rewards. We consider the Half-Cheetah and Ant environments from the OpenAI Gym suite, which are widely used MuJoCo locomotion benchmarks (Brockman et al., 2016) for evaluating continuous control algorithms. In Half-Cheetah,

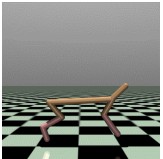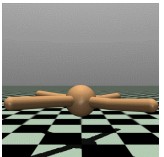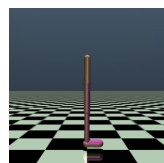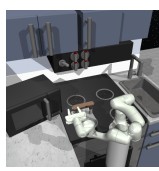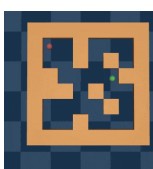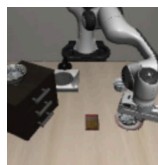

Figure A3: **Illustrations of the Benchmarks.** From left to right: Half-Cheetah, Ant, Walker, Franka-Kitchen, Maze2D, and LIBERO.

the agent is a planar bipedal robot with a 17-dimensional state space and a 6-dimensional continuous action space, where the goal is to move forward by applying torques to six actuated joints. In Ant, a quadrupedal robot operates in a 3D space with a 111-dimensional state space and an 8-dimensional action space, requiring more complex coordination across its four legs. In both environments, the reward encourages forward velocity while penalizing excessive control inputs and, in the case of Ant, also promotes stable contact with the ground. We consider variants of the Half-Cheetah environment to study changes in dynamics, specifically **Cheetah-Wind-E** and **Cheetah-Wind-S**, which introduce external wind forces applied to the agent. In **Cheetah-Wind-E**, an opposing wind force is applied at the beginning of each episode and remains constant throughout, defined as $f_w = 10 + 5\sin(0.8i)$, where $i$ is the episode index. For this case, since $\mathbf{c}$ change over episode, we use data from several consecutive episodes to estimate it. In **Cheetah-Wind-S**, the wind force varies at every time step according to the same formula $f_w = 5 + 5\sin(0.5t)$, with $t$ now representing the time step in each episode. We also consider variations in the reward function. In **Cheetah-Dir-E**, the reward depends on a time-varying goal direction, requiring the agent to alternate between moving forward and backward. Specifically, the reward at episode $t$ is defined as

$$r_t = d_t \cdot v_t - 0.1\|\mathbf{a}_t\|^2,$$

where $v_t$ is the agent's forward velocity, $\mathbf{a}_t$ is the action vector (torques applied), and $d_t \in \{-1, +1\}$ indicates the target direction at time $t$. The direction signal $d_t$ changes, giving a non-stationary reward function that challenges the policy to adapt to shifting goals. Specifically, we consider

$$d_t = \sigma(5 \cdot \sin(2\pi t/200)),$$

where $\sigma(\cdot)$ denotes the sigmoid function, $\alpha$ controls the sharpness of the transition, and $T$ determines the switching period. This formulation induces a smooth periodic change in the preferred direction of movement, requiring the policy to adapt to gradually shifting objectives.

We also consider a directional reward variant for the **Ant** environment, denoted as **Ant-Dir-E**, where the agent is required to alternate its movement direction over time. The reward function at time step $t$ is defined as

$$r_t = (2d_t - 1) \cdot v_t^x - 0.1\|\mathbf{a}_t\|^2,$$

where $v_t^x$ is the velocity of the agent's torso along the x-axis (forward direction), $\mathbf{a}_t$ is the 8-dimensional action vector, and $d_t \in [0, 1]$ is a smooth directional signal. Similarly, we define $d_t$ as:

$$d_t = \sigma(5 \cdot \sin(2\pi t/200)),$$

where $\sigma(\cdot)$ denotes the sigmoid function. This formulation causes the preferred movement direction to alternate approximately every 100 steps. Notably, for these settings with periodic changes (i.e., where latent factors do not evolve at every timestep), we estimate the latent variables periodically and perform refinement in the causal diffusion model only when changes are detected. This follows the same overall framework, but operates at a coarser temporal resolution aligned with the latent change frequency.

## F.2 OVERVIEW ON OTHER BENCHMARKS

Fig. A3-A4 give the illustrations on the used benchmarks. Specifically, other than Cheetah and Ant we introduced before, for others, we consider the basic settings in offline RL. Specifically,

**Maze2D.** Maze2D tasks focus on goal-directed navigation in a 2D plane, where the agent must traverse a maze-like environment to reach specified targets. These settings are designed to evaluate an agent's ability to reason spatially and follow optimal trajectories based solely on positional and velocity observations.

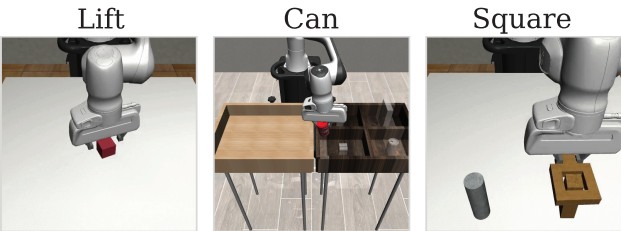

Figure A4: **Illustrations of RoboMimic Benchmark.**

**Franka-Kitchen.** The Franka-Kitchen environment (Gupta et al., 2020) involves a robotic arm interacting with a series of articulated objects in a realistic kitchen setting. Tasks are composed of multiple stages, such as opening doors or toggling switches, and are intended to assess an agent's capability in handling long-horizon, multi-step manipulation.

**Walker.** The Walker2D environment features a two-legged robot that must learn to walk and balance using continuous torque control. The agent's objective is to maintain forward motion while remaining upright, which requires learning dynamic stability and coordination.

**LIBERO (Liu et al., 2023).** The Libero benchmark offers a diverse set of continual learning tasks focused on object manipulation and generalization:

- **LIBERO-Object:** The robot is required to manipulate a variety of novel objects through pick-and-place operations. Each task introduces previously unseen objects, encouraging the agent to incrementally build knowledge about object-specific properties and behaviors.
- **LIBERO-Goal:** All tasks share a common object set and spatial layout, but vary in goal specifications. This setup tests the agent's ability to continually adapt to new task intents and motion targets without changes in the visual scene.
- **LIBERO-Spatial:** Tasks involve repositioning a bowl onto different plate locations. Although the objects remain fixed, the spatial configurations vary across tasks, requiring the robot to incrementally acquire relational spatial understanding.

**RoboMimic.** RoboMimic (Mandlekar et al., 2021) provides a set of manipulation tasks based on human teleoperation demonstrations, varying in difficulty and required precision:

- **Lift:** The robot arm is tasked with lifting a small cube off the table. This task serves as a foundational manipulation scenario focused on grasping and vertical motion.
- **Can:** The robot must retrieve a cylindrical can from a cluttered bin and place it into a designated smaller container. This task introduces greater complexity due to object shape and the need for accurate placement.
- **Square:** A fine-grained insertion task where the robot picks up a square nut and places it onto a vertical rod. This is the most challenging of the three, requiring precise alignment and control for successful completion.

## G  OTHER DETAILS ON ADA-DIFFUSER

### G.1  LATENT ACTION PLANNER

For the latent action planner, we align our settings with those used in LDP (Xie et al., 2025), specifically focusing on learning directly from image-based demonstrations. We first use a variational autoencoder (VAE) to extract latent representations $\mathbf{z}$ from raw images via image encoders. An inverse dynamics model is then trained to recover actions $\mathbf{a}_t$ from pairs of latent states $(\mathbf{z}_t, \mathbf{z}_{t+1})$. A planner is subsequently trained to forecast future latents $\mathbf{z}$. Hence, the objective function is $\mathcal{L}_{\text{IDM}}(\xi, \mathbf{z}) = \mathbb{E}_{t,\epsilon}\left[\left\|\epsilon_\xi\left(\hat{a}_k; \mathbf{c}_k, \mathbf{z}_k, \mathbf{z}_{k+1}, t\right) - \epsilon\right\|^2\right]$, where where $k$ is the time step and $t$ is the diffusion step.

In our framework, we treat the latent factors $\mathbf{c}$ as high-level latent actions that influence the evolution of $\mathbf{z}$. These latent factors are jointly used with $\mathbf{z}$ to perform both inverse dynamics modeling and latent forecasting, enabling structured planning in the latent space.

We follow the experimental settings established in LDP (Xie et al., 2025). Specifically, we use expert demonstrations alongside action-free and suboptimal demonstrations. All hyperparameters and architectural choices for the diffusion models are kept identical to those used in the original LDP implementation. We also directly utilize the pre-trained image encoder provided by LDP. The only modification in our framework is the introduction of an additional latent factor $\mathbf{c}$ trained by our latent factor identification stage, which is incorporated into the model to enhance latent action planning.

### G.2 NOISE SCHEDULING

In the autoregressive setting, we consider a monotonic increasing denoising schedule $\{k_1, \ldots, k_T\}$. In practice, we use a linear schedule where $k_i = \frac{i}{T}K$, with $K$ denoting the maximum diffusion step used in both training and sampling. We segment the sequence into temporal blocks of length $T_x + 1$ ($T_x = T_o$ in all settings), and slide the time window forward by one step at a time. This design ensures that the denoising steps progressively increase across the block, aligning the diffusion process with the underlying temporal structure. Such a schedule encourages early steps to rely more on strong priors and later steps to refine based on more contextual information. Additionally, for better illustration, Fig. A2 provides a detailed illustration of the zig-zag sampling process within a temporal block of 4 timesteps.

## H SPECIFIC DESIGN CHOICES FOR BASELINES

For all baselines, unless otherwise specified, we use the same set of diffusion parameters detailed in Appendix D.2.2–D.2.3. Below, we provide additional details on how specific methods are evaluated. While their diffusion backbones remain consistent as in Appendix D.2.2–D.2.3, these methods include custom design choices and method-specific hyperparameters that are evaluated accordingly.

### H.1 DETAILS ON LILAC AND DYNAMITE

In these settings, we extend both LILAC and DynaMITE by incorporating a context encoder to infer latent context variables $\mathbf{c}_t$, following their respective designs. Both methods learn belief state embeddings from historical observations. For a fair comparison, we use the same latent identification network architecture as in our framework, but modify the inputs according to each method's assumptions.

Specifically, LILAC and DynaMITE condition their inference networks solely on the historical trajectory $\mathbf{x}_{1:t}$, without access to current and future information. Additionally, consistent with their original implementations, we do not include a separate prior head on top of the GRU; both methods share the encoder for posterior inference and prior prediction. And the primary difference (in implementation) between these two methods lies in the temporal context used: LILAC maintains the full belief over the entire history, i.e., it conditions on $\mathbf{x}_{1:t}$ to infer $\mathbf{c}_{t+1}$, while DynaMITE uses only the most recent context, i.e., it infers $\mathbf{c}_{t+1}$ based solely on $\mathbf{x}_t$.

All other hyperparameters are aligned with those used in our Stage 1 training. The estimated context variables are then provided as additional conditioning inputs to the diffusion-based models.

### H.2 DETAILS ON DIFFUSION FORCING

For Diffusion Forcing, we adopt the same autoregressive noise schedule as in our method, which accounts for causal uncertainty, similarly to the formulation in Eq. D.1 of (Chen et al., 2024), to ensure a fair comparison. Additionally, we use the Monte Carlo Guidance (MCG) mechanism introduced in (Chen et al., 2024) for Maze2D, following the original setup. For all other environments, we use the same classifier guidance scheme as the other baselines to maintain consistency in evaluation.

# I  ABLATION ANALYSIS

## I.1  TRAINING/INFERENCE TIME ANALYSIS

We conduct all experiments on 4× NVIDIA A100 or 8× RTX 4090 GPUs, depending on the model scale and environment requirements. The main computational overhead in our framework arises from two components: (i) the latent factor identification network, and (ii) the denoise-and-refine steps in the diffusion model. During sampling, the additional cost comes from zig-zag latent exploration and latent variable sampling. However, these steps do not substantially increase either training or inference time.

To quantify this, we report the training and inference speed of our method compared to the base models DD and DP across all environments (Table A12). Our framework introduces only a moderate computational overhead — typically 1.2–1.3× the runtime of vanilla diffusion backbones, corresponding to roughly 20–30% extra training time and inference latency. This cost can be further reduced through parallel latent sampling, lightweight context encoders, or refinement only at inference. Moreover, we additionally evaluate a Picard-accelerated variant (Table A13, Shih et al. (2023)), where iterative refinement is parallelized by conditioning each denoising step on previously denoised nodes. With Picard iteration, inference time drops to 0.7–0.8× of our default iterative sampler while maintaining comparable performance, demonstrating the potential for further acceleration.

| Environment | Training Time (sec/epoch) | | Inference Latency (ms) | |
|---|---|---|---|---|
| | Ours vs DD | Ours vs DP | Ours vs DD | Ours vs DP |
| Cheetah | 72.1 / 60.1 (1.20) | 69.8 / 58.4 (1.20) | 182 / 114 (1.16) | 160 / 125 (1.28) |
| Ant | 79.5 / 64.3 (1.24) | 76.0 / 62.0 (1.23) | 148 / 125 (1.19) | 172 / 139 (1.24) |
| Walker | 85.3 / 67.1 (1.27) | 81.5 / 64.2 (1.27) | 182 / 144 (1.28) | 170 / 130 (1.31) |
| Maze2D | 90.2 / 72.0 (1.25) | 88.3 / 69.2 (1.28) | 184 / 149 (1.24) | 196 / 152 (1.29) |
| Libero | 104.0 / 81.0 (1.28) | 102.1 / 78.0 (1.31) | 209 / 169 (1.24) | 219 / 162 (1.35) |
| Kitchen | 117.8 / 88.1 (1.34) | 115.3 / 85.0 (1.36) | 228 / 180 (1.27) | 211 / 168 (1.26) |

Table A12: Training and inference time comparison for `Ada-Diffuser-planning` and `Ada-Diffuser-policy` variants. We report absolute times (sec/epoch or sec/rollout) and relative overheads.

| Environment | Ours (sec) | Ours+Picard (sec) |
|---|---|---|
| Cheetah | 1.51 | 1.15 |
| Ant | 1.67 | 1.25 |
| Walker | 1.83 | 1.40 |
| Maze2D | 1.94 | 1.47 |
| Libero | 2.18 | 1.62 |
| Kitchen | 2.45 | 1.84 |

Table A13: Picard-accelerated inference.

## I.2  ABLATION RESULTS

### I.2.1  FULL RESULTS SUPPLEMENT TO TABLE 2

Table A14 presents the full ablation results across all environments, as a supplement to Table 2. Overall, the results highlight the importance of the two key components in causal diffusion modeling: latent identification and autoregressive diffusion, both of which are critical for performance.

### I.2.2  NOISE SCHEDULE: LINEAR VS. LOGISTIC VS. SIGMOID

We adopt a linear noise schedule by default since any monotonic, bounded schedule suffices to model the data-generation process and linear is simple and stable in practice. To validate this choice, we

compare linear, logistic, and sigmoid schedules on three representative tasks. As shown in Table A15, performance remains stable across schedules with no significant differences, supporting our default choice.

| Task | Schedule | Performance |
|---|---|---|
| Cheetah | linear | −68.9 |
| | logistic | −63.6 |
| | sigmoid | −70.4 |
| Maze2D | linear | 161.4 |
| | logistic | 157.6 |
| | sigmoid | 168.5 |
| Franka-Kitchen | linear | 0.70 |
| | logistic | 0.72 |
| | sigmoid | 0.66 |

Table A15: Ablation on noise schedules. "Performance" is the task score (higher is better for Maze2D/Kitchen; lower magnitude negative is better for Cheetah as per the benchmark).

### I.2.3 Effect of Temporal Block Length on Latent Identification

We further analyze the impact of temporal block length on latent identification. As shown in Fig. A5, the results are consistent with findings reported in the main paper. When the number of observations is insufficient (e.g., $\leq 4$), identification performance degrades. Performance improves when the block length is in a moderate range (5–20), indicating that sufficient temporal context is beneficial. However, using overly long blocks ($> 20$) introduces redundancy and increases optimization difficulty, which in turn harms performance.

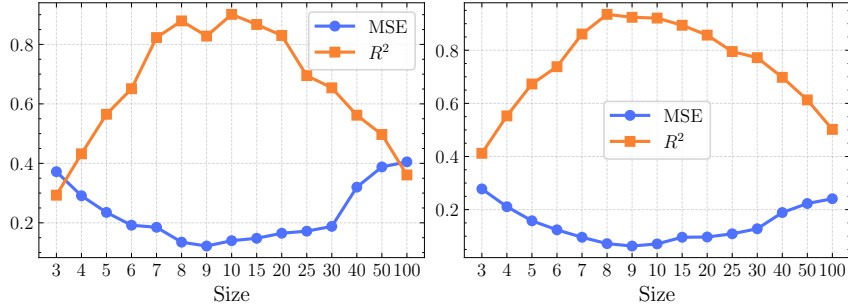

Figure A5: **Identification results (MSE of linear probing and $R^2$) versus the length of temporal blocks.** Left: Cheetah with time-varying wind; Right: Cheetah with time-varying rewards.

**Clustering** We assess whether the learned latent space organizes states by the underlying context on the Cheetah wind-change task, where the ground-truth latent evolves as $f_w(t) = 5 + 5\sin(0.5t)$. We sample 1000 time steps, discretize $f_w(t)$ into five equal-frequency bins to define target clusters, embed the corresponding observations into the 20-dimensional learned latent representation, and run $k$-means with $k = 5$. We compare our method with LILAC and DynaMIE, together with an ablation that without refinement. Results are given in Fig. A6.

### I.2.4 Latent Probing: Effect of Backward Refinement and Zig–Zag

To test whether backward refinement and zig–zag primarily help by correcting posterior mismatch, we perform a latent probing analysis on CHEETAH with changing wind. We linearly map the learned latent representation to the ground-truth wind variable using a simple least-squares probe (trained on a subset of blocks and evaluated on held-out blocks). Table A16 reports the mean squared error

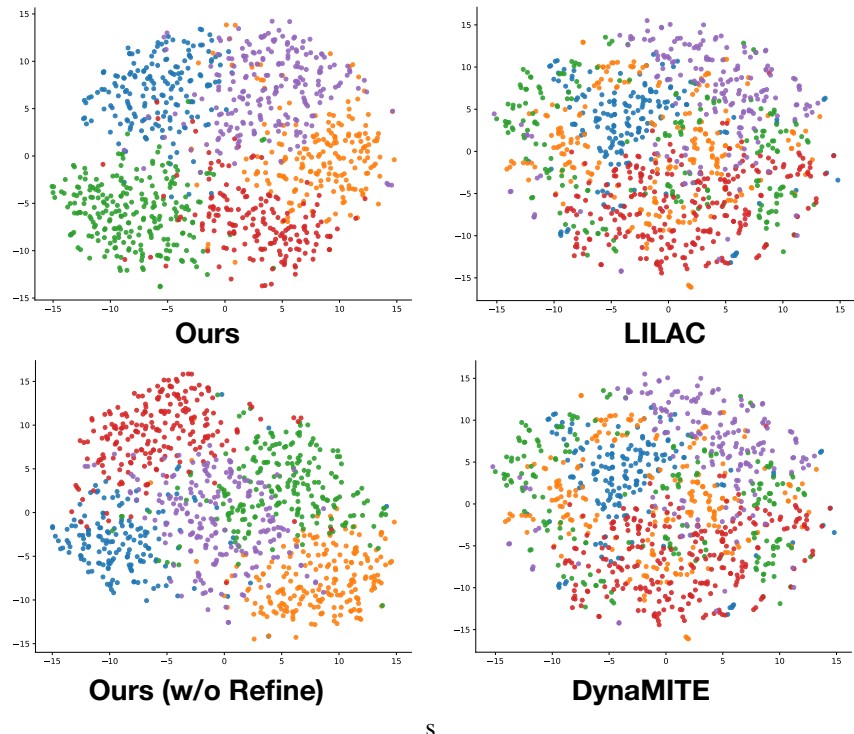

s

Figure A6: Clustering (t-SNE)results on Cheetah wind-change.

(MSE) of this probe for three variants: (i) the full model with backward refinement and zig–zag; (ii) without refinement; and (iii) without zig–zag.

| Variant | MSE |
|---|---|
| Full (with refinement + zig–zag) | **0.18** |
| w/o refinement | 0.28 |
| w/o zig–zag | 0.23 |

Table A16: Linear probing MSE for recovering the ground-truth wind latent on CHEETAH (changing wind). Lower is better.

**Analysis.** The full model achieves the lowest MSE, indicating more accurate recovery of the latent wind. Removing backward refinement yields the largest degradation ($0.18 \rightarrow 0.28$), consistent with the role of refinement in letting future evidence within a block update the latent posterior and reduce temporal lag. Disabling zig–zag also harms accuracy ($0.18 \rightarrow 0.23$), suggesting that alternating conditioning helps align the denoising trajectory with the latent dynamics rather than purely following the forward temporal pass. Together, these results support our claim that both components reduce posterior mismatch and improve latent identifiability, which in turn benefits planning and control in settings with evolving hidden factors.

### I.2.5 ON THE EFFECT OF PLANNING AND EXECUTION HORIZONS: LONG-HORIZON PLANNING

We study the robustness of our approach under increased planning and execution horizons ($T_p$ and $T_a$). Specifically, we evaluate on two challenging tasks—Franka-Kitchen-Partial and Libero-Long, where the original settings are Kitchen ($T_p = 16$, $T_a = 8$) and Libero ($T_p = 10$, $T_a = 8$). Results are in Fig. A7. When we increase these horizons, we observe that the baselines, DP and DF, suffer significant performance drops. In contrast, Ada-Diffuser maintains relatively high

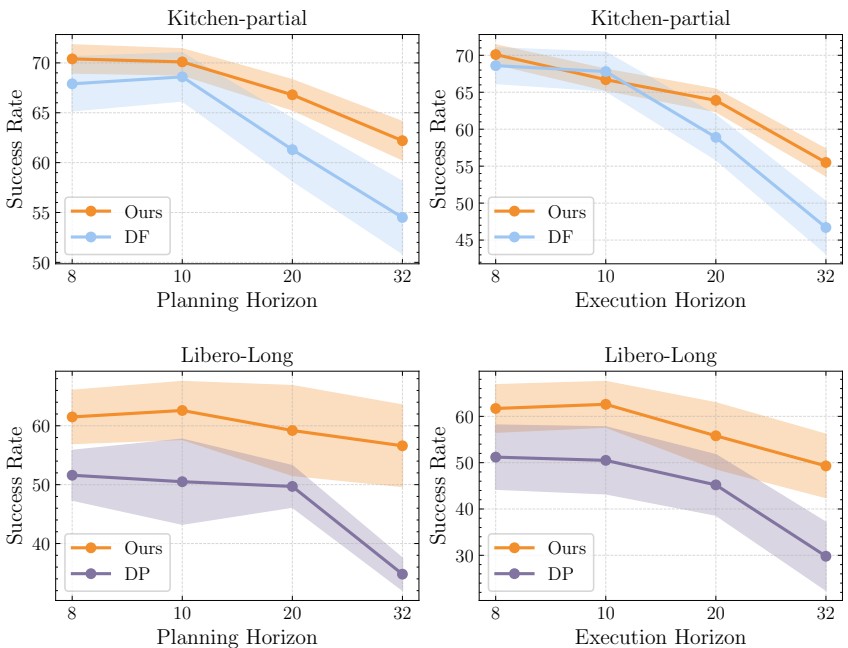

Figure A7: **Results with different planning and execution horizons.** We evaluate on Kitchen-partial and Libero-Long experiments.

performance. This demonstrates that modeling the underlying causal generative process, through autoregressive structure and latent representations, enables better **long-horizon planning**. Although we do not explicitly impose latent variables, our model implicitly learns representations that can track stochasticity and support smooth control.

| Environment | Diffuser | DF | MetaDiffuser | Diffuser + DynaMITE | Diffuser + LILAC | Ours | Ours + Meta | Oracle |
|---|---|---|---|---|---|---|---|---|
| Cheetah-Wind-E ($\mathbf{c}^s$) | -120.4 ± 12.7 | -105.8 ± 9.6 | -89.7 ± 6.5 | -79.2 ± 11.0 | -95.3 ± 7.4 | **-68.9** ± 7.6 | −62.4 ± 3.9 | −57.8 ± 6.7 |
| Cheetah-Wind-S ($\mathbf{c}^s$) | -148.5 ± 9.8 | -102.0 ± 10.2 | -106.8 ± 11.4 | -94.3 ± 9.6 | -105.6 ± 14.5 | **-73.5** ± 8.7 | -65.3 ± 11.2 | -58.1 ± 9.0 |
| Cheetah-Dir-E ($\mathbf{c}^r$) | 850.8 ± 54.2 | 902.1 ± 45.8 | 912.5 ± 37.9 | 930.4 ± 29.5 | 908.5 ± 37.6 | **943.3** ± 25.6 | 949.8 ± 24.1 | 962.1 ± 21.9 |
| Cheetah-Vel-E ($\mathbf{c}^r$) | -102.4 ± 18.2 | -85.6 ± 18.3 | -69.2 ± 7.5 | -76.3 ± 11.7 | -62.6 ± 11.1 | **-45.8** ± 9.5 | -39.2 ± 7.6 | -38.3 ± 8.9 |
| Ant-Dir-E ($\mathbf{c}^r$) | 188.6 ± 39.2 | 195.4 ± 47.0 | 245.9 ± 41.0 | 262.8 ± 27.5 | 229.4 ± 32.6 | **285.3** ± 24.5 | 296.4 ± 22.2 | 300.7 ± 23.6 |

Table A4: **Results (average returns) on `Ada-Diffuser`-Planner with latent factors that affects dynamics and rewards.** $\mathbf{c}^s$ and $\mathbf{c}^r$ indicate the changes on dynamics and reward, $E$ and $S$ represent the episodic and time-step changes. The results are with 5 random seeds. The bold ones are the best-performing ones, excluding meta-learning and oracle ones.

| Environment | DP | DP + DynaMITE | Ours + DP | Ours + DP (Oracle) | IDQL | IDQL + DynaMITE | Ours + IDQL | Ours + IDQL (Oracle) |
|---|---|---|---|---|---|---|---|---|
| Cheetah-Wind-E ($\mathbf{c}^s$) | -104.8 ± 10.9 | -72.2 ± 5.9 | **-58.5** ± 4.6 | -52.0 ± 3.5 | -97.5 ± 9.4 | -59.0 ± 11.2 | **-48.5** ± 7.9 | -41.6 ± 6.2 |
| Cheetah-Wind-S ($\mathbf{c}^s$) | -120.6 ± 11.5 | -76.5 ± 15.6 | **-52.9** ± 9.8 | -42.3 ± 6.7 | -87.8 ± 12.2 | -63.4 ± 6.7 | **-48.0** ± 7.2 | -44.7 ± 6.1 |
| Cheetah-Dir-E ($\mathbf{c}^r$) | 892.5 ± 60.8 | 949.6 ± 36.1 | **960.7** ± 40.2 | 972.4 ± 37.5 | 902.4 ± 45.2 | 938.6 ± 49.4 | **965.0** ± 37.5 | 969.8 ± 39.2 |
| Cheetah-Vel-E ($\mathbf{c}^r$) | -87.9 ± 6.5 | -72.7 ± 5.8 | **-41.0** ± 7.2 | -39.8 ± 6.7 | -80.2 ± 11.4 | -59.4 ± 6.5 | **-38.6** ± 7.7 | -33.8 ± 6.5 |
| Ant-Dir-E ($\mathbf{c}^r$) | 182.5 ± 41.2 | 275.2 ± 27.0 | **290.4** ± 49.4 | 312.5 ± 37.2 | 204.6 ± 25.6 | 269.3 ± 29.5 | **295.8** ± 32.7 | 309.6 ± 25.4 |

Table A5: **Results (average returns) on `Ada-Diffuser`-Policy with latent factors.** $\mathbf{c}^s$ and $\mathbf{c}^r$ indicate the changes on dynamics and reward, $E$ and $S$ represent the episodic and time-step changes. The results are with 5 random seeds. The bold ones are the best-performing ones, excluding meta-learning and oracle ones.

| Environment | Diffuser | DD | DF | LDCQ | Ours (DD) |
|---|---|---|---|---|---|
| Mixed | $52.6 \pm 2.3$ | $\mathbf{75.2} \pm 1.4$ | $73.7 \pm 1.9$ | $73.3 \pm 0.5$ | $74.6 \pm 1.6$ |
| Partial | $55.8 \pm 1.9$ | $57.3 \pm 1.2$ | $68.6 \pm 2.4$ | $67.8 \pm 0.8$ | $\mathbf{70.1} \pm 1.3$ |

Table A6: **Results (success rate (%)) on `Ada-Diffuser`-Planner without explicit latent factors on Franka-kitchen environment.** The results are with 5 random seeds.

| Environment | Diffuser | DD | DF | LDCQ | Ours (DD) |
|---|---|---|---|---|---|
| umaze | $113.5 \pm 2.8$ | $114.8 \pm 3.2$ | $116.7 \pm 2.0$ | $134.2 \pm 4.1$ | $\mathbf{148.6} \pm 3.7$ |
| medium | $121.5 \pm 5.6$ | $129.6 \pm 2.9$ | $\mathbf{149.4} \pm 7.5$ | $125.3 \pm 2.5$ | $148.6 \pm 3.1$ |
| large | $123.0 \pm 4.8$ | $131.5 \pm 4.2$ | $159.0 \pm 2.7$ | $150.1 \pm 2.9$ | $\mathbf{161.4} \pm 3.2$ |

Table A7: **Results on `Ada-Diffuser`-Planner without explicit latent factors on Maze-2D environment**. The results are averaged across 5 random seeds.

| Environment | Diffuser | DD | DF | LDCQ | Ours (DD) |
|---|---|---|---|---|---|
| medium-expert | $106.2 \pm 0.7$ | $108.8 \pm 2.0$ | $105.4 \pm 3.2$ | $109.3 \pm 0.4$ | $\mathbf{115.7} \pm 2.1$ |
| medium | $79.6 \pm 9.8$ | $82.5 \pm 1.6$ | $66.2 \pm 1.9$ | $69.4 \pm 2.4$ | $\mathbf{83.6} \pm 3.5$ |
| medium-replay | $70.6 \pm 0.6$ | $75.0 \pm 3.2$ | $72.2 \pm 2.6$ | $68.5 \pm 4.3$ | $74.3 \pm 2.8$ |

Table A8: **Results on `Ada-Diffuser`-Planner without explicit latent factors on Walker environment**. The results are averaged across 5 random seeds.

| Environment | DP | Ours (DP) |
|---|---|---|
| Spatial | $78.3 \pm 3.9$ | $\mathbf{79.2} \pm 4.2$ |
| Object | $92.5 \pm 2.6$ | $\mathbf{93.4} \pm 2.8$ |
| Long | $50.5 \pm 7.2$ | $\mathbf{62.6} \pm 4.9$ |

Table A9: **Results on `Ada-Diffuser`-Policy without explicit latent factors on Libero environment.** The results are averaged across 5 random seeds.

| Component | Type (i): Full Trajectory | Type (ii): State-Only |
|---|---|---|
| **Model Backbone** | 1D U-Net (Ronneberger et al., 2015) | Transformer (DiT) |
| **Architecture** | Kernel size: 5; channels: (1,2,2,2); base: 32 | Hidden dim: 256; head dim: 32 |
| **# DiT Blocks** | – | 2 (Walker, Kitchen, Maze2D), 8 (LIBERO) |
| **Optimizer** | Adam, lr = $3 \times 10^{-4}$ | Adam, lr = $3 \times 10^{-4}$ |
| **Batch Size** | 64 | 128 |
| **Training Steps** | 1M | 1M |
| **Diffusion Timesteps** | 150 | 200 |
| **Observation Horizon** $T_o$ | 10 | 4 (Kitchen), 2 (LIBERO), 10 (others) |
| **Planning Horizon** $T_p$ | 16 (Cheetah), 32 (Ant) | 16 (Kitchen), 10 (LIBERO), 32 (others) |
| **Execution Horizon** $T_o$ | 1 | 8 (Kitchen, LIBERO), 10 (others) |
| **Guidance** | CG, $\omega = 1.5$ | CFG |
| **Inverse Dynamics Model** | – | 2-layer embed (64), 3-layer MLP (128), 1M steps |
| **Refinement Loss Cofficient** | 0.1 | 0.1 |

Table A10: Planner configurations for type (i): full trajectory generation and type (ii): state-only generation with inverse dynamics.

| Cases | Cheetah-1 | Cheetah-2 | Ant | Maze2D | Walker | Kitchen | RoboMimic | LIBERO |
|---|---|---|---|---|---|---|---|---|
| Original | **-73.5** | **-52.9** | **295.8** | **161.4** | **115.7** | **0.70** | **0.85** | **93.4** |
| w/o refine | -82.0 | -60.7 | 261.2 | 156.5 | 107.4 | 0.63 | 0.78 | 90.2 |
| w/o zig-zag | -91.6 | -56.1 | 258.3 | 147.6 | 107.9 | 0.59 | 0.75 | 91.6 |
| same NS | -89.7 | -62.4 | 259.7 | 140.1 | 105.8 | 0.56 | 0.72 | 85.2 |
| random NS | -84.6 | -62.9 | 266.4 | 146.3 | 109.1 | 0.61 | 0.76 | 88.5 |

Table A14: **Ablation on Design Choices.** We conduct ablation studies across a diverse set of tasks, including: Cheetah-Wind-S with a planner-based approach (denoted as Cheetah-1 in the table), Cheetah-Wind-S with a diffusion policy (Cheetah-2), Ant-Dir-E (policy, IDQL-based), Maze2D-Large (planner), Walker2D-Medium-Expert (planner), Kitchen-Partial (planner), LIBERO-Object (diffusion policy), and RoboMimic-Can.

