# OpenReview forum: "Ada-Diffuser: Latent-Aware Adaptive Diffusion for Decision-Making"
_ICLR.cc/2026/Conference — ICLR 2026 Poster_

### Official Review · Reviewer_5j5o · 2025-10-24

**Soundness:** 3
**Presentation:** 3
**Contribution:** 2
**Rating:** 4
**Confidence:** 3

**Summary:**

This paper introduces a unified framework that explicitly incorporates latent dynamic inference into generative decision-making, particularly focusing on partially observable environments. The authors highlight that existing generative models (like Diffusion Models) often overlook evolving latent factors essential to environment transitions, reward structures, and high-level agent behavior. Algorithmic Innovations include causal autoregressive denoising schedule, denoise-and-refine training procedure and zig-zag sampling for inference. Extensive experiments across locomotion, navigation, and robotic manipulation benchmarks show improved performance in accurate latent inference, long-horizon planning, and adaptive policy learning compared to diffusion baselines and variants augmented with existing latent modeling techniques.

**Strengths:**

1. Originality. The paper demonstrates high originality, primarily through the unique integration of theoretical identifiability results into the architecture of a causal diffusion model. Specifically, establishing sufficient conditions under which latent factors can be identified from short temporal windows of RL trajectories is a novel theoretical contribution. This theoretical insight directly guides the algorithmic design, leading to the development of the minimal-sufficient block concept and the backward refinement step for online latent recovery, which contrasts with prior latent-augmented diffusion approaches. The combination of autoregressive denoising, latent factor identification, and the tailored zig-zag sampling scheme represents a creative and non-trivial synergy of existing ideas applied to sequential decision-making.

2. Quality. The quality of the work is high, supported by rigorous technical details and thorough empirical validation. The identification theory provides a strong foundation for the method. The experimental evaluation is comprehensive, covering eight environments across 23 different settings, including latent factors affecting dynamics and rewards, latent actions from action-free data, and environments without explicit latents. Furthermore, the paper provides detailed ablation studies confirming the effectiveness of the key design choices, particularly the refinement and zig-zag sampling mechanisms, which significantly reduce probing MSE and improve performance.

3. Clarity. The paper is well-structured. The framework's modular design, dividing the process into latent factor identification (Stage 1) and the causal diffusion model (Stage 2), enhances clarity. The authors clearly define the Latent Contextual POMDP with Time-dependent Context and formalize the data generation process using Structural Causal Models (SCMs). The purpose and justification for complex components like the latent identification loss and zig-zag sampling are explicitly tied back to the theoretical findings (Theorem 1), providing a clear narrative thread.

4. Significance. The significance of Ada-Diffuser lies in its ability to handle time-varying, unobserved latent factors which are pervasive in real-world applications such as robotics and autonomous driving. By providing a unified generative framework for planning and policy learning that effectively adapts to these latent variations, the method offers substantial improvements over state-of-the-art diffusion baselines and meta-RL approaches in settings with explicit latent factors. Moreover, the finding that latent modeling can improve performance even in environments without explicit latent factors suggests broad utility beyond traditionally defined POMDPs.

**Weaknesses:**

1. Mismatch between Theoretical and Practical Block Sizes: The central theoretical result (Theorem 1) establishes that four consecutive time steps (x_t−2:t+1) are sufficient for latent identifiability. However, the authors state that in practice, they do not strictly limit the block size to four, finding that slightly larger blocks (typically between 6 and 20 steps) lead to more stable optimization and better performance. This practical necessity for larger blocks suggests that the assumptions required for the theoretical four-step result may be too stringent or idealized for the complexity of the empirical environments and the parametric function approximators used. The reliance on large blocks somewhat diminishes the immediate algorithmic guidance derived from the "minimal-sufficient block" result. This adds to the suspicion that the authors reverse-engineered the justification, deliberately finding an obscure mathematical theorem to back a model they already wanted, instead of allowing the model to emerge naturally from the theory.

2. Dependency on Future Observations during Online Inference: The latent identifiability theory relies on incorporating future observations (x_t+1) to form an accurate posterior q_ψ(c_t∣x_{t−T_x:t+1}). In online inference, future observations are unavailable, creating a mismatch. The zig-zag sampling scheme attempts to resolve this by conditioning on a predicted next noisy step x_{t+1}^{k_2}​ to refine \hat{c}_t^post. This means the quality of the online latent inference hinges critically on the accuracy of the diffusion model's prediction of x_{t+1}^{k_2}. If the future prediction is inaccurate, the latent refinement may degrade, potentially leading to cascading errors in long-horizon planning. This key dependency is not fully analyzed or quantified in the provided sources, particularly in terms of how errors propagate when using the noisy future estimate.

3. Increased Model Complexity and Training Stage Separation: Ada-Diffuser requires training two distinct stages: (1) an offline VAE for latent factor identification (optimizing the ELBO loss), and (2) the causal diffusion model (optimizing along with the contrastive loss L_d−r). This two-stage training process is inherently more complex and potentially less robust than end-to-end training. While the authors claim the VAE introduces minimal computational overhead, the overall architectural complexity, including the GRU encoder, multiple MLPs for priors/posteriors, and the explicit contrastive loss L_rel demands careful tuning of multiple hyperparameters (λ_KL, λ_prior, λ_rel, m).

4. Novelty of Component Integration: While the overall framework is original, the individual components draw heavily from existing literature. The concept of latent belief state learning in POMDPs is standard (e.g., LILAC, DynaMITE, VAE usage). The autoregressive noise schedule is derived from recent autoregressive diffusion models (e.g., DF). The novelty rests heavily on the specific way the blocks, refinement, and zig-zag work together. However, a deeper analysis justifying why this specific combination is necessary—rather than sufficient—to satisfy the identifiability constraints (especially compared to simpler VAE-in-loop approaches like DynaMITE/LILAC integrated with diffusion) would strengthen the contribution.

**Questions:**

- Questions on Identifiability and Online Inference

1. Causal Timeline and Lookahead in Zig-Zag Sampling: During online inference (zig-zag sampling), the refined latent \hat{c}_t^post is conditioned on x_{t+1}^{k_2}. In a standard sequential decision-making loop, s_t+1 (and thus x_t+1) is only observed after the action a_t has been executed. Please clarify the precise temporal availability of the conditioning inputs x_t^0 x_t^{k_1} x_{t+1}^{k_2} used to calculate \hat{c}_t^post during online generation of a_t. Does x_{t+1}^{k_2} represent a predicted state from the generative model itself? If so, this suggests \hat{c}_t^post is recursively dependent on the model's own (noisy) future prediction, which could introduce instability or drift, especially for long planning horizons (e.g., T_p=32).

2. Practical Implication of Block Size (Beyond Stability): The ablation shows that performance degrades with blocks that are too small (≤4) or too large (>20). Given that the theory proves 4 steps are sufficient, what specific phenomena in the practical optimization environment (e.g., approximation error, high variance) necessitate using blocks in the range of 6 to 20 steps, rather than the theoretically suggested minimum? Does this suggest a fundamental limitation of using finite-capacity function approximators (like the GRU/MLP encoder) to capture the non-parametric identifiability conditions (Assumptions 2 and 3)?

- Questions on Algorithmic Design

3. Function and Tuning of the Contrastive Improvement Loss (L_{rel}): The refinement stage uses a contrastive improvement loss $L_{\text {rel }}=$ softplus $\left(\log L_{\text {post }}-\log s g\left(L_{\text {prior }}\right)+m\right)$ to encourage the posterior latent (\hat{c}_post) to produce better reconstructions than the prior latent (\hat{c}_prior). This is crucial for avoiding posterior collapse, a common issue in VAE-based latent modeling. What is the impact of the margin hyperparameter m and the weighting coefficient λ_rel on the training process, particularly regarding the trade-off between maximizing reconstruction accuracy (low L_post) and maintaining latent expressiveness? Have the authors performed an ablation on m to demonstrate its necessity?

4. Impact of Latent Dimensions: The details provided focus on the architecture (GRU encoder, MLP decoders) but not the specific dimensionality of the inferred latent context C for each environment. How does the choice of latent dimension d_c affect: (i) the satisfaction of the identifiability assumptions (e.g., Assumption 3 requires sufficient variability induced by c), and (ii) the required size of the minimal block T_x? Since the model captures high-level factors, rewards, or even latent actions, the complexity of c likely varies widely.

- Clarification on the Optimization Roles of the Denoise-and-Refine Losses

5. Parameter Update Paths: Please clarify which network parameters (θ for ϵ_θ, ψ for q_ψ, or ϕ for p_ϕ) are updated by which loss terms. Does minimizing L_post (reconstruction using the posterior latent c_post) update only θ, or does it also backpropagate through c_post to update the posterior encoder ψ? Similarly, does minimizing L_prior update θ and the prior network ϕ?

6. Role of Stop-Gradient in L_{rel}: The contrastive loss L_rel uses a stop-gradient on L_prior(logsg(L_prior)). This typically means L_rel is designed to primarily improve the posterior network q_ψ by pushing c_post towards a representation that achieves better reconstruction compared to a frozen reconstruction accuracy baseline derived from the prior c_prior. Please confirm whether the optimization signal derived from L_rel is used solely to update the posterior encoder ψ, or if it also affects the denoising network θ.

- Questions on Latent Correlation and Prior Quality

7. Sensitivity to Latent Dynamics: The prior p_ϕ(c_t∣c_t−1) relies on the Markovian assumption of the latent context c. In cases where the latent dynamics are highly stochastic or non-Markovian (i.e., c_t is weakly dependent on c_t−1), the prior estimate c_prior may be very poor. Does the overall framework, particularly the contrastive loss L_rel, remain effective when the latent temporal correlation is weak, or is Ada-Diffuser implicitly relying on environments where the latent context exhibits strong temporal smoothness?

8. Prior Collapse Mitigation: Since Stage 1 is an offline VAE optimization using L_ELBO, the term DKL(q_ψ||p_ϕ) serves to regularize the posterior towards the prior. Given the introduction of L_rel in Stage 2 (Causal Diffusion Model), which aims to make the posterior estimate significantly better than the prior estimate, how do the authors ensure that this competitive pressure does not induce posterior collapse (where the VAE minimizes the KL term to zero, ignoring the observations) in Stage 1, or that the training stages remain balanced?

- Detailed Questions on Autoregressive Denoising and Complexity

9. Sequential Steps vs. Total Diffusion Steps: Standard diffusion models perform K noise reduction steps. The autoregressive denoising process described appears to perform a sequential denoising across T time steps. If the trajectory length T is large (e.g., T=32), does the overall inference procedure involve T×K total denoising evaluations, or is the process structured such that the total number of ϵ_θ evaluations remains close to the standard K? Specifically, how does the sequential nature described in Equation (2) translate into the number of required calls to ϵ_θ(x^k, k, c)?

10. Computational Overhead of Zig-Zag: The ablation shows that using a Picard-accelerated variant reduces inference time (e.g., 1.51s → 1.15s for Cheetah). Given that the zig-zag sampling scheme requires an additional "update" step where the posterior latent c_post is calculated using a predicted noisy next step x_{t+1}^{k_2}, what is the approximate percentage of the total inference time consumed specifically by the latent inference and refinement steps (non-denoising operations) in the default zig-zag sampler?

**Details Of Ethics Concerns:**

No.

---

> ### Author Response · Authors · 2025-11-23
> **Author Response - Part 1**
>
> Thank you for your thoughtful review. We are encouraged by your positive assessment of the originality, quality, clarity, and significance of our work, and we appreciate the detailed comments you provided. Below, we discuss each point and summarize the related revisions.
>
> ---
>
> > W1: Mismatch between Theoretical and Practical Block Sizes
>
> **R1**: Thank you for raising this point. As discussed in `Sec. 5.2` and `Q8` in `Appendix A.4`, we would like to clarify that Theorem 1 provides a sufficient identifiability condition in principle, rather than a hard architectural constraint. Empirically, we indeed find that using slightly larger blocks (typically 6–20 steps) improves optimization stability and performance, which is expected in finite-sample, function-approximation regimes. Importantly, our ablations in `Fig. 4(a)` and `Fig. A.5` show a pattern that is fully consistent with the theory: removing future steps substantially harms identifiability and policy performance, while overly long blocks introduce redundancy and again degrade results. Thus, the key takeaway from Theorem 1 remains intact: future observations are essential for identification, and the required block need not be long.
>
> More concretely, Theorem 1 guides our algorithm in three ways. First, it motivates small, future-inclusive temporal blocks: while the minimal sufficient length is four in theory, the practical optimum can be slightly larger depending on task complexity and model capacity, but it should remain short and include future steps. Second, it motivates how latents are embedded into diffusion: we design (i) an autoregressive noise schedule aligned with temporal causal structure with latent factors (reflecting `Fig 1`), (ii) a denoise-and-refine procedure that alternates between denoising and latent posterior updates (so the model explicitly learns to exploit future-conditioned information by the backtracking refinement), and (iii) a zig–zag sampling strategy that approximates future-inclusive blocks online when true futures are unavailable. Also, diffusion’s multi-step denoising naturally aligns with the identifiability requirement by providing progressively refined “soft future’’ predictions for posterior inference.
>
> In summary, our theory is not reverse-engineered: it provides a principled sufficient condition that explains why small, future-aware blocks work, when they fail (redundant or history-only blocks), and how to design the diffusion–latent coupling to respect these conditions in practice.
>
> ---
>
> > W2: Dependency on Future Observations during Online Inference.
>
>
> **R2**: Yes, as you noted, the zig–zag procedure is designed to mitigate this train–test mismatch. We address this concern from both an algorithmic and an empirical perspective.
>
>
>
>
> **Algorithmic perspective.**
> The refinement mechanism relies on three synergistic design choices:
> 1. **Contrastive improvement loss during training.**
>  When true future observations are available, we enforce a margin between the posterior network (conditioned on future steps) and the prior network (with stop-gradient for stability). This explicitly teaches the model *how* future observations improve latent recovery.
> 2. **Appending a clean predicted future at test time.**
>  During zig–zag refinement, we only append a lightly noised predicted future $x\_{t+1}^{k_2}$, which reduces error propagation compared to using a heavily noised or fully generated future.
> 3. **Global diffusion matching as an (implicit) regularizer.**
>  The main diffusion objective still performs distribution matching over full trajectories conditioned on the latent context, helping absorb residual mismatch from predicted futures.
>
>
> These mechanisms do not remove the theoretical gap entirely, but they directly target the source of the mismatch and provide practical optimization mitigation.
>
>
> **Empirical perspective.**
> In addition to policy-level ablations, we evaluate latent recovery quality directly. Below is the probing MSE (lower is better) for recovering the ground-truth latent:
>
> | **Variant**        | **MSE** |
> |--------------------|---------|
> | Oracle             | **0.12** |
> | Full (ours)        | 0.18    |
> | w/o refinement     | 0.28    |
> | w/o zig–zag        | 0.23    |
>
> Removing refinement or zig–zag significantly worsens latent recovery. Moreover, the gap between the oracle model (0.12) and our full method (0.18) is small (0.06), indicating that predicted futures are sufficiently informative for reliable posterior alignment in practice. We believe this is the way we can quantify the exact posterior error incurred by using predicted futures.
>
>
> In summary, our design explicitly mitigates this mismatch, and the probing results show that zig–zag refinement materially improves latent identifiability and downstream performance.
>
>
> **Related Revision**: We move the probing for zig-zag sampling from the appendix to the main paper (`Sec 5.3`, Page 10).
>
> ---

---

> ### Author Response · Authors · 2025-11-23
> **Author Response - Part 2**
>
> > W3: Increased Model Complexity and Training Stage Separation
>
> **R3**: We would like to mention that our original submission already includes compute and latency comparisons in `Appendix I.1`, where we report both **training time** and **inference cost per trajectory rollout**. However, based on Reviewer `ZU5m`'s suggestion, we have replaced rollout-time reporting with **latency per inference step** and provide a more explicit comparison against strong diffusion baselines. The updated compute results are shown below.
>
> - **Training Time and Inference Latency**
>
> We compare our method against Decision Diffuser (DD) and Diffusion Policy (DP) across all environments.
>
> | Environment | Training (sec/epoch) Ours / DD | Ours / DP | Latency (ms) Ours/DD|Ours/DP|
> |-|--|--|---|--|
> | Cheetah | 72.1 / 60.1 (1.20×) | 69.8 / 58.4 (1.20×) | 182 / 114 (1.16×)| 160/125 (1.28×) |
> | Ant     | 79.5 / 64.3 (1.24×) | 76.0 / 62.0 (1.23×) | 148 / 125 (1.19×) | 172/139 (1.24×) |
> | Walker  | 85.3 / 67.1 (1.27×) | 81.5 / 64.2 (1.27×) | 182 / 144 (1.28×) | 170 / 130 (1.31×) |
> | Maze2D  | 90.2 / 72.0 (1.25×) | 88.3 / 69.2 (1.28×) | 184 / 149 (1.24×) | 196 / 152 (1.29×) |
> | Libero  | 104.0 / 81.0 (1.28×) | 102.1 / 78.0 (1.31×) | 209 / 169 (1.24×) | 219 / 162 (1.35×) |
> | Kitchen | 117.8 / 88.1 (1.34×) | 115.3 / 85.0 (1.36×) | 228 / 180 (1.27×) | 211 / 168 (1.26×) |
>
> Across all settings, our framework introduces only a **moderate overhead**: typically **1.2–1.3× runtime**, corresponding to about **20–30% additional training time and latency**.
>
> In all experiments, we set them as: $\lambda\_\text{KL}=0.01$; $\lambda\_\text{rel}=0.1$; $\lambda\_\text{prior}=0.1$; $m$ is set to be the $0.05 \times \mathcal{L}\_\text{prior}$ during the beginning of each epoch. These have been specified in `Appendix 2.1` and `Appendix 2.4`.
>
> **Related Revisions**
>
> - We updated all inference-time reporting to **latency (ms)** as suggested.
> - Please refer to `Appendix I.1` (`Table A12–A13`) for the full updated compute profile.
> - We also added a pointer to these results in `Sec. 5.3` (Page 10) of the main text.
>
> ---
>
> > W4: Novelty of Component Integration
>
> **R4**: We would like to say that this is not an pure integration first. The whole framework, which captures the causal generative process of RL trajectories with latent variables, as formalized by the SCM in Fig. 1(a). Guided by Theorem 1, our goal is to ensure that the diffusion model not only learns to generate trajectories but also identifies the true latent factors driving them. The model must capture: (1) **temporal dependencies**, (2) on **both observable and latent variables**, and (3) **conditions for identifiability**.
>
> - **Algorithm Design**
>
>  - *Autoregressive Denoising*: our model temporal dependencies over both observable and latent variables, we adopt an autoregressive diffusion model with noise scheduling . Each step $x_t$ is denoised while conditioning on partially denoised past steps (cleaner than the variables at $t$) and inferred latent variables from short temporal blocks, enabling structured temporal-latent modeling.
>
>  - *Backward Refinement*: As you mentioned, to identify the latent context $c_t$, whose posterior depends on future observations (e.g., $x_{t+1}$, **guided by Theorem 1**), we introduce a backward refinement step. We first denoise $x_t$ into a partial state $x_t^{k_1}$ using $x_{t-1}^0$ and an initial estimate of $c_t$ sampled from prior, then refine $c_t$ with $x_{t+1}$ to obtain the final $x_t^0$. During training, we enforce this backward refinement to satisfy the theoretical identifiability conditions. At inference time (zig-zag sampling), we substitute $x_{t+1}$ with a predicted estimate.
>
> ---
> **Comparison with Baselines**:
> - Diffuser, DP,  IDQL, DD do not model or infer latent context.
> - Meta-Diffuser, LILAC, and DynaMITE learn latents via meta-learning but lack our autoregressive and refinement structure. And IDQL and LDP only consider latent actions/skills. None of these used the minimal or sufficient block for latent inference.
> - DF includes autoregressive denoising but does not identify latent contexts.
>
> |Method|Latent Factors|AR Denoising|Min. & Suff. Obs|
> |-|-|-|-|
> |Ours|Yes (dyn., rew., act.)|Yes|Yes (refine, zig-zag)|
> |Diffuser/DP/DD/IDQL |No| No|No|
> |DF|No|Yes|No|
> |Meta-Diff./LILAC/DynaMITE|Yes (dyn., rew. only)|No|No|
> |LDCQ|Yes (hi-level act.)|No|No|
> |LDP|Yes (hi-level act.)|No|No|
>
>
> To sum up, to the best of our knowledge, **no** prior work has modeled the full causal generative process within diffusion models as we do, nor has any existing method explicitly considered the identifiability of the latent posterior through a “backward” refinement mechanism. Furthermore, our algorithmic design, including the **denoise-and-refine mechanism, zig-zag sampling, and fully autoregressive latent-observation modeling**, is, to our knowledge, novel to the field. In our ablation (`Sec 5.3`), we verified and analyzed the individual benefits from all these components.
>
> --

---

> ### Author Response · Authors · 2025-11-23
> **Author Response - Part 3**
>
> > Q1: Causal Timeline and Lookahead in Zig-Zag Sampling
>
> **R5**: Part of it overlaps with W2. Here we do a quick follow-up after **R2** to clarify more on the actions.
>
> First, you are correct that in a standard control loop, $x_{t+1}$ is only observed after executing $a_t$. We do not violate this. The zig–zag refinement uses **a predicted future observation** $x_{t+1}^{k_2}$ produced by the diffusion model itself at a lower noise level.
> So this predicted step is used only for *latent refinement*, not as an actual state for environment interaction.
>
> In terms of "actions", where are two cases depending on the diffusion backbone:
>
> #### **(1) Diffusion Planner (state-trajectory generation)**
> - The model generates an entire *state* trajectory first:
>  $(x_t, x_{t+1}, \ldots, x_{t+T_p})$.
> - Zig–zag operates **within this generated trajectory**, refining latent $\hat{c}_t$ using the denoised prediction $x\_{t+1}^{k_2}$ inside the diffusion loop.
> - After the whole refined trajectory is obtained, we retrieve actions using a separate inverse-dynamics model (standard practice in Diffuser planners).
>
>
> Thus, for planners, zig–zag never uses unobserved real future state information—only the model-generated future that is already part of the diffusion sample.
>
> #### **(2) Diffusion Policy (action-trajectory generation)**
> - The model generates **future actions** at progressively lower noise levels.
> - These noisy predicted future actions correspond to *implicit* predicted future states via the model dynamics encoded in the diffusion process.
> - During zig–zag, we refine $\hat{c}_t$ using this partially denoised predictive information.
> - Finally, the current action $a_t$ is generated *after* latent refinement, so the process is always causally aligned.
>
>
> Importantly, diffusion policies operate in a **chunked autoregressive** manner: the model always generates future predictions *before* deciding the immediate action. This ensures no lookahead violation occurs.
>
>
> Hence, although zig–zag uses a predicted $x_{t+1}^{k_2}$, this is fully consistent with normal generative-model rollouts:
>
> - The model predicts (noised) futures first.
> - Then uses those predictions to refine its latent belief.
> - Then produces the actual action sent to the environment.
>
>
> No real future information is leaked or accessed.
>
>
> ---
>
>
> > Q2: Practical Implication of Block Size (Beyond Stability)
>
>
> **R6**: This question overlaps with W1, here we add some follow-ups. In theory, four steps are *sufficient* for identifiability, but this does not imply that four steps are always *optimal* under finite-sample and finite-capacity conditions. In practice, using a slightly larger block (6–20 steps) reduces approximation error in two ways:
> (i) it provides more temporal evidence for the encoder to denoise high-variance observations, and
> (ii) it improves the stability of the posterior network under GRU/MLP parameterization.
>
>
> We view this not as a limitation of the theory, but as a consequence of practical function approximation: the identifiability result is non-parametric and assumes access to exact conditionals, while neural networks must learn these relations from noisy data. Thus, using small-but-slightly-larger blocks improves robustness, whereas overly long blocks reintroduce redundancy that harms performance. In this sense, the need for 6–20 steps reflects standard finite-capacity approximation effects rather than a fundamental theoretical mismatch.
>
> ---
>
> > Q3: Function and Tuning of the Contrastive Improvement Loss $\mathcal{L}_\text{rel}$
>
>
> **R7**: Yes, as you pointed out, $\mathcal{L}\_{\text{rel}}$ is crucial for avoiding posterior collapse, a common issue in VAE-based latent modeling. We implement it as a **relative margin** to keep the scale comparable across tasks. Concretely, at the beginning of each epoch we set the margin to $(0.05 \times \mathcal{L}\_{\text{prior}})$. This works well in practice because $\mathcal{L}\_{\text{prior}}$ becomes stable in later training, yielding a simple and effective trade-off between reconstruction accuracy and latent expressiveness.
>
> Following your suggestion, we conducted an ablation over the margin ratio. We sweep
> ${0.01, 0.05, 0.08, 0.1, 0.2, 0.5, 0.8\}$
> and evaluate the probing MSE between the learned latent and the ground-truth wind speed in Cheetah:
>
> | Margin ratio | Probing MSE |
> |---:|--:|
> |0.01|0.16|
> |0.05|0.18|
> |0.08|0.22|
> |0.10|0.21|
> |0.20|0.25|
> |0.50|0.31|
> |0.80|0.42|
>
>
> We observe that ratios below $\sim 0.1$ give similar probing errors, indicating the method is not overly sensitive in a reasonable range. However, large ratios (e.g., $\ge 0.5$) degrade latent quality and training stability. Intuitively, an overly strong margin forces the posterior to diverge too aggressively from the prior, causing the model to overfit to future-conditioned signals and weakening general latent expressiveness. Overall, this supports our default choice as both stable and effective.
>
> ---

---

> ### Author Response · Authors · 2025-11-23
> **Author Response - Part 4**
>
> > Q4: Impact of Latent Dimensions
>
> **R8**: Thank you for raising this point. Here we choose a slightly overparameterized latent dimension:
> - $d_c = 20$ for Cheetah, Walker, and Maze2D;
> - $d_c = 64$ for robotic manipulation (Kitchen, RoboMimic, LIBERO).
> These choices are uniform across tasks and not tuned per environment.
>
> **(i) Relation between latent dimension and identifiability assumptions.**
> Assumptions 2 and 3 require that different latent values induce distinguishable variation in the transition distributions. These are *properties of the environment*, not of the dimensionality of the parameterization. In practice, a larger $d_c$ merely provides more representational capacity for the encoder; it does **not** change whether the true environment dynamics satisfy the identifiability conditions.
>
> However, excessively large $d_c$ increases approximation difficulty and introduces redundancy, which can *harm* the encoder’s ability to map observations into a consistent latent space, effectively reducing identifiability in practice even if the environment itself satisfies the assumptions. Moderate dimensionality (0.5×–2× of our default) generally behaves similarly.
>
> **(ii) Relation between latent dimension and minimal block size $T_x$**
>
> The theoretical block-size requirement depends only on the causal structure (past–future information), not on the dimensionality of $c$. A larger latent dimension does not change the fact that future observations are necessary, nor the sufficiency of a small temporal block. In practice, higher-dimensional latents may require slightly more temporal context for stable optimization, but our ablations show the effect is mild.
>
>
> **Latent-dimension ablation.**
> We report additional ablation results below, evaluating the effect of scaling the latent dimension:
>
>
> | **Latent dim** | Orig. | 0.5× | 2× | 4× | 6× |
> |-------------------|-------|------|----|----|----|
> | Cheetah-wind-s | **–73.5** | –85.2 | –77.6 | –89.5 | –102.4 |
> | LIBERO | **93.4** | 89.3 | 89.4 | 87.6 | 85.0 |
>
>
> These results show that:
> - **0.5×–2×** latent dimensionality has a limited impact on performance.
> - **Overly large latent dimensions (4×–6×)** lead to redundancy and heavier optimization burden, which degrades performance and weakens identifiability in practice.
>
>
> Overall, while the theoretical assumptions depend on environmental distinguishability, not latent dimensionality, our ablations demonstrate that moderate latent sizes are robust, and extreme overparameterization is the only case that hurts performance.
>
>
> **Related Revisions**: We highlighted the choice of dimensionality in `Appendix D.2.1` and added the ablation in `Sec 5.3` (Page 10).
>
> ---
>
> > Q5: Parameter Update Paths
>
>
> **R9**: Thank you for the question—this is an important detail. The parameter update paths are as follows:
>
>
> - $\mathcal{L}\_\text{prior}$ updates **only** the parameters of the *prior encoder* $p_\phi(c_t \mid x_{t-2:t})$.
>  Gradients **do not** flow into the denoising network $\epsilon_\theta$.
>
>
> - $\mathcal{L}\_\text{post}$ updates **only** the parameters of the *posterior encoder* $q_\psi(c_t \mid x_{t-2:t+1})$.
>  As designed, gradients are **not** backpropagated through $c_t^{\text{post}}$ into $\epsilon_\theta$.
>
>
> -  $\mathcal{L}\_\text{rel}$  (contrastive improvement loss) updates only $\psi$.
>  Again, gradients do **not** reach $\theta$.
>
>
> - **$\mathcal{L}_\text{diff}$** (the main diffusion denoising objective) updates **only** the parameters of the denoiser $\epsilon_\theta$.
>  The latent $c_t$ is treated as a conditioning variable, and the gradient does **not** flow backward into $q_\psi$ or $p_\phi$.
>
>
> Hence, the update paths are cleanly separated:
>
>
> | Loss term | Updates $\epsilon_\theta$ | Updates $q_\psi$ | Updates $p_\phi$ |
> |----------|------------------------------|----------------------|----------------------|
> | $\mathcal{L}_\text{diff}$ | ✓ | ✗ | ✗ |
> | $\mathcal{L}_\text{post}$ | ✗ | ✓ | ✗ |
> | $\mathcal{L}_\text{prior}$ | ✗ | ✗ | ✓ |
> | $\mathcal{L}_\text{rel}$ | ✗ | ✓ |  ✗ |
>
>
> **Related Revision**: We have emphasized this in `Sec 4.3` (Page 7).
>
> ---
>
> > Q6: Role of Stop-Gradient in L_{rel}
>
>
> **R10**: Yes. This is because $\mathcal{L}\_{\text{rel}}$ applies a stop-gradient to the prior branch, the optimization signal is used **only to update the posterior encoder $q_\psi$** (and, in our implementation, also the prior encoder $p_\phi$ through the margin term). Importantly, $\mathcal{L}_{\text{rel}}$ **does not update the denoising network $\theta$**.
>
> ---

---

> ### Author Response · Authors · 2025-11-23
> **Author Response - Part 5**
>
> > Q7: Sensitivity to Latent Dynamics
>
>
> **R11**: Beyond smooth latent changes, our paper already evaluates some cases with weaker temporal depence on latents. Specifically, we consider *episodic abrupt changes* (denoted “E”), where the latent context (dynamics or rewards) changes sharply only at the **beginning of each episode**. This setting requires online adaptation at episode start, and the results are reported below.
>
>
> **Episodic abrupt changes (E).**
>
>
>
>
> | Environment | Diffuser | DF | DF + DynaMITE | DF + LILAC | MetaDiffuser | Ours |
> |------------|----------|----|---------------|------------|--------------|------|
> | Cheetah-Wind-E $c^s$ | –120.4 ± 12.7 | –105.8 ± 9.6 | –82.3 ± 8.2 | –91.5 ± 7.8 | –95.3 ± 7.4 | **–68.9 ± 7.6** |
>
>
> ---
>
>
> **Within-episode abrupt changes.**
> As a more challenging generalization test, we additionally evaluate **within-episode** with sharpe shifts. Concretely, we modify the wind to evolve as
> $f_w = k + 5 sin(0.5 t)$,
> where $k$ changes abruptly every 20–40 steps. In this case, we trigger online adaptation whenever the VAE reconstruction error rises above a threshold; we then update the latent model using a small buffer of 50 recent samples before continuing. We compare against the most relevant latent-context baselines:
>
>
>
>
> | Environment | DF + DynaMITE | DF + LILAC | MetaDiffuser | Ours |
> |------------|----------------|------------|--------------|------|
> | Cheetah-Wind (within-episode abrupt) | –119.2 ± 15.2 | –123.4 ± 11.7 | –136.3 ± 9.2 | **–108.5 ± 10.6** |
>
>
>
>
> Under this case, Ada-Diffuser remains more robust than prior latent-context diffusion baselines, showing that our latent refinement mechanisms enable fast online adaptation without full retraining.
>
>
> Overall, these two stress tests clarify that our method does not rely on purely smooth latent temporal evolution: it can handle both episodic and within-episode abrupt changes (under these cases, for many time steps, $c_t$ is weakly dependent on $c\_{t−1}$ through lightweight online latent updates.
>
>
> ---
>
>
> > Q8: Prior Collapse Mitigation
>
>
> **R12**: In practice, two mechanisms prevent this. First, Stage 1 is warmed up allow both the prior and posterior encoders to reach a stable regime before Stage 2 begins. Second, during Stage 2, the prior $p_\phi$ continues to be updated, so the margin in $\mathcal{L}\_{\text{rel}}$ does *not* force the posterior to dominate a fixed prior; instead, the two distributions co-adapt.
>
>
> ---
>
>
> > Q9: Sequential Steps vs. Total Diffusion Steps
>
>
> **R13**: This is not $T \times K$.
> For a trajectory of length $T$, our model does **not** run a full $K$-step denoising chain for every time index. Instead, we assign a **single noise level per time step**, monotonically increasing along the horizon:
> $
> \{k_1,\dots,k_T\}, \quad k_i=\frac{i}{T}K.
> $
> where $i=1,\dots,T$
> Then each state (or state–action pair) $x_i$ is denoised **once** at its designated level $k_i$. Therefore, one autoregressive denoising sweep requires about **$T$ calls** to $\epsilon_\theta(x^{k_i}_i, k_i, c)$, not $T\times K$.
>
>
> In other words, Eq. (2) describes a *sequential ordering over time*, while the diffusion “depth” is spread across the horizon via different $k_i$’s, keeping the total number of $\epsilon_\theta$ evaluations close to **standard diffusion cost** linear in $T$. Any extra cost only comes from the small number of refinement/zig–zag passes, which adds a constant-factor overhead rather than a multiplicative $K$.
>
>
> > Q10: Computational Overhead of Zig-Zag
>
>
> **R14**: We tested the inference pipeline and found that the latent refinement step (posterior evaluation + zig–zag update) contributes approximately 20–30% of the total inference time in the default sampler. This is given in the planning-latency overhead reported in **R3**, where the additional cost comes almost entirely from running the latent-identification network once per refinement cycle. So, the denoising network still dominates the overall compute, while zig–zag adds only a moderate constant-factor overhead that is further reducible via the Picard-accelerated variant.
>
>
> ---
>
>
> Again, we thank you for the fruitful comments, which helped us strengthen the work, and we would be very happy to continue the discussion if any further questions arise.

---

> > ### Comment · Reviewer_5j5o · 2025-11-27
> >
> > I thank the authors for their thorough response and for addressing my questions. I change to positive attitude of the work and increase my score.

---

> > > ### Author Response · Authors · 2025-11-27
> > >
> > > Thank you again for your thoughtful review and the really fruitful questions. We are glad the responses made sense. If any other question comes up, feel free to let us know. We are always happy to discuss further.

---

### Official Review · Reviewer_DUvX · 2025-10-26

**Soundness:** 3
**Presentation:** 3
**Contribution:** 2
**Rating:** 6
**Confidence:** 3

**Summary:**

The paper proposes Ada‑Diffuser, a diffusion‑based framework for decision‑making that explicitly models time‑varying latent factors influencing dynamics and/or rewards. Under a set of injectivity conditions, the posterior over the latent at time t is identified from a short temporal window. This motivates block‑wise latent identification via a VAE prior/posterior with an ELBO and a causal, autoregressive diffusion model.

**Strengths:**

1. Incorporating explicit latent context into diffusion-based decision models addresses an important gap in decision making under uncertainty. The paper presents a unified approach for planning and policy learning.

2. The identifiability result provides a principled justification for blockwise latent inference and motivates a denoise-then-refine design. The proof strategy follows the nonparametric identification literature and is clearly presented in the appendix.

3. The empirical evaluation is broad, spanning (i) latents in dynamics and reward, (ii) latent actions and action-free settings, and (iii) settings without explicit latents.

**Weaknesses:**

1. The theorem uses $x_{t+1}$ (a future observation) for identifiability, but test-time planning cannot access the future. The paper proposes a zig-zag procedure that uses a predicted future observation during refinement. While ablations show empirical gains, the method would be challenging to quantify the error incurred by using predicted futures for posterior alignment.

2. The discussion of the effectiveness of latents in environments without explicitly designed latents is not fully convincing. Doesn’t expert action noise still imply deterministic dynamics? How does the latent help a policy learned from datasets collected under stochastic behavior policies?

**Questions:**

1. Can you provide an example of real-world scenarios indicating when Assumptions 2–3 plausibly hold (or fail)?

---

> ### Author Response · Authors · 2025-11-23
> **Author Response - Part 1**
>
> Thank you for the thoughtful review and for your recognition of our method, theory, and empirical validations. Below, we address each comment and question point by point.
>
> ---
>
> > W1: The theorem uses $x_{t+1}$  (a future observation) for identifiability, but test-time planning cannot access the future. The paper proposes a zig-zag procedure that uses a predicted future observation during refinement. While ablations show empirical gains, the method would be challenging to quantify the error incurred by using predicted futures for posterior alignment.
>
> **R1**: Yes, as you noted, the zig–zag procedure is designed to mitigate this train–test mismatch. We address this concern from both an algorithmic and an empirical perspective.
>
> **Algorithmic perspective.**
> The refinement mechanism relies on three synergistic design choices:
> 1. **Contrastive improvement loss during training.**
>   When true future observations are available, we enforce a margin between the posterior network (conditioned on future steps) and the prior network (with stop-gradient for stability). This explicitly teaches the model *how* future observations improve latent recovery.
> 2. **Appending a clean predicted future at test time.**
>   During zig–zag refinement, we only append a lightly noised predicted future $x\_{t+1}^{k_2}$, which reduces error propagation compared to using a heavily noised or fully generated future.
> 3. **Global diffusion matching as a regularizer.**
>   The main diffusion objective still performs distribution matching over full trajectories conditioned on the latent context, helping absorb residual mismatch from predicted futures.
>
>
> These mechanisms do not remove the theoretical gap entirely, but they directly target the source of the mismatch and provide practical optimization mitigation.
>
>
> **Empirical perspective.**
> In addition to policy-level ablations, we evaluate latent recovery quality directly. Below is the probing MSE (lower is better) for recovering the ground-truth latent:
>
>
> | **Variant**        | **MSE** |
> |--------------------|---------|
> | Oracle             | **0.12** |
> | Full (ours)        | 0.18    |
> | w/o refinement     | 0.28    |
> | w/o zig–zag        | 0.23    |
>
>
> Removing refinement or zig–zag significantly worsens latent recovery. Moreover, the gap between the oracle model (0.12) and our full method (0.18) is small (0.06), indicating that predicted futures are sufficiently informative for reliable posterior alignment in practice. We believe this is the way we can quantify the exact posterior error incurred by using predicted futures.
>
>
> In summary, our design explicitly mitigates this mismatch, and the probing results show that zig–zag refinement materially improves latent identifiability and downstream performance.
>
>
> **Related revision**: We move the probing for zig-zag sampling from the appendix to the main paper (`Sec 5.3`, Page 10).
>
> ---

---

> ### Author Response · Authors · 2025-11-23
> **Author Response - Part 2**
>
> > W2: The discussion of the effectiveness of latents in environments without explicitly designed latents is not fully convincing. Doesn’t expert action noise still imply deterministic dynamics? How does the latent help a policy learned from datasets collected under stochastic behavior policies?
>
>
> **R2**: Thank you for raising this important question. We address it from two angles: (i) why latent modeling remains useful even when the environment does not contain explicitly designed latent variables, and (ii) how latents help when datasets are generated by stochastic or suboptimal behavior policies.
>
>
> **(i) Stochastic behavior policies implicitly induce latent variability.**
> We agree with the reviewer that, under expert demonstrations, the underlying dynamics are still deterministic. However, in offline RL settings the *data-generating process* is typically stochastic due to suboptimal or mixed behavior policies. In these cases, variability in action choices, partial failures, and inconsistent strategies act as *implicit latent factors* that influence observed transitions. Our model does not assume a pre-specified partition of what latents affect (dynamics vs. rewards vs. actions). Instead, it flexibly identifies whichever latent variability is needed to explain the transitions.
>
>
> This is already reflected in our experiments.
> - In `RoboMimic`, we evaluate on both expert and **50% suboptimal** datasets (`Table A.3`).
> - In `D4RL`, we evaluate `Walker-medium` and `Walker-medium-replay` (`Table A.8`), where medium is a partially trained policy and medium-replay contains highly stochastic behavior.
> Across all these settings (none of which have explicit latent context), we consistently observe that latent modeling improves policy performance.
>
>
> As suggested by Reviewer ZU5m, we further tested **MimicGen**, which provides extremely diverse and noisy demonstration distributions. The results confirm the same trend.
>
>
> ---
>
>
> ### **Robustness to Noisy Observations (Square–Sawyer)**
> *(success rate / std)*
>
>
> | Setting                | Ours        | LDP         |
> |------------------------|-------------|-------------|
> | Square–Sawyer (clean) | 0.56 / 0.05 | 0.49 / 0.06 |
> | Square–Sawyer (noise) | 0.51 / 0.08 | 0.39 / 0.08 |
>
>
> Since MimicGen does not include noisy camera frames, we inject **temporal Gaussian noise**. Our latent-aware method remains robust, while LDP degrades substantially.
>
>
> ---
>
>
> ### **Suboptimal Demonstrations (Square–Panda)**
> *(success rate / std)*
>
>
> | Dataset quality | Ours        | LDP         |
> |-----------------|-------------|-------------|
> | better          | 0.70 / 0.09 | 0.62 / 0.06 |
> | okay            | 0.65 / 0.07 | 0.51 / 0.05 |
> | worse           | 0.59 / 0.08 | 0.49 / 0.08 |
>
>
> The gap is **largest under the “worse” setting**, where behavior is highly inconsistent—precisely when implicit latent variability is strongest.
>
>
> These results collectively support that latent identification improves robustness even without explicit environment latents, by capturing the hidden variability introduced by stochastic or suboptimal policy data.
>
> ---
>
> **(ii) Why latents help even when demonstrations come from near-deterministic expert policies and dynamics.**
> Even in expert settings with fully observed states and deterministic dynamics, the mapping from states to optimal actions is not always deterministic. For exmaple, some stochasticity arise from contact-rich or underactuated dynamics where tiny state differences induce different action or some behavior cloning noise or demonstration inconsistencies.
>
> Diffusion models can, of course, represent multi-modality (some level of stochasticity), but they must explain all action variability purely, which mixes distinct modes. Here, the latent provides a *Bayesian filtering–like* mechanism that organizes different sources of variability into coherent trajectories, which will make more stable planning.
>
>
> ---

---

> ### Author Response · Authors · 2025-11-23
> **Author Response - Part 3**
>
> > Q1: Can you provide an example of real-world scenarios indicating when Assumptions 2–3 plausibly hold (or fail)?
>
> **R3**: This will be a longer response, and it overlaps with concerns raised by other reviewers. We therefore carried out a careful check to support our intuition under these assumptions.
>
> We would like to discuss from both  **conceptual** and **empirical** perspectives. First, we explain why Assumptions 2–3 are mild in the context of our latent contextual POMDP setting and how they directly reflect the requirement that the latent context induces observable variation in trajectories. Second, we provide empirical evidence that these assumptions are satisfied in our benchmarks. We also discuss the relevant failure modes: when the assumptions are violated, it typically means the latent context has little or no effect on dynamics or rewards, in which case identifying the latent is neither possible nor necessary for policy learning. Finally, we clarify how these assumptions connect to RL.
>
> ---
>
> **Assumpation 2**: Conceptually, the injectivity of the operators $L$ simply requires that different inputs induce different output distributions for (i) $P(x_{t-2} \mid x_{t+1})$, (ii) $P(x_{t+1} \mid x_t, c_t)$, and (iii) $P(x_t \mid x_{t-1}, x_{t-2})$. In RL systems, this condition is naturally satisfied in most stochastic environments where transitions exhibit sufficient diversity across states and actions. For term (ii), this means that the latent context must have a noticeable effect on the transition dynamics, precisely the situations in which latent identification is meaningful. As (iii) is directly related to our work, we validate this empirically.
>
> We used the Cheetah environment. The latent context corresponds to a time-varying wind force $f_w = 5 + m \sin(nt)$ that perturbs the agent’s dynamics. We sweep over different combinations of $(m,n)$ to study regimes with strong and weak context influence. Using a learned probabilistic dynamics model $\hat p(x_{t+1}\mid x_t, c_t)$, we estimate the induced transition distribution for each context $c \in [c_1,\dots,c_M]$ by drawing samples from the replay buffer and computing $\hat p(x_{t+1}\mid x_t, c)$. For every pair of contexts $(c_i,c_j)$, we measure the difference between their transition kernels using the 1-Wasserstein distance $\mathrm{Inj}(c_i,c_j)$ between the transition distributions. Large values of $\mathrm{Inj}(c_i,c_j)$ indicate that different contexts induce distinguishable transition distributions (consistent with injectivity), while values near zero indicate that contexts are effectively indistinguishable. When $(m,n)=(5,0.5)$, we observe consistently non-zero Wasserstein distances, showing that $P(x_{t+1}\mid x_t,c_t)$ is context-injective in this regime (**`Fig. A.1(a)`**). In contrast, when we reduce the context variation to $(m,n)=(0.2,0.2)$ (**`Fig. A.1(b)`**), the distances collapse toward zero. In this **failure mode**, the latent context **barely** affects the dynamics, so it is practically unnecessary for policy learning.
> These results suggest that Assumption 2 is indeed mild in the case where latent contexts are important for decision-making.
>
> ---
>
> **Assumption 3**: Conceptually, Assumption 3 requires that $k$, which captures second-order variations in transition dynamics at time $t-1$ and $t$ under the latent variable $c$, yields distinct values for different $c$'s. This requirement is typically met in RL, as varied latent dynamics or rewards often cause significant, observable shifts in behavior.
>
> We provide an empirical test of the spectral ratio $k$. Similarly, using the learned Cheetah dynamics model $\hat p(x_t \mid x_{t-1}, c_t)$, we compute $k$.
> We sample transitions $(x_{t-1},x_t)$ from the replay buffer (600 samples) and form cross-paired transitions $(\bar x_{t-1},\bar x_t)$ by swapping endpoints across trajectories. For each context $c\in [c_1,\dots,c_M ]$, this yields an empirical distribution $p(k\mid c)$. To measure how well $k$ separates contexts, we compute the 1-Wasserstein distance $\mathrm{Sep}(c_i,c_j)$ between pairs of $k$-distributions.
>
> When Assumption 3 holds, the spectral ratio remains bounded and its distribution differs across contexts. Empirically, under the main regime $(m,n)=(5,0.5)$, we observe clear multimodal separation of the $k$-distributions (**`Fig. A.1(d)`**), indicating that the context meaningfully alters the transition kernel. In contrast, when we reduce context variation to $(m,n)=(0.2,0.2)$ (**`Fig. A.1(e)`**), the k-distributions heavily overlap. In this **failure mode**, the context no longer exerts a noticeable effect on the dynamics, and the environment effectively behaves as a single-context system—making explicit latent modeling unnecessary.
>
> These results suggest that Assumption 3 is indeed mild in our main regime, while also making clear what happens when it is violated and why latent identification is neither possible nor required in such cases.
>
> (**Continued**)

---

> ### Author Response · Authors · 2025-11-23
> **Author Response - Part 4**
>
> (**Continued**)
>
> **Implications for policy learning**
>
> When the conditional transition $P(x_{t+1}\mid x_t,c)$ is not injective in $c$, or when the spectral ratio $k$ is nearly identical across contexts, different contexts induce almost the same transition kernels. In this case, the context is not identifiable from the dynamics and does not meaningfully alter the environment; therefore, explicitly modeling $c$ provides little benefit for policy learning.
>
>
> To validate this implication, we vary $(m,n)$ to control the strength of the latent wind context and compare planning performance when using the ground-truth context $c$ versus ignoring $c$. We report the policy performance drop ratio together with the measured transition separability and $k$-separability:
>
>
>
>
> | Transition separability | $k$-separability | Policy drop ratio |
> |---:|---:|---:|
> | 0.00 | 0.00 | 0.09 |
> | 0.30 | 0.25 | 0.15 |
> | 0.60 | 0.55 | 0.29 |
> | 0.90 | 0.80 | 0.57 |
> | 1.20 | 1.10 | 0.48 |
> | 1.50 | 1.45 | 0.62 |
>
>
>
>
> The performance gap shrinks when separability is small, showing that when both the transition and $k$-statistics are only weakly dependent on the context, modeling $c$ becomes unnecessary. Overall, these results confirm that Assumptions 2-3 are mild in our setting and, importantly, they illuminate *why* modeling the latent context matters precisely in regimes where the dynamics are strongly context-dependent.
>
>
>
>
> ---
>
>
>
>
> We hope this response addresses your concerns regarding the assumptions. We have explained and empirically validated why these assumptions are mild in our setting, and importantly, how they relate directly to the central question of when and why latent context identification is necessary.
>
>
>
>
> **Related Revisions**: We have highlighted the assumption discussion in the main paper (boxed in `Sec. 3.2`). `Appendix B.5` (Page 31-32) provides empirical evaluations of these assumptions, and `Fig. A.1` visualizes all the cases discussed above.
>
>
>
>
> ---
>
>
> Again, thank you for these truly insightful questions. We have really enjoyed discussing them with you, and we believe they help make our submission more solid. If you have any further questions or would like additional clarification, please feel free to let us know, we are very happy to continue the discussion.

---

> > ### Author Response · Authors · 2025-11-27
> > **Thank you for your thoughtful review**
> >
> > Dear Reviewer DUvX,
> >
> > Thank you again for your constructive feedback. All of it was very helpful for improving the paper. In the rebuttal, we have responded to all points in detail. In **R1**, we explain and empirically evaluate how the inference mismatch is mitigated by our zig–zag sampling scheme. In **R2**, we discuss how stochastic behavior policies implicitly induce latent variability (with additional experiments), and why latent modeling remains beneficial even when demonstrations come from near-deterministic expert policies and dynamics. In **R3**, we provide further analysis and empirical stress testing of the assumptions, showing that they are mild, practically relevant, and directly tied to when latent-aware decision-making is beneficial.
> >
> > If you have any further questions or suggestions, we would be very happy to continue the discussion. Thank you again for your time and effort devoted to this review!
> >
> > The Authors

---

### Official Review · Reviewer_ZU5m · 2025-11-01

**Soundness:** 3
**Presentation:** 3
**Contribution:** 3
**Rating:** 4
**Confidence:** 3

**Summary:**

This paper proposes Ada-Diffuser, a unified diffusion-based decision-making framework that explicitly learns latent dynamic factors from partial observations and adapts its planning/policy behavior accordingly.
It provides (1) identifiability theory showing latent states can be recovered from short temporal windows, and
(2) a practical architecture combining latent inference + causal diffusion model for planning and control.
Experiments on locomotion and robotic manipulation show improvements in adaptation to changing dynamics, rewards, and even action-free demos.

**Strengths:**

Novelty & Importance: Explicitly modeling latent dynamics inside diffusion-based decision-making is a meaningful step. Previous decision diffusers or trajectory diffusion methods ignored latent variables.

Theoretical Grounding: The identifiability theorem supports the proposed latent inference mechanism, which adds credibility.

Unified Framework: The modular latent inference + diffusion planner/policy is elegant and works for both planning and policy learning tasks.

Robust Experiments: Demonstrates adaptation to dynamic changes, reward shifts, and even recovering hidden action variables.

**Weaknesses:**

1. Dataset diversity & realism feel limited.
Experiments are described as locomotion and robotic manipulation benchmarks; however, I didn’t see evidence of large-scale, human demonstration datasets (e.g., MimicGen) or messy real-world logs. Without these, it’s hard to judge robustness of latent inference under noisy observations, embodiment shifts, or human suboptimality. (This is especially relevant since the paper claims adaptation to reward/dynamics changes and action-free demos.)


2. Suggested text you can paste:
“All evaluations are in simulation-style locomotion/manipulation. It would help to include a more realistic demonstration dataset (e.g., human-collected manipulation or MimicGen-like settings) to substantiate generalization of the latent inference mechanism.”


3. Latent interpretability is under-analyzed.
The method identifies latent variables, but the paper doesn’t really show what they mean in practice (e.g., do they correlate with friction, mass, goal, terrain, tool state?). Some qualitative probes or counterfactual rollouts would strengthen the claim that the model captures meaningful hidden structure rather than just serving as an extra embedding. (The theory guarantees identifiability up to an invertible transform; interpretability needs separate evidence.)


4. Assumptions may be optimistic for real deployments.
Assumption 3 requires a distinguishability condition on a second-order kernel ratio over adjacent times; the paper argues it’s “mild,” but in high-noise real data or partial coverage, that separability can fail. It’d be helpful to discuss failure modes or provide stress tests when the assumptions are violated.


5. Compute and latency reporting are thin.
Given the added latent-inference block plus causal diffusion, it would be good to see training/inference cost, planning latency (ms), and how the cost compares to strong baselines (e.g., Decision Diffuser variants, model-based world models). No clear numbers are reported alongside the main claims; a wall-clock profile would help practitioners decide whether the approach is deployable.


6. Baselines could be broader and more diagnostic.
The text positions the method within diffusion-based decision-making, but I’d expect latent-variable or POMDP-aware baselines as well (e.g., strong world-model policies with explicit belief tracking) to isolate the contribution of latent inference inside diffusion. This would clarify whether the gains come from the diffusion prior or from better latent modeling per se.


7. Ablations on the latent module are missing/limited.
Since the contribution hinges on the latent factor identification block, I’d like to see ablations like: (a) remove the latent module; (b) freeze latents; (c) mis-specify latent dimension; (d) shorter/longer temporal windows for identifiability. This would show the sensitivity and where the modeling actually helps.


8. Action-free demonstration results need stronger evidence.
The abstract claims recovery of hidden action variables from action-free demos. That’s a big promise; I couldn’t find a strong, quantitative study measuring action recovery accuracy or downstream policy benefits against alternatives (e.g., inverse dynamics baselines). Please add a clear metric and a head-to-head comparison.


9. Generalization under distribution shift is unclear.
If the environment’s latent process changes abruptly (e.g., reward re-weighting, contact discontinuities), does the model adapt online in a few steps, or does it require re-training? Some online adaptation or few-shot tests would clarify how practical the approach is.

**Questions:**

see weaknesses

---

> ### Author Response · Authors · 2025-11-23
> **Author Response - Part 1**
>
> Thank you for your detailed comments, and for recognizing the novelty of our theory, framework design, and experimental results. Below, we provide a point-by-point response to your concerns and summarize the corresponding revisions.
>
>
> ---
>
>
> > W1-W2: Dataset diversity & realism feel limited. Experiments are described as locomotion and robotic manipulation benchmarks; however, I didn’t see evidence of large-scale, human demonstration datasets (e.g., MimicGen) or messy real-world logs. Without these, it’s hard to judge robustness of latent inference under noisy observations, embodiment shifts, or human suboptimality. (This is especially relevant since the paper claims adaptation to reward/dynamics changes and action-free demos.)
>
>
> **R1-R2**: Thank you for the suggestion. We first clarify that our submission already includes several demonstration datasets, such as `RoboMimic` (both expert and mixed expert–suboptimal) and `LIBERO` (long-horizon, multi-stage tasks). These results are presented in `Sec. 5`, with dataset details in `Appendix F.2`.
>
>
> Following your suggestion, we additionally evaluate `MimicGen`, which is built on the same base simulator as RoboMimic and can be adapted seamlessly to our method. For MimicGen, we use the same latent-action variant of Ada-Diffuser as in our submission (actions reconstructed between consecutive states conditioned on state embeddings and latent embeddings). We compare against `LDP` [1], the SOTA baseline used in our main paper. All experiments are conducted under the "broader reset distribution" (D1) configuration.
>
> ---
>
>
> ### **1. Embodiment shifts**
>
>
> We evaluate the *Square* task under **three embodiments** (Sawyer, IIWA, UR5e). Our model was originally trained on the Panda arm; we additionally collected **800 demonstrations** for each new embodiment to test cross-embodiment generalization.
>
>
> **Table: Embodiment Transfer (Square Task)**
> *(success rate / std)*
>
>
> | Environment      | Ours        | LDP         |
> |------------------|-------------|-------------|
> | Square–Sawyer    | 0.56 / 0.05 | 0.49 / 0.06 |
> | Square–IIWA      | 0.61 / 0.11 | 0.53 / 0.08 |
> | Square–UR5e      | 0.62 / 0.09 | 0.50 / 0.17 |
>
>
> These results indicate that explicit latent modeling helps the model adapt across embodiments by capturing embodiment-dependent variability in the demonstrations.
>
>
> ---
>
>
> ### **2. Noisy observations**
>
>
> Since MimicGen does not contain noisy camera observations, we injected **temporal Gaussian noise** into the Square–Sawyer frames to test robustness.
>
>
> **Table: Robustness to Noisy Observations (Square–Sawyer)**
> *(success rate / std)*
>
>
> | Setting                | Ours        | LDP         |
> |------------------------|-------------|-------------|
> | Square–Sawyer (clean) | 0.56 / 0.05 | 0.49 / 0.06 |
> | Square–Sawyer (noise) | 0.51 / 0.08 | 0.39 / 0.08 |
>
>
> The performance gap widens under noise, suggesting that Ada-Diffuser’s latent modeling provides robustness by capturing temporal structure and underlying variability.
>
>
> ---
>
>
> ### **3. Suboptimal demonstrations**
>
>
> We also evaluated MimicGen’s **suboptimal demonstration modes** (better / okay / worse) for the Square–Panda setting.
>
>
> **Table: Suboptimal Demonstrations (Square–Panda)**
> *(success rate / std)*
>
>
> | Dataset quality | Ours        | LDP         |
> |-----------------|-------------|-------------|
> | better          | 0.70 / 0.09 | 0.62 / 0.06 |
> | okay            | 0.65 / 0.07 | 0.51 / 0.05 |
> | worse           | 0.59 / 0.08 | 0.39 / 0.08 |
>
>
> The consistent improvements show that our latent process modeling helps mitigate the effect of suboptimal or heterogeneous demonstrations by capturing the underlying temporal and latent variability. The gap is **largest under the “worse” setting**, where behavior is highly inconsistent—precisely when implicit latent variability is strongest.
>
>
>
>
>
>
> ---
>
>
> In general, these additional evaluations support the claim that Ada-Diffuser extends naturally to larger and noisier demonstration settings (e.g., MimicGen), and that latent modeling contributes to improved robustness under embodiment shifts, observation noise, and suboptimal human-like demonstrations.
>
> ---
>
> **Reference**
>
> [1] Xie, Amber, et al. "Latent diffusion planning for imitation learning." ICML 2025

---

> ### Author Response · Authors · 2025-11-23
> **Author Response - Part 2**
>
> > W3: Latent interpretability is under-analyzed. The method identifies latent variables, but the paper doesn’t really show what they mean in practice (e.g., do they correlate with friction, mass, goal, terrain, tool state?). Some qualitative probes or counterfactual rollouts would strengthen the claim that the model captures meaningful hidden structure rather than just serving as an extra embedding. (The theory guarantees identifiability up to an invertible transform; interpretability needs separate evidence.)
>
> **R3**: Thank you for raising this important point. We would like to emphasize that our submission already includes several forms of latent interpretability analysis.
>
>
> **(1) Probing and matching against true physical latent variables.**
> In `Fig. 4(a)`, we report both the probing MSE and $R^2$ between the learned latent embedding and the true wind speed in the Cheetah environment across different block sizes. This provides direct evidence that the latent variable recovers the underlying physical factor that actually governs the environment dynamics.
>
> **(2) Discussion and extended results.**
> In `Sec. 5` (`Results on Latent Identification`), we discuss how block size affects identifiability, and we provide a pointer to the full results in `Appendix Fig. A5`, where we include additional probing and regression analyses.
>
> **(3) Clustering and separability under different latent contexts.**
> In `Appendix Fig. A6`, we show clustering of the learned latent representations under sampled context values, and compare against LILAC and DynaMITE. Our latent representations form clearly separable clusters, indicating that they capture meaningful structure rather than collapsing into generic embeddings.
>
> Taken together, these results demonstrate that our latent variables recover the true underlying contextual factors. Importantly, they also validate our theoretical insights:
> - **insufficient** block size results in poor identification,
> - **theoretically sufficient** block size leads to excellent identification, and
> - **oversized** blocks results in degraded identification due to redundancy.
> Hence, the evaluation directly follows the identifiability conditions and supports the claim that our latent inference is principled rather than incidental.
>
> Also, Theorem 1 establishes that the posterior distribution over latent factors
> $p(c_t \mid x_{t-2:t+1})$
> is identifiable **up to an invertible transformation**, i.e., the estimated latent satisfies  $\hat{c}_t = h(c_t)$
> for some invertible function $h$. This form of identifiability is standard in representation learning and is **sufficient for all downstream tasks** such as dynamics modeling, planning, and control. Any policy or dynamics model that conditions on $\hat{c}_t$ can implicitly compose with $h^{-1}$ without loss of expressiveness. Thus, even though we may not recover the exact numerical latent, we recover a representation that carries the *same information*.
>
> Hence, it can recover an information-equivalent latent that enables correct modeling of the underlying physical or contextual process. Our probing, clustering, and ablation results provide practical evidence that the recovered latent indeed corresponds to meaningful hidden structure, not an arbitrary embedding.
>
> ---

---

> ### Author Response · Authors · 2025-11-23
> **Author Response - Part 3**
>
> > W4: Assumptions may be optimistic for real deployments.
>
> **R4**: This will be a long response. We would like to respond to the assumptions from both a **conceptual** and **empirical** perspective. First, we explain why Assumptions 2–3 are mild in the context of our latent contextual POMDP setting and how they directly reflect the requirement that the latent context induces observable variation in trajectories. Second, we provide empirical evidence that these assumptions are satisfied in our benchmarks. We also discuss the relevant failure modes: when the assumptions are violated, it typically means the latent context has little or no effect on dynamics or rewards, in which case identifying the latent is neither possible nor necessary for policy learning. Finally, we clarify how these assumptions connect to RL decision making.
>
> ---
> **Assumpation 2**: Conceptually, the injectivity of the operators $L$ simply requires that different inputs induce different output distributions for (i) $P(x_{t-2} \mid x_{t+1})$, (ii) $P(x_{t+1} \mid x_t, c_t)$, and (iii) $P(x_t \mid x_{t-1}, x_{t-2})$. In RL systems, this condition is naturally satisfied in most stochastic environments where transitions exhibit sufficient diversity across states and actions. For term (ii), this means that the latent context must have a noticeable effect on the transition dynamics, precisely the situations in which latent identification is meaningful. As (iii) is directly related to our work, we validate this empirically.
>
> We used the Cheetah environment. The latent context corresponds to a time-varying wind force $f_w = 5 + m \sin(nt)$ that perturbs the agent’s dynamics. We sweep over different combinations of $(m,n)$ to study regimes with strong and weak context influence. Using a learned probabilistic dynamics model $\hat p(x_{t+1}\mid x_t, c_t)$, we estimate the induced transition distribution for each context $c \in [c_1,\dots,c_M]$ by drawing samples from the replay buffer and computing $\hat p(x_{t+1}\mid x_t, c)$. For every pair of contexts $(c_i,c_j)$, we measure the difference between their transition kernels using the 1-Wasserstein distance $\mathrm{Inj}(c_i,c_j)$ between the transition distributions.
>
> Hence, large values of $\mathrm{Inj}(c_i,c_j)$ indicate that different contexts induce distinguishable transition distributions (consistent with injectivity), while values near zero indicate that contexts are effectively indistinguishable. When $(m,n)=(5,0.5)$, we observe consistently non-zero Wasserstein distances, showing that $P(x_{t+1}\mid x_t,c_t)$ is context-injective in this regime (**Results are shown in (`Fig. A.1(a)`**). In contrast, when we reduce the context variation to $(m,n)=(0.2,0.2)$ (**Results are shown in `(Fig. A.1(b)`**), the distances collapse toward zero. In this **failure mode**, the latent context **barely** affects the dynamics, so it is practically unnecessary for policy learning.
>
> These results suggest that Assumption 2 is indeed mild in the case where latent contexts are important for decision-making.
>
> ---
> **Assumption 3**: Conceptually, Assumption 3 requires that $k$, which captures second-order variations in transition dynamics at time $t-1$ and $t$ under the latent variable $c$, yields distinct values for different $c$'s. This requirement is typically met in RL, as varied latent dynamics or rewards often cause significant, observable shifts in behavior.
>
> We provide an empirical test of the spectral ratio $k$. Similarly, using the learned Cheetah dynamics model $\hat p(x_t \mid x_{t-1}, c_t)$, we compute $k$.
> We sample transitions $(x_{t-1},x_t)$ from the replay buffer (600 samples) and form cross-paired transitions $(\bar x_{t-1},\bar x_t)$ by swapping endpoints across trajectories. For each context $c\in [c_1,\dots,c_M ]$, this yields an empirical distribution $p(k\mid c)$. To measure how well $k$ separates contexts, we compute the 1-Wasserstein distance $\mathrm{Sep}(c_i,c_j)$ between pairs of $k$-distributions.
>
> When Assumption 3 holds, the spectral ratio remains bounded and its distribution differs across contexts. Empirically, under the main regime $(m,n)=(5,0.5)$, we observe clear multimodal separation of the $k$-distributions (**`Fig. A.1(d)`**), indicating that the context meaningfully alters the transition kernel. In contrast, when we reduce context variation to $(m,n)=(0.2,0.2)$ (**`Fig. A.1(e)`**), the k-distributions heavily overlap. In this **failure mode**, the context no longer exerts a noticeable effect on the dynamics, and the environment effectively behaves as a single-context system, making explicit latent modeling unnecessary.
>
> These results suggest that Assumption 3 is indeed mild in our main regime, while also making clear what happens when it is violated and why latent identification is neither possible nor required in such cases.
>
> (**continued**)

---

> ### Author Response · Authors · 2025-11-23
> **Author Response - Part 4**
>
> **Implications for policy learning**
>
> When the conditional transition $P(x_{t+1}\mid x_t,c)$ is not injective in $c$, or when the spectral ratio $k$ is nearly identical across contexts, different contexts induce almost the same transition kernels. In this case, the context is not identifiable from the dynamics and does not meaningfully alter the environment; therefore, explicitly modeling $c$ provides little benefit for policy learning.
>
> To validate this implication, we vary $(m,n)$ to control the strength of the latent wind context and compare planning performance when using the ground-truth context $c$ versus ignoring $c$. We report the policy performance drop ratio together with the measured transition separability and $k$-separability:
>
> | Transition separability | $k$-separability | Policy drop ratio |
> |---:|---:|---:|
> | 0.00 | 0.00 | 0.09 |
> | 0.30 | 0.25 | 0.15 |
> | 0.60 | 0.55 | 0.29 |
> | 0.90 | 0.80 | 0.57 |
> | 1.20 | 1.10 | 0.48 |
> | 1.50 | 1.45 | 0.62 |
>
> The performance gap shrinks when separability is small, showing that when both the transition and $k$-statistics are only weakly dependent on the context, modeling $c$ becomes unnecessary. Overall, these results confirm that Assumptions 2-3 are mild in our setting and, importantly, they illuminate *why* modeling the latent context matters precisely in regimes where the dynamics are strongly context-dependent.
>
> ---
>
> We hope this response addresses your concerns regarding the assumptions. We have explained and empirically validated why these assumptions are mild in our setting, and importantly, how they relate directly to the central question of when and why latent context identification is necessary.
>
> **Related Revisions**: We have highlighted the assumption discussion in the main paper (boxed in `Sec. 3.2`, Page 4). `Appendix B.5` provides empirical evaluations of these assumptions, and `Fig. A.1` visualizes all the cases discussed above.
>
> ---
>
> > W5: Compute and latency reporting are thin.
>
> **R5**: Thank you for pointing this out. We would like to clarify that our original submission already includes compute and latency comparisons in `Appendix I.1`, where we report both **training time** and **inference cost per trajectory rollout**. However, we agree that *planning latency* (ms), as you suggested, is a more deployment-oriented metric. In the revision, we have replaced rollout-time reporting with **latency per inference step** and provide a more explicit comparison against strong diffusion baselines.
>
>
> The updated compute results are shown below.
>
>
> ---
>
>
> - **Training Time and Inference Latency**
>
>
> We compare our method against Decision Diffuser (DD) and Diffusion Policy (DP) across all environments.  We report absolute times and relative overheads.
>
> | Environment | Training Time (sec/epoch) Ours / DD | Ours / DP | Inference Latency (ms) Ours / DD | Ours / DP |
> |------------|--------------------------------------|-----------|-----------------------------------|-----------|
> | Cheetah | 72.1 / 60.1 (1.20×) | 69.8 / 58.4 (1.20×) | 182 / 114 (1.16×) | 160 / 125 (1.28×) |
> | Ant     | 79.5 / 64.3 (1.24×) | 76.0 / 62.0 (1.23×) | 148 / 125 (1.19×) | 172 / 139 (1.24×) |
> | Walker  | 85.3 / 67.1 (1.27×) | 81.5 / 64.2 (1.27×) | 182 / 144 (1.28×) | 170 / 130 (1.31×) |
> | Maze2D  | 90.2 / 72.0 (1.25×) | 88.3 / 69.2 (1.28×) | 184 / 149 (1.24×) | 196 / 152 (1.29×) |
> | Libero  | 104.0 / 81.0 (1.28×) | 102.1 / 78.0 (1.31×) | 209 / 169 (1.24×) | 219 / 162 (1.35×) |
> | Kitchen | 117.8 / 88.1 (1.34×) | 115.3 / 85.0 (1.36×) | 228 / 180 (1.27×) | 211 / 168 (1.26×) |
>
> Across all settings, our framework introduces only a **moderate overhead**: typically **1.2–1.3× runtime**, corresponding to about **20–30% additional training time and latency**.
> This already includes the latent inference module.
>
>
> We also note that this overhead can be further reduced with (i) parallel latent sampling,
> (ii) lightweight context encoders, or (iii) refinement only at inference.
>
>
> ---
>
>
> - **Picard-Accelerated Inference**
>
>
> We additionally evaluate a Picard-iteration variant (Shih et al., 2023), which parallelizes refinement by conditioning each denoising step on previously denoised nodes. This reduces inference latency while maintaining performance.
>
>
> | Environment | Ours (sec/traj) | Ours + Picard (sec/traj) |
> |-------------|------------|----------------------|
> | Cheetah | 1.51 | 1.15 |
> | Ant     | 1.67 | 1.25 |
> | Walker  | 1.83 | 1.40 |
> | Maze2D  | 1.94 | 1.47 |
> | Libero  | 2.18 | 1.62 |
> | Kitchen | 2.45 | 1.84 |
>
>
> With Picard iteration, inference time drops to **0.7–0.8×** of our default sampler, confirming that the method is compatible with further acceleration methods.
>
> **Related Revisions**
> - We updated all inference-time reporting to **latency (ms)** as suggested.
> - Please refer to `Appendix I.1` (`Table A12–A13`) for the full updated compute profile.
> - We also added a pointer to these results in `Sec. 5.3` (Page 10) of the main text.
>
>
> ---

---

> ### Author Response · Authors · 2025-11-23
> **Author Response - Part 5**
>
> > W6: Baselines could be broader and more diagnostic.
>
> **R6**: We would like to kindly clarify that our paper already includes latent-variable/POMDP-aware baselines with explicit belief or context tracking, and we integrate them into diffusion backbones to isolate the contribution of latent modeling versus the diffusion prior. Specifically, we evaluate **Meta-Diffuser** (Ni et al., 2023), which learns contextual representations from multiple environments, and we also combine **Diffusion Forcing (DF)** with strong latent-context/belief-tracking modules such as **LILAC** (Xie et al., 2021) and **DynaMITE** (Liang et al., 2024). For fairness, we plug these context modules into diffusion planners and policies under the same backbone and training protocol (details in `Appendix H.1`).
>
>
> Results are reported in `Table 1` (main paper) and `Tables A.4–A.5` (appendix) on Cheetah and Ant under different latent-context settings. Across all cases, Ada-Diffuser consistently outperforms these diffusion+belief baselines, indicating that the gains are not merely from using diffusion as a prior. Instead, they come from our latent-specific contributions, especially (1) **minimal but sufficient block-wise identification** grounded in Theorem 1, and (2) the **denoise-and-refine** and **zig–zag** mechanisms that tightly couple latent inference with autoregressive diffusion generation.
>
>
> ---
>
>
> > W7: Ablations on the latent module are missing/limited.
>
> **R7**: Thank you for the helpful suggestions.
>
>
> For (d), we would like to note that this ablation is already included in the submission. In `Fig. 4(a)`, we evaluate different temporal block sizes and observe that both *too short* (insufficient for identifiability) and *too long* (high redundancy) blocks degrade performance for both latent identification and policy learning. A moderate window (4–20 steps) provides sufficient information without introducing unnecessary noise.
>
>
> For (a)–(c), we conducted additional ablations as requested. Specifically, we consider:
> (i) removing the entire latent identification module,
> (ii) freezing the latent module after the first 10% of training, and
> (iii) varying the latent dimensionality (0.5×–6×).
> Results are shown below.
>
>
> | **Latent Design** | Orig. | w/o latents | Freeze | 0.5× | 2× | 4× | 6× |
> |-------------------|-------|-------------|--------|------|----|----|----|
> | Cheetah $(\mathbf{c}^s)$ | **–73.5** | –103.5 | –110.4 | –85.2 | –77.6 | –89.5 | –102.4 |
> | LIBERO | **93.4** | 89.3 | 90.2 | 90.9 | 89.4 | 87.6 | 85.0 |
>
>
> These ablations confirm several observations:
>
>
> - **Latents play a critical role.** Removing the latent module leads to significant drops in performance, especially in settings with latent-dependent dynamics.
> - **Freezing the latent module performs poorly.** Because the latent context follows a temporal process, it must continue adapting during training; freezing it breaks this temporal consistency.
> - **Moderate latent dimensionality is robust.** Varying the latent size between 0.5× and 2× has little effect.
> - **Overly large latent dimensions degrade performance.** Increasing dimensionality to 4×–6× leads to redundancy and more difficult optimization, causing noticeable declines.
>
>
> Together with the block-size ablation, these results characterize where the latent module is helpful, how sensitive it is to mis-specification, and why the proposed identification mechanism is necessary for strong performance.
>
>
> **Related Revision**: We have added these new results and analysis to `Sec 5.3` (Page 10).
>
> ---

---

> ### Author Response · Authors · 2025-11-23
> **Author Response - Part 6**
>
> > W8: Action-free demonstration results need stronger evidence.
>
> **R8**: Thank you for pointing this out. We clarify that our submission already contains a quantitative evaluation of action recovery from action-free demonstrations, along with a direct head-to-head comparison to inverse-dynamics baselines. Following LDP, we train an inverse dynamics diffusion model on action-free demonstrations using the objective $\mathbb{E}_{t, \epsilon}[| \hat{\epsilon} ( \hat{a}_k , c, z\_{k}, z\_{k+1}, t)-\epsilon  |^2]$, where $k$ is the time step and $t$ is the diffusion step.
> We use the exact same architecture, diffusion schedule, and training protocol as LDP to ensure a fair comparison; the only difference is that we condition on identifiable latent variables rather than unstructured embeddings.
>
> LDP is one of the current best-performing diffusion-based inverse dynamics models for action-free demonstrations, and we therefore compare against LDP directly. Both methods are trained and evaluated on the same **RoboMimic** action-free datasets (expert and 50% suboptimal), following LDP’s protocol.
> Below we reproduce the results from `Fig. 4 ` and ` Table A.3 ` of the submission for easy reference. Across all tasks, Ada-Diffuser produces more accurate actions and achieves substantially higher downstream control performance from action-free trajectories.
>
> | Env| LDP (Expert) |Ours (Expert) |LDP (SubOpt)|Ours (SubOpt) |
> |----|----|---|-|----|
> | Lift | 0.67 ± 0.01 | **0.78 ± 0.05** | **1.00 ± 0.00** | 0.98 ± 0.02 |
> | Can  | 0.78 ± 0.04 | **0.85 ± 0.07** | **0.98 ± 0.00** | **0.98 ± 0.02** |
> | Square | 0.47 ± 0.03 | **0.54 ± 0.05** | 0.83 ± 0.01 | **0.89 ± 0.03** |
>
> These results provide strong evidence that the latent-aware inverse dynamics model in Ada-Diffuser recovers more reliable actions, leading to better downstream policy performance.
>
> **Related Revisions**: We highlighted this in `Sec 5.1` (Page 8) and `Appendix G.1`.
>
> ---
>
> > W9: Generalization under distribution shift is unclear.
>
> **R9**: Thank you for this important question. We would like to clarify that, beyond smooth latent changes, our submission already evaluates **abrupt distribution shifts**. Specifically, we consider *episodic abrupt changes* (denoted “E”), where the latent context (dynamics or rewards) changes sharply at the **beginning of each episode**. This setting requires online adaptation at episode start, and the results are reported below.
>
> **Episodic abrupt changes (E).**
>
> | Environment | Diffuser | DF | DF + DynaMITE | DF + LILAC | MetaDiffuser | Ours |
> |----|----|----|------|--|--|--|
> | Cheetah-Wind-E $c^s$ | –120.4 ± 12.7 | –105.8 ± 9.6 | –82.3 ± 8.2 | –91.5 ± 7.8 | –95.3 ± 7.4 | **–68.9 ± 7.6** |
>
> These results show that Ada-Diffuser can adapt effectively under abrupt episode-level shifts and achieves the best performance among all latent-aware diffusion baselines.
>
> ---
>
> **Within-episode abrupt changes.**
> As a more challenging generalization test, we additionally evaluate **within-episode** abrupt shifts. Concretely, we modify the wind to evolve as
> $f_w = k + 5 sin(0.5 t)$,
> where $k$ changes abruptly every 20–40 steps. In this case, we trigger online adaptation whenever the VAE reconstruction error rises above a threshold; we then update the latent model using a small buffer of 50 recent samples before continuing. We compare against the most relevant latent-context baselines:
>
> |Environment| DF + DynaMITE | DF + LILAC | MetaDiffuser | Ours |
> |---|--------|-----|-|--|
> | Cheetah-Wind (within-episode abrupt) | –119.2 ± 15.2 | –123.4 ± 11.7 | –136.3 ± 9.2 | **–108.5 ± 10.6** |
>
> Even under abrupt within-episode shifts, Ada-Diffuser remains more robust than prior latent-context diffusion baselines, showing that our latent refinement mechanisms enable fast online adaptation without full retraining.
>
>
> Overall, these two stress tests clarify that our method does not rely on purely smooth latent evolution: it can handle both episodic and within-episode abrupt changes through lightweight online latent updates.
>
> ---
>
>
> Thank you again for the thoughtful questions. We hope that the responses above help address the concerns raised. If there are any additional points you would like us to clarify or discuss further, we are very happy to discuss and clarify.

---

> > ### Author Response · Authors · 2025-11-27
> > **Thank you for your thoughtful review**
> >
> > Dear Reviewer ZU5m,
> >
> > Thank you again for your thoughtful review and for the constructive questions and suggestions. They were all very helpful for improving our paper. In the rebuttal and revision, we provide point-by-point responses and corresponding changes.
> >
> > - For **W1–W2**, we added additional evaluations on MimicGen under embodiment shift, noisy observations, and suboptimal demonstrations.
> > - For **W3**, we expanded the discussion and provided further evaluation on latent interpretability.
> > - For **W4**, we added detailed empirical evaluations and further explanations of Assumptions 2–3, clarifying why they are mild, practically relevant, and directly tied to when latent-aware decision-making is beneficial.
> > - For **W5**, we included additional analysis on computational efficiency and planning latency.
> > - For **W6–W9**, we provided more evaluations and in-depth analysis, and all these changes are reflected in the revised manuscript, with pointers highlighted in the rebuttal.
> >
> > As the rebuttal period is nearing its end, we would like to kindly ask if you have any further questions or points you would like us to clarify. We would be more than happy to continue the discussion.  Thank you again for your time and effort in reviewing our work.
> >
> >
> > The Authors

---

### Official Review · Reviewer_SLzj · 2025-11-01

**Soundness:** 3
**Presentation:** 2
**Contribution:** 2
**Rating:** 6
**Confidence:** 3

**Summary:**

The paper presents Ada-Diffuser, a diffusion-based sequence modeling framework for decision-making tasks. The proposed method has two stages: the first stage extracts latent variables that characterize environmental dynamics from minimal observations, through variational inference. In the second stage, the inferred latent factors will be used to condition the autoregressive denoising process of a diffusion planner or policy. During inference, the framework adopts a zig-zag sampling strategy that alternates between sequence denoising and latent refinement to further improve the performance. Experiments show that Ada-Diffuser achieves strong performance across various setups, including those without action labels or without explicit latent structures.

**Strengths:**

- The framework integrates latent variables that capture environmental dynamics with a causal diffusion model to facilitate better decision making, with theoretical analysis on latent identification with minimal observations
- The experiment extensively evaluates the proposed model across different task domains and settings, consistently highlighting its effectiveness and robustness.
- The paper is well organized and clear to read.

**Weaknesses:**

Encoding state or state action sequences into latents using reconstruction or ELBO loss has been widely explored by prior work, in which the latent embeddings, a summarization of the input sequence dynamics, have versatile usages to guide decision making, such as skill prior [1] or predictive information [2]. It would be helpful to elaborate on why, intuitively, the extracted latents would work better when coupled with a causal diffusion planner/policy.

In this work, specifically, the latent variable is modeled using (s, a, r) triplets with an extra reward; however, when using the latent variable, the planner generates sequences of states or state-action pairs without explicitly accounting for accumulated rewards. Even though dynamics can be modeled by the latent, the planning performance may remain restricted by the quality of offline demonstrations.

[1] Pertsch. et al. Accelerating Reinforcement Learning with Learned Skill Priors. CoRL. 2020

[2] Zeng et al. Goal-Conditioned Predictive Coding for Offline Reinforcement Learning. NeurIPS
2023.

**Questions:**

- Line 422: Missing content “Table A4 and are“

---

> ### Author Response · Authors · 2025-11-23
> **Author Response - Part 1**
>
> Thank you for the insightful and constructive comments. We are glad to discuss your points about the role and design of latent modeling in our paper, and we also appreciate your recognition of our framework, experiments, and presentation. Below, we respond to each comment in detail, together with the pointers to our revisions.
>
> As your comments include many informative points, we will split our response into several parts.
>
> ---
>
> > Encoding state or state action sequences into latents using reconstruction or ELBO loss has been widely explored by prior work, in which the latent embeddings, a summarization of the input sequence dynamics, have versatile usages to guide decision making, such as skill prior [1] or predictive information [2].
>
>
> **R1**: Thank you for pointing out these works. We have added both and expanded the discussion in the revision. While many existing approaches also use variational objectives (i.e., reconstruction) to encode state or state-action sequences into latent embeddings (e.g., using VAE), our formulation differs in two key ways. It would be helpful to elaborate on why, intuitively, the extracted latents would work better when coupled with a causal diffusion planner/policy.
>
>
> - **Block-wise latent identification guided by the theory**
>
>
> Prior methods often aggregate long trajectory segments into a single latent summary, which can contain redundant information and does not guarantee element-wise identifiability. In contrast, our approach is guided directly by our identifiability theory: for RL (POMDP setups), we show that a **minimal** temporal block containing both past and future observations is **sufficient** to identify the latent factor at each time step (and that access to future observations is essential). Accordingly, our latent module explicitly operates on these small, theoretically-motivated temporal blocks to recover per-step latent variables, instead of compressing long trajectories into coarse summaries.
>
>
> - **Integration into a causal diffusion framework via novel mechanisms**
>
>
> Beyond the use of variational objectives, our technical novelty also comes from how these identifiable latents are integrated into a diffusion-based decision-making model. Concretely, we introduce (i) an autoregressive noise schedule that reflects the temporal causal structure, (ii) a denoise-and-refine procedure that alternates between latent inference and denoising, and (iii) a zig–zag sampling strategy that enables online latent refinement without access to future observations. Together, these components work in synergy to make the diffusion process faithfully mirror the true RL data-generation mechanism under latent contexts.
>
>
> Among them, (i) is related to existing autoregressive diffusion designs, whereas (ii) and (iii) are newly-proposed in our setting and are specifically motivated by the identifiability requirement and the offline–online mismatch. As a result, the generative process is repeatedly forced to align denoising with the evolving latent context, allowing the diffusion model to exploit latent factors much more effectively than standard reconstruction-based latent embeddings.
>
>
> **Related Revisions**: We added discussion on related works you suggested in both `Sec 4.2` (Page 6) and `Appendix E.2`. Also, we added an explanation on the latent identification in `Sec 4.2` based on your comments.
>
> ---

---

> ### Author Response · Authors · 2025-11-23
> **Author Response - Part 2**
>
> > It would be helpful to elaborate on why, intuitively, the extracted latents would work better when coupled with a causal diffusion planner/policy.
>
>
> **R2**: Temporal latent identification and diffusion models act in synergy to learn the underlying structure of sequential decision-making. We clarify the intuition from two complementary perspectives: (i) why identifiable latents benefit a causal diffusion planner/policy, and (ii) why the diffusion architecture in turn enhances latent identification.
>
>
> **Latents -> Diffusion** As discussed above, once the latent variables are properly modeled, the diffusion process can more faithfully represent the true data-generation mechanism of RL trajectories. The latent factors capture slow-changing or unobserved influences on transitions and rewards, and our causal diffusion model explicitly conditions the denoising trajectory on these estimates.
>
>
> - The **autoregressive noise schedule** enforces the correct temporal dependence among $(s_t,a_t)$ pairs and the latent $c_t$.
>
>
> - The **denoise-and-refine mechanism** lets the diffusion model repeatedly update the trajectory using progressively more accurate latent estimates.
>
>
> - The **zig–zag sampling** further ensures that the generated trajectory and latent context remain consistent, even during online sampling where future observations are unavailable.
>
>
> Together, these mechanisms allow the planner/policy to reason over latent-conditioned transition dynamics rather than treating the environment as fully stationary.
>
>
> **Diffusion -> Latents**  Conversely, the diffusion architecture naturally supports accurate latent inference, for two reasons.
>
>
> - **Multi-step denoising aligns well with our identifiability condition and implementation.**
> Theorem 1 indicates that small temporal blocks containing both past and *future* steps are necessary for identifying $c_t$. Specifically, to identify the latent context $c_t$, whose posterior depends on future observations (e.g., $x_{t+1}$, **guided by Theorem 1**), we introduce a backward refinement step.
> Diffusion model thereby provides *progressively refined* intermediate predictions along the process.  We first denoise $x_t$ into a partial state $x_t^{k_1}$ using $x_{t-1}^0$ and an initial estimate of $c_t$ sampled from prior, then refine $c_t$ with $x_{t+1}$ to obtain the final $x_t^0$. During training, we enforce this backward refinement to satisfy the theoretical identifiability conditions. At inference time (zig-zag sampling), we substitute $x_{t+1}$ with a predicted estimate.
> These intermediate estimates act as “(soft) future observations,” enabling the posterior network to approximate the required block $x_{t-k:t+1}$ even at test time.
> Hence, diffusion’s iterative denoising gives us exactly the structure needed to approximate the future-augmented block and recover $c_t$ online.
>
>
> - **Diffusion is a strong backbone for modeling RL/IL trajectories.**
> Prior work such as Diffuser, Diffusion Policy, and Diffusion Forcing has shown that diffusion models provide expressive multi-modal predictive distributions, stable training, and strong performance in sequential decision-making. Using it can make that the latent module receives high-quality, temporally consistent denoised signals, which further stabilizes and improves latent recovery and policy learning.
>
>
> In summary, identifiable latents and causal diffusion reinforce each other: latents make diffusion-based planning more accurate and adaptive, while diffusion provides the temporal refinement structure needed to identify latents reliably, even under partial observability and in the absence of future observations during inference.
>
> **Related Revision**: We have added the above discussion to `Appendix A.4` (Page 22) as a good supplement to the main paper.
>
> ---

---

> ### Author Response · Authors · 2025-11-23
> **Author Response - Part 3**
>
> > In this work, specifically, the latent variable is modeled using (s, a, r) triplets with an extra reward; however, when using the latent variable, the planner generates sequences of states or state-action pairs without explicitly accounting for accumulated rewards.
>
> **R3**: We use $(s, a, r)$ due to the consideration of the general cases where
> the latent context $c_t$ is allowed to influence **dynamics, rewards, and even actions** simultaneously. This is exactly the data-generating structure assumed in our latent contextual POMDP/SCM (Main paper Fig. 1), where
> $c_t = h(c_{t-1}, \eta_t)$,
> $s_t = f(s_{t-1}, a_{t-1}, c_t, \epsilon_t)$,
> $a_t = \pi(s_t, c_t)$,
> $r_t = g(s_t, a_t, c_t, \delta_t)$.
>
> Therefore, observing only $(s,a)$ would not (in general) provide sufficient information to identify a latent that may also manifest through rewards.
>
> Second, when Ada-Diffuser is used for planning or policy learning, rewards **are not ignored**. Following standard diffusion-based decision-making backbones (e.g., Diffuser/DP/DF), we incorporate rewards (or cumulative ones derived from them) through diffusion guidance. Concretely, we train the conditional diffusion model with reward-related guidance (reward or goal/value signals depending on the setting) (`2nd paragraph, sec 4.1`). Thus, accumulated rewards influence the generated trajectories through the same guidance mechanism used in prior diffusion planners/policies, with the additional benefit that our denoising is conditioned on the inferred latent context.
>
> **Related Revision**: We have highlighted this in `sec 4.1` (Page 5).
>
> ---
>
> > Even though dynamics can be modeled by the latent, the planning performance may remain restricted by the quality of offline demonstrations.
>
> We fully agree that this is a general limitation of **all** generative planners trained purely from offline data. However, we note that explicitly modeling the latent factors can *mitigate* this limitation. By recovering the latent variables that capture stochasticity, nonstationarity, or unobserved structure in the offline trajectories, the model can produce rollouts that better match the underlying dynamics, even when the demonstrations are imperfect. This is also reflected in our experiments: **even in settings without explicitly designed latents**, Ada-Diffuser achieves consistently higher performance, suggesting that latent modeling helps compensate for suboptimal or heterogeneous offline data by capturing its underlying variability.
>
> **Related Revision**: We have highlighted this in `Sec 5.2` (Page 9).
>
>
> ---
>
>
> > Line 422: Missing content “Table A4 and are“
>
>
> Thanks for catching this. We have corrected the typo.
>
>
> ---
>
>
> Again, thank you for your insightful and constructive comments. We hope the discussions above address your concerns clearly. As noted, we have added some discussions accordingly in the revision and highlighted the corresponding changes in the main text. If you have any further questions or suggestions, we would be very happy to discuss them and provide additional clarification.

---

> > ### Author Response · Authors · 2025-11-27
> > **Thank you for your thoughtful review**
> >
> > Dear Reviewer SLzj,
> >
> > Again, we would like to thank you for your time and effort in reviewing our paper. All suggestions and comments were very helpful, and we have provided detailed discussions in the rebuttal.
> >
> > In **R1–R2**, we clarify why and how the latent identification algorithm is designed, and in particular how it works synergistically with diffusion models and our novel algorithmic designs. In **R3**, we explain the role and importance of reward in our model, and in **R4** we discuss the offline–online gap and why our method can outperform baselines in this setting. We also point to the corresponding revisions.
> >
> > Thank you again for your constructive feedback and for helping us improve the work.
> >
> > The Authors

---

### Author Response · Authors · 2025-11-26
**General Response (1/2)**

Thank you again for all the thoughtful reviews and insightful suggestions.

We are really encouraged that the reviewers found the paper’s contributions meaningful and well-supported. Several reviewers highlighted the value of our **theoretical analysis**, describing it as “theoretical analysis on latent identification with minimal observations” (`SLzj`), “a meaningful step” (`ZU5m`), “the theoretical grounding adds credibility” (`ZU5m`), “a principled justification” and “clearly presented” (`DUvX`), as well as “a novel theoretical contribution that directly guides the algorithmic design” and “a strong foundation” (`5j5o`). Reviewers also noted that our **algorithm framework** is “elegant” (`ZU5m`), “addresses an important gap with a unified approach” (`DUvX`), and exhibits “high originality, non-trivial synergy, and substantial improvements” (`5j5o`). On the **empirical side**, the reviewers described our experiments as “extensive”, demonstrating “effectiveness and robustness” (`SLzj`), with “robust results” (`ZU5m`), “broad coverage” (`DUvX`), and “comprehensive evaluation” (`5j5o`).

Below, we provide a general response summarizing the main recurring themes across the reviews, along with pointers to where each concern is addressed in detail and the corresponding revisions in the manuscript.

 **Common Questions**

- **About assumptions 2-3** (`ZU5m`, `DUvX`)

Several reviewers asked about the realism of Assumptions 2–3 and how they behave under stress tests. In the rebuttal, we expand the conceptual explanation and show that these assumptions are mild: *they essentially require that the latent context has a meaningful effect on dynamics, actions, or rewards, precisely the situations in which identifying and modeling the latent is necessary for decision-making*.

We did the **empirical stress testing on both assumptions**. Specifically, we validate this in MuJoCo Cheetah by directly estimating the corresponding operators and spectral ratios. We find that in regimes where the latent context has a clear effect (e.g., wind that significantly perturbs dynamics), the assumptions are clearly satisfied. In contrast, when we artificially weaken the latent influence so that it almost does not affect the environment, the assumptions fail, but in these regimes, policy performance is essentially unchanged whether we model the latent or not, so identification is neither possible nor needed.

We further sweep over different settings and measure both assumption realism (transition injectivity and spectral distances) and policy performance gaps. Across these stress tests, we observe a consistent pattern: *the more strongly Assumptions 2–3 are satisfied, the more important it becomes to model and identify the latent context, as reflected by larger performance drops when the latent is ignored*. This supports our claim that the assumptions are **mild, practically relevant, and directly tied to when latent-aware decision-making is beneficial**.

Details are given in our responses to `ZU5m` (**Part 3-4, R4**) and `DUvX` (**Part 3-4, R3**). Revisions are in the main paper (boxed in `Sec. 3.2, Page 4`) and `Appendix B.5` provides empirical evaluations of these assumptions, and `Fig. A.1` gives the quantitative results.

- **About zig-zag sampling** (`DUvX`, `5j5o`)

Several reviewers raised concerns about that at test time, future observations are not available, and whether our zig–zag refinement might propagate error by relying on predicted futures. As noted in the detailed responses, zig–zag is explicitly designed to mitigate this mismatch. Algorithmically, it combines: (i) **a contrastive improvement loss** during training that teaches the model *how true future observations refine the posterior relative to the prior*, (ii) the use of a **lightly noised predicted future at test time** (rather than a heavily corrupted or fully generated state) to limit error propagation, and (iii) **global diffusion matching** over full trajectories, which *regularizes the generative process and absorbs residual mismatch*.

Empirically, we quantify the effect via **probing MSE** for recovering the ground-truth latent: the oracle model (with true future) achieves 0.12, our full method 0.18, while removing refinement or zig–zag degrades to 0.28 and 0.23, respectively. The **small gap to the oracle** (0.06) and the clear deterioration without zig–zag show that predicted futures are informative enough for stable posterior alignment, and that zig–zag materially improves latent identifiability and downstream performance despite the inherent train–test difference.

Details are given in our responses to `DUvX` (**Part 1, R1**) nd `5j5o` (**Part 1, R2**). Revisions are in the main paper (`Sec 5.3`, Page 10).

---

> ### Author Response · Authors · 2025-11-26
> **General Response (2/2)**
>
> - **About computational efficiency** (`ZU5m`, `5j5o`)
>
> Several reviewers asked about the extra cost of latent modeling and zig–zag refinement. As detailed in `Appendix I.1`, across all environments our method incurs only a moderate overhead of about only **1.2–1.3×** in training time and planning latency (≈20–30% extra), even though this already includes the latent inference module. Moreover, a Picard-accelerated variant reduces inference time to 0.7–0.8× of our default sampler while preserving performance, showing that Ada-Diffuser remains compatible with standard acceleration techniques and is practically deployable.
>
> Details are given in our responses to `ZU5m` (**Part 4, R5**) nd `5j5o` (**Part 2, R3**). For revision, please refer to `Appendix I.1` (`Table A12–A13`) for the full updated compute profile. We also added a pointer to these results in `Sec. 5.3` (Page 10) of the main text.
>
> - **More evaluations**
>
> (i) More complicated benchmark: MimicGen with cases on embodiment shift; noisy observations and suboptimal demonstrations (`ZU5m`; **Part 1, R1-2**); (ii) More ablations on the dimensionality of latents and temporal latent transitions (freeze or not; w/o latents) (`ZU5m`; **Part 5, R7** and `Sec 5.3` in the main paper);  (iii) More distribution shift modes: within-episode abrupt changes and episodic changes (`ZU5m`; **Part 6, R9** & `5j5o`; **Part 5, R11**).
>
> ---
>
> Other specific questions and all corresponding revisions are addressed in the individual responses below and reflected in the revised manuscript. Again, we sincerely thank you for the time and effort you invested in reviewing our work, and we are grateful for the opportunity to discuss it with you. If you have any further questions or suggestions, we would be more than happy to continue the discussion.

---

### Meta-Review · Area_Chair_FesX · 2026-01-04

**Summary:**

I recommend accepting the paper. I have summarized the main concerns of the reviewers below as well as how they are addressed by the authors, which seem sufficient.

**Reviewer Concerns:**

Main concerns:

Reviewer SLzj
- Mismatch between how latent variables are encoded and how they are used
- - The authors clarified that the mismatch is apparent but the mechanism resolves it

Reviewer ZU5m
- Missing strong POMDP or belief-tracking and latent-variable baselines
- - Authors clarify that they are included
- The paper claims action recovery, but needs clear quantitative
 action-recovery metrics and head-to-head comparison.
- - Authors clarify that some evaluations are already included
- The core contribution is latent inference, but sensitivity and “remove or freeze latents” tests are limited.
- - Additional ablations provided while clarifying that some are already included


Reviewer 5j5o
- Mismatch between theoretical and practical block sizes
- - Author clarified that the theory holds but larger block sizes help due to finite-sample and function-approximation effects
- Unavailability of future observation during online inference
- “While the overall framework is original, the individual components draw heavily from existing literature”
- - The author clarifies that some of the components are novel too

**Reviewer Scores:**

I would have increased the slightly negative scores to positive given the author rebuttals.

---

### Decision · Program_Chairs · 2026-01-26

Accept (Poster)